# The Poetics of Physics

**Chris Jeynes** [1,*] , **Michael C. Parker** [2] and **Margaret Barker** [3]

1    Ion Beam Centre, University of Surrey, Guildford GU2 7XH, UK
2    School of Computer Science & Electrical Engineering, University of Essex, Colchester CO4 3SQ, UK
3    Independent Researcher, Borrowash DE72 3HT, UK
*    Correspondence: c.jeynes@surrey.ac.uk

**Abstract:** Physics has been thought to truly represent reality since at least Galileo, and the foundations of physics are always established using philosophical ideas. In particular, the elegant naming of physical entities is usually very influential in the acceptance of physical theories. We here demonstrate (using current developments in thermodynamics as an example) that both the epistemology and the ontology of physics ultimately rest on poetic language. What we understand depends essentially on the language we use. We wish to establish our knowledge securely, but strictly speaking this is impossible using only analytic language. Knowledge of the meanings of things must use a natural language designed to express meaning, that is, poetic language. Although the world is really there, and although we can indeed know it truly, this knowledge is never either complete or certain but ultimately must rest on intuition. Reading a recently discovered artefact with a palaeo-Hebrew inscription as from the first century, we demonstrate from it that this ontological understanding long predates the Hellenic period. Poetic language is primary, both logically and temporally.

**Keywords:** thermodynamics; info-entropy; ontology; epistemology; poetry; palaeo-Hebrew

## 1. Introduction

This essay has in mind Marshall McLuhan's idea, "*The Medium is the Message*" [1]: our epigraph and epilogues are *poems* because the very purpose of the work is to establish the idea that we always *mean* more than we *say*; an idea that is true even in—perhaps *especially* in–doing physics. So the canonical textbooks of physics (for example, Landau & Lifshits [2]) are notoriously brief. Why are they brief? Because the students are supposed to *grasp* the material, and fill in the "trivial" (actually very challenging) gaps in the treatment for themselves. It is this activity of *grasping* that we will focus on here.

We will seek to prove that the terms which we use to *understand* any material, here especially including the material of mathematics and physics, are rooted in a poetic use of language in which ambiguity cannot be eliminated. The very terms used for our opening and closing poems, *epigraph* and *epilogue*, make use of the ambiguity of the Greek prefix έπί (*epi*) which can mean (among other things) either "upon" (that is in this case "before") or "in addition" (that is in this case "after").

The renowned physicist Carlo Rovelli has been dubbed "*the poet of physics*" by Richard Webb in his review of one of Rovelli's popular science books, which Webb calls "*enriching, illuminating, eclectic*" [3]. This present paper is "eclectic" since it ranges over subjects not usually considered together (physics, poetry, palaeo-Hebrew); but then the issue is reality itself, and the boundaries we erect around our disciplines are merely for our own convenience [1]. For example, the DDT molecule behaves the way it behaves completely independently of whether we are considering the applicable quantum mechanics, or its enthalpy of formation, or the chemical and process engineering needed for its industrial production, or the regulatory issues related to its safe use, or the political issues raised seminally by Rachel Carson who took her title "*Silent Spring*" [5] from a poem by Keats (1819) [6].

## Epigraph: "*Ku wown biyuke*" (losing language hurts!)

| | |
|---|---|
| Ku wown biyuke | If our language is lost |
| nikwe ukanuhwan amin madikte arikna inurikyene, | then all our knowledge of things above |
|    (warukma, kamuw, kayg) |    (stars, sun, and moon) |
| hawwata ukannuhwan umin wis amadgaya inin, | and the knowledge of us humans on earth |
|    (uhiyakemni akak uwakemni) |    (our thoughts and our deepest feelings) |
| in ka ekkepka akisyavrik akiw | will not be properly expressed again |
| ewka awen wownavrik. | when our language is gone. |
| | |
| Ku wown biyuke | If our language is lost |
| nikwe madikte amadgaya inin, | then everything in the world, |
|   (parahwokwa, warik, puwiknebdi akak ahavvi) |   (seas and rivers, animals and plants) |
| in ka kinetihwaka nimin akiw, | may never again be spoken |
| akak uhiyakemni payak akak uwegewni | with our understanding and insights |
| mmanawa in kuwis menwe. | for these will have already vanished. |
| | |
| Ku wown biyuke | If our language is lost |
| in ke wotbe pahayku lapot sabukwiyebe. | it will be as though a door were closed |
| Nikwe hiyeg amedgenevwi inin | to the peoples of the world |
| awetuvye pukuha | and they will never understand |
| ku samah wowskawni ay amadga inin. | how we lived here on earth. |
| | |
| Ku wown biyuke, | If our language is lost |
| unetni adah kiyathaki akak amnihka | our words of respect and love, |
| unetni adah kayahka akak batekka | our expressions of pain or fondness |
| wavan, westwa, unetni, uvigyepkawni, | our songs, our stories, our talk, our prayers, |
| amekenegben gikehnikis | the accomplishments of our ancestors |
| in ka kinetihwakati nimin akiw. | will never be spoken of again. |
| | |
| Ku wown biyuke, | If our language is lost |
| — aa, ka aynsima iwit kuwis biyuke, | — oh, many languages have already been lost |
| ka aynsima iwit biyuknene akiw, | and many more are almost lost, |
| kewa pahak waruwbe bekbetepka aritnanyuvwi — | like mirrors forever shattered — |
| nikwe wahawkrivwiy gikuvimnakis | then our ancestors' voices |
|    tinwohawsepka adah avavyekwa |    will be silenced forever |
| in ke igiskabe ku pariye wis biyukse adah avavyekwa. | and a great treasure will be forever lost to us. |

**Epigraph: "Ku wown biyuke".** A poem in the Palikur language (© 2016 Aldiere Orlando, reproduced by permission with translation by Diana Green © 2020) after Miguel Leon-Portilla's famous poem: *Cuando Muere Una Lengua* (1998; in *Náhuatl*). See Supplementary Materials for the audio file of the poet speaking the poem in *Palikur* (and for its *Portuguese* translation), also for the *Náhuatl* original of *Cuando Muere* Una Lengua (and its translation into *English* from the *Spanish*).

Tom McLeish opens his book [7] with a "*powerful list of words*", that is: "*Creativity, Inspiration, Passion, Imagination, Composition, Representation*" (the capitalisation is his), which he proceeds to argue apply to the sciences just as much as to the humanities. He quotes Karl Popper [8] saying, "*A great work of music, like a great scientific theory, is a cosmos imposed on chaos—in its tensions and harmonies inexhaustible even for its creator*". McLeish asks, "*Is a dualistic division into arts and science really faithful to our history, our capacities and needs?*" arguing that we should not "*reinforce the well-worn narrative*" of the "*Two Cultures*" [9]. Oliver Sacks [10] echoes McLeish, saying that "*science is far from being coldness and calculation . . . but is shot through with passion, longing and romance*". Above all, we should not get lost in our own specialisms! Graeber and Wengrow [11] are sarcastic about "*specialists refusing to generalise*".

We believe that our account of *science* should acknowledge the seminal contribution of *Inspiration* to the process of gaining knowledge, and this essay is an attempt to do this. Of course, this is not a new idea although every generation seems to need to find it out anew: in previous generations, the great Henri Poincaré (for example) also considered *intuition* and *inspiration* as basic an ingredient in physics as it is in poetry (see Szpiro's review of Gray's biography [12,13].

We wish to know *what things are* (that is, their ontology), and we also wish to know *how we know* what things are (that is, their epistemology). We seek here to explore the idea of *knowledge* itself, and to do this we will have to go beyond the usual Hellenocentric accounts that lead us to believe that philosophy started with Socrates (or at least, with the pre-Socratic Greeks) [2]. *Knowing* is a characteristic human activity that in all the ancient societies we know of has been linked to *seeing* (which is why the wise were called "*seers*"), and the "*blind seer*" is an ancient archetype [3]. We will explore the roots of our ideas of knowledge since it is obviously a fundamental error to think that true things could have been known only in modern times.

The trouble with this is that there exist today widespread prejudices not only that for knowledge to be "true" it must be "scientific" (with poets operating somehow on a different plane), but also that any talk that may be called "religious" is necessarily irrelevant to science, even though what we now think of as "religious" ideas pervaded all ancient poetry. Michael Polanyi has investigated the far-reaching philosophical consequences of having a false conception of what science is for, and how it is done: "*As long as science remains the ideal of knowledge, and detachment the ideal of science, ethics cannot be secured from complete destruction by skeptical doubt*" (Polanyi & Prosch, 1975 [22]).

However, no one has a monopoly on talking nonsense, and in any case we insist on the unity of truth [4]. We are investigating *how we come to know* new things, and our discussion will range from the details of modern developments in thermodynamics to the ancient poets composing in an ancient Hebrew: the fact that this poetry is now pigeon-holed as "religion" here concerns us not at all. Instead, we wish to point to the characteristic humanity of both the poetry and the science: we will show that the knowledge of both the poet and the scientist is, ultimately, the *same sort of thing* (even if they use vastly different methods). After all, both the poet and the scientist want to explain reality, which may be seen in multiple different (but complementary) ways.

It is also often thought today that scientific concepts are not constructed in a "natural" language, being higher order abstractions, and therefore that other accounts (such as the present one) are simply irrelevant. It is of course certainly true that modern physics is normally discussed by physicists in eye-wateringly sophisticated mathematical terms (and the thermodynamics we will describe is no exception); nevertheless, we will demonstrate that at the foundations of every field of physics (with thermodynamics as our example) is a "natural" language explanation of how to *grasp* (or *see*) the phænomena of interest. Mathematics is required to expose logical consequences, but words are required to illuminate meaning.

*Structure of the Paper*

The paper is constructed as an essay on ontics and epistemics (what things *are* and how we *know* them: more properly, since they are inseparable, "*onto-epistemics*"(Karen Barad's idea [24]). We will show that *knowing things* (even in physics) is properly a poetic activity: knowledge is necessarily *personal* and cannot be obtained mechanically, it is the sort of thing that can only be obtained by a "poetic" (or transcendental) approach.

We start by considering *meaning* (§2). Helping us to touch the deep meaning of things is obviously the job of poets: what is not usually appreciated, and what we wish here to underline, is that students of physics always ask the same sorts of questions when confronted with a complex mathematical argument—*what does it mean?* And such questions are (and always have been) fundamental to doing physics. It is not widely enough known that physics cannot be done without imagination.

Exploring the *thinginess of things* (§3; that is, the rational structure both of reality itself and of our knowledge of it) is surprising. Poets have always delighted in the thinginess of things: William Carlos Williams' line "*no ideas but in things*" [25] has become a cliché. But what is not sufficiently appreciated is the logical status of the knowledge of things. We explore Gödel incompleteness in its historical and philosophical context, and its disturbing

consequences for the ideas of "subjectivity" and "objectivity" (and the "scientific method") that we had thought were straightforward (even "obvious"). They are not!

We then, separately, summarise the surprising development of the ideas of *entropy* and *information* (§4) as a specific example of how *meaning is negotiated* (Martin Edwardes' idea [26]) in physics. This is a complicated story, full of arcane detail, of which our very brief summary is barely adequate. But we hope it is full enough to appreciate the complexity of knowledge even in this very restricted field. Usually, a discipline is presented pedagogically as timelessly elegant, with no loose ends either technically or philosophically. But this is not the way things are! If we wish to understand how the scientific method actually works, with all its limitations, we need to go into the details of how knowledge is really obtained.

We underline (§5) this *negotiating of meaning* in the development of knowledge as being an exercise that necessarily involves poetics, even concluding with a discussion of some canonical poetry (§5.4). We discuss what *definitions* involve (§5.1), the necessary status of this whole discussion as *metaphysical* (§5.2), emphasising that the pejorative meaning usually ascribed to this term is not the only available meaning. Also included is an important discussion of *ambiguity* (§5.3).

The whole essay revolves around the recognition of *language* as the primary and essential medium of knowledge, and we give an example of this (§6) that uses a detailed analysis of an artefact that is demonstrably a mnemonic of a very sophisticated view of *knowledge* long predating the Hellenic schools of philosophy. It is a reasonable approximation to say that what we now call "philosophy" started with the Greeks, but of course people have always wanted to know things, and we demonstrate a very ancient and intricate account of how to think about things that is startlingly profound. No one ever talks seriously of modern physics and ancient artefacts with palaeo-Hebrew inscriptions in the same breath, so that this example of ours is very surprising. Nevertheless, it underlines our thesis that *analytic language* is necessary but not sufficient to *know* even physical things adequately. *Poetic language* is also required.

We gather the threads of the argument together (§7) and finally conclude (§8).

## 2. Meaning as Poetry

In what way can a scientist be like Shakespeare? Tom McLeish recently quoted Shakespeare's 100th Sonnet ("*Where art thou Muse . . .* ") saying, "*it has never been easy to speak with clarity* about *moments of imaginative conception*" ([7], p. 7), and we will also quote Dante Alighieri also speaking of his Muse (§5.4). McLeish eloquently discusses a variety of cases showing how scientists *imagine* reality before they are able to establish their new theories, and how these imaginative (creative) approaches are actually central because of "*new patterns and connections that they offer for specific creative demands*" ([7], p. 331). Seeing new things requires imagination!

Almost a century ago, Owen Barfield famously spoke of "*poetic diction*", that is: "*the language of poetic* compositions" ([27], III:5):

> When we start explaining the language of famous scientists as examples of '*poetic diction*' . . . [it is no] waste of time [if it helps anyone to be convinced] how essentially parochial is the fashionable distinction between Poetry and Science as modes of experience
>
> Owen Barfield, *Poetic Diction* VIII:6 (1928) [27]

seeking to establish, like McLeish, that epistemologically there is little distinction between artists and scientists: they are all similar in how they come to know new things.

If I say, "*information has calculable entropy and obeys physical laws*"[28] [5], what do I mean? And how can you understand me? Barfield says that "*the poet's relation to terms is that of maker*" [27] (VIII:4) [6]; *information* and *entropy* here are terms referring to certain aspects of physical reality and it is clear that the terms are made by the physicists: are they (as both Barfield and McLeish outrageously seem to say) in some sense thereby <u>poets</u>? Indeed,

Oliver Sacks ([33], p. 23) speaks of a time when "*there still existed a union of literary and scientific cultures; there was not the dissociation of sensibility that was so soon to come.*"

We do not think that physicists ought to be poets, nor even that at least *some* physicists should. We regard such a position as absurd. But we do think that *ultimately*, physicists cannot avoid using language "poetically": that is, using a "natural" language [7] (together with its unavoidable ambiguity) to set up the model proposed for the phænomenon in view. As an example of this we will explore the specific case of how we address the scientific concepts of *entropy* and its close companion *information*, which together represent difficult ideas in a currently very active (and contentious) field of research. We point out that the very close relation between information and entropy is now well established by workers who articulate this relation in mathematical detail as a "new" concept of *info-entropy* within the overall theory that they call "Quantitative Geometrical Thermodynamics" (QGT) [28].

Using these test cases, and explicitly using one of the first papers on entropy [35], we seek to show how the development of scientific ideas necessarily depends in the first instance on an intuitive understanding that relies on intrinsically poetic language. We emphasise that "poetic language" is not restricted to poetry! Even in 1928 Owen Barfield recognised that "famous scientists" used "poetic diction". The basic ideas of any theory have to be "negotiated" [26] [8] using some sort of "natural" language, and any subsequent mathematical representation is only a formal method of displaying the logical consequences of these ideas.

This assertion is disturbing since it is widely thought today that there is a sharp distinction between the "hard sciences" in which things are known certainly (or at least, pretty much certainly for practical purposes) and the humanities which (supposedly) prize feeling above thought. Supposedly, everyone agrees in science, but no one agrees in politics, philosophy and religion. But we point out that knowledge is fuzzy: the "hard sciences" are not as hard as we might like them to be. The old joke (repeated by Rabbi Weinreb) goes, "*two Jews, three opinions*" [37], but Weinreb himself points to the value of debate in "*an atmosphere of civility and mutual respect and a willingness to concede one's original position in order to achieve the truth*". Who would disagree with that? It turns out on closer inspection that the "scientific method" [9] is more poetic than we might have expected.

Summarising the programme of this essay: before a scientific concept can be understood, it must be articulated, and language is essential to articulate scientific ideas: we cannot *know* anything without being able to *say* what it is we know (without language we have inchoate feeling, not *knowledge*). Science is effected by humans acting humanly—that is, using language! Stones do not know things: people do. Our knowledge of the world is necessarily based ultimately on intuition [10], and the articulation of intuited knowledge is ultimately the business of poets. Before it is anything else, natural language is poetic.

Saying that *knowledge is necessarily mediated by words* sounds like the *linguistic determinism* famously proposed by Benjamin Lee Whorf (1941) [40]. We do not take this position, but rather that of the "*relay results*" advocated by McLeish [7] who relies on Jacques Hadamard's *The Psychology of Invention in the Mathematical Field* [41]:

> . . . James Clerk Maxwell would urge mathematicians to formulate their thinking in 'words without the aid of symbols', not because he would sympathize with the linguists, but because he knew the creative force of communicating ideas

> Tom McLeish (2019) p. 243

where of course Maxwell was one of the giants honoured by all subsequent physicists for his beautiful (and seminal) description of the electromagnetic field [42], on which all modern technology depends. McLeish understands the importance of using *words* to explain the meaning of the work, as do all good scientists: when we write papers, the Introductory section almost always eschews mathematics [11], and McLeish is asserting that this should not be understood as a pedestrian way of explaining the significance of the work but rather as "*communicating ideas creatively*". Paul Sen [45] (p. 79) says the same of Clausius' ground-

breaking papers: "*[Clausius] persuades the reader in ordinary language before backing up his reasoning with formulae and algebra . . . *".

We note that McLeish explicitly considers the parallels between scientific creativity and the (wordless) creativity of painters and musicians: that is, there does exist a "knowledge" that is **not** mediated by words [12], but this wider view of knowledge is outside our present scope. Michael Polanyi also considered such knowledge, which he called "**tacit**" [13]; here we only consider scientific knowledge from the point where it becomes crystallised in words:

> The formulation of the fruitful question, posed in the right way, constitutes the great imaginative act in science
>
> Tom McLeish *The Poetry and Music of Science* p. 10 (2019)

We are also distinguishing sharply between "*information*" (which is physical) and "*knowledge*" (which is mental). *I know* precisely because *I am informed*. Stones incorporate *information* from which geologists can glean *knowledge* [14]. Similarly, Oliver Sacks [49] (p.255) observed that *information*, however "*wide-ranging . . . [is] different from knowledge*", being "*centerless*" [15].

The thesis of this paper is that where physics must use *analytic* language, metaphysics [16] must involve irreducibly *poetic* language. Language is intrinsically metaphorical: all our words have concrete referents but none of them is *merely* concrete, they all come with a cluster of connotations. Iris Murdoch [57] observed long ago of metaphors:

> Metaphors are not merely peripheral decoration or even useful models, they are fundamental forms of the awareness of our condition . . . it seems to me impossible to discuss certain kinds of concepts without the resort to metaphor, since the concepts are themselves deeply metaphorical and cannot be analysed into non-metaphorical components without a loss of substance.
>
> Iris Murdoch, *The Sovereignty of Good over other Concepts* (1967)

and W.B. Yeats [58] was making a similar point in 1900 when he insisted on "*The Symbolism of Poetry*" which he claimed (and he should know!) involves a "*continuous indefinable symbolism, which is the substance of all style*" (*ibid*. §II). Yeats speaks of poets (and others) "*making and unmaking mankind*" (*ibid*.—he repeats this phrase!); that is, the *symbolism* of poetry (or the *metaphor* it embodies) actually touches the *substance* of things, just as Murdoch says [17]. Yeats underlines this when he asserts that "*meanings . . . are held [together] by the bondage of subtle suggestion*" (*ibid*. §IV). "Substance" is a resonant word in physics, and also in European philosophy ever since the Cappadocian Settlement [18] in the 4th century CE. It is indicative that both Murdoch and Yeats refer to "*substance*".

A "*metaphor*" (after the ancient Greek μεταφέρειν, to transfer) can be thought to *translate* (or transfer) between elements of this connotation cluster, and this idea of "*translation*" is essential to our thesis [19]. We will show (using the particular case of *entropy*) that the narrative of physics is only established in the context of a metanarrative (in this case called "metaphysics") which constructs the meanings of the ideas to be used in a natural language as unambiguous as possible. This metaphysical step is usually carefully ignored by philosophers of science: Nicholas Maxwell's "*aim-oriented empiricism*" approach (predicated on the *metaphysical priority* of unified theories) is a welcome exception [63]. But *standard empiricism* glosses over the idealistic foundations of how we *interpret* observations [20].

There is a complexity here. We believe that Maxwell's insistence on the *idealistic* nature of physics (since we always prefer unified theories, however wrong they might be) does not affect the common view of physicists that successful theories are *true*. That is, physicists are usually both realists and idealists. Logically, these two attitudes appear to be mutually exclusive: how then can they be compatible (if indeed they are)? We acknowledge that the naïve realist [21] and the naïve idealist positions are both untenable, but we will argue here for the truth that the physicist needs an *idealist* approach to recognise a

promising theory, while depending on a philosophical attitude that regards the world as real, rational, and comprehensible in principle (that is, being some sort of *realist*). And formally, this philosophical attitude must be couched in a 'natural' language (however tacitly), there being no alternative. Of course, one's underlying philosophical attitudes are rarely made explicit.

Note that natural language is always ultimately poetic, especially where new meanings are being created. Meaning is always *negotiated* between speakers, and poets find new and resonant ways of doing this: Martin Edwardes (2019, [26]) has shown how this *negotiated meaning* must be central to ontology. When scientists establish new concepts, they must "negotiate the meaning" of the terms they use for these new concepts. We will show here how this works in the case of *entropy* (and *info-entropy*).

Understanding physical concepts therefore always involves an intuitive leap in meaning from the concrete to the metaphysical, which we could also arguably (and nearly equivalently) call the *spiritual*. The very word *spirit* exemplifies this intuitive leap. Today the English word *spirit* has a range of metaphysical connotations, but in the original Latin it also carried the concrete meaning *wind* (which English word has an Anglo-Saxon etymology). So for example, there is a Greek record of Jesus' saying (John 3:8):

το πνευμα οπου θελει πνει . . . που υπαγει ουτως εστιν πας

ο γεγεννημενος εκ του πνευματος

Textus Receptus (<70 CE [22])[67]

Spiritus ubi vult spirat . . . sic est omnis qui natus est ex spiritu

*transl:* Jerome (c.400 CE) [68]

The wynde bloweth where he listeth [where it wills] . . .

so is every man that is boren of the sprete [born of the spirit]

*transl:* Tyndale (1526) [23] [69]

where we give William Tyndale's highly influential English translation, standard (in the form of the largely derivative 1611 *King James* version) until at least the 1960s (the "*New English Bible*" was only published in 1961).

Note that cognates of the same word are used in both Greek and Latin (πνευμα, πνει, πνευματος/*spiritus, spirat, spiritu*) where three different words are needed in English (*wind, blow, spirit*). Translation of nuance is irreducibly creative: both Jerome and Tyndale had poets' ears.

Speaking of "spirit", we should perhaps mention the perennial "mind/body" debate (summarised helpfully by Brian Dolan, 2007 [70]): is "mind" expressible in terms of neurological function? Should we regard "mind" (whatever that is) as "emergent" from body? Is "mind" ultimately reducible to matter? Dolan shows that the materialists certainly have not settled these questions in their favour: they all remain open. We will show that new results in entropy (Velazquez 2022 [71], Velazquez et al. 2022 [72]) point to the value of a "holistic" (*not* reductionist) approach (see §4.6 *passim* below).

Returning to the original question, what is *entropy* and what is *information*? These are ontological questions. How do we understand entropy and its relationship to information? These are epistemological ones. To answer these questions, we have to translate from the concrete to the general; that is, from specific observations to an articulation of a coherent theory. We will proceed to explore these issues, taking as examples the meanings of "*information*" and "*entropy*". Our thesis is that moving from the concrete observation of physical reality to the general articulation of a physical theory we cannot avoid brushing with the spiritual (in the sense explained above, which in this context would also usually be called "metaphysical").

Barfield already knew a century ago that there is no clear line between *poetry* and *prose*: in reality these are undefinable categories, strictly speaking. But there is a clear distinction between poetic language and the analytic language that scientists must use.

The poet relishes ambiguity [24], which is fundamental in language and essential to poetry. But the point of analytical language is to reduce the inherent ambiguities as far as possible.

To be explicit here (since we will systematically contrast *poetic* and *analytic* language), poets have a free hand to use words any way they choose to invoke meaning to the hearers, making as full use as they like of the range of connotation (the *ambiguity*) of the words used. If the poet is successful, then the hearer perceives meaning in the poem. On the other hand, scientists must *analyse* the ideas they wish to develop into components that are specified and combined as unambiguously as possible. But where do the scientists' ideas come from in the first place?

It should not be thought that because the use of 'natural' language is inescapable (and therefore that fundamental ambiguities necessarily remain in our theories), our knowledge of the world is thereby undermined. We will here underline what is common sense: all knowledge is ultimately incomplete—that is, we cannot know *everything* about anything. We wish to underpin our knowledge by giving a more correct account of it. No knowledge is absolute, and it is time to give a more nuanced account of the basis of our epistemology. Ultimately, we cannot avoid ambiguity: therefore, let us—like poets—start to treat it positively.

The analytical narrative must be encased in a metanarrative (as we will show explicitly in §4); moreover, poetic perception cannot be spoken of analytically. The early Wittgenstein famously said, "*Whereof one cannot speak, thereof one must be silent*" [25], but the later Wittgenstein changed his mind, saying instead, "[*in most cases*] . . . *the meaning of a word is its use*" [26]. In our terms, he switched from believing that analytic language was sufficient, to recognising that poetic language was ontologically indispensable [27]. Something similar can be said of Richard Rorty: in 1982 he famously said (citing William James) that truth is "*a compliment paid to sentences that seem to be paying their way*" (Rorty, 1982 [79]); but in 2000 he says: "*it was a mistake on my part to go from criticism of attempts to define truth as an accurate representation of the intrinsic nature of reality to a denial that true statements get things right*" (Rorty, 2000 [80]). In his influential essay Bruno Latour also said, "*do we now have to reveal the real objective and incontrovertible facts hidden behind the* illusion *of prejudices?*" (emphasis original, Latour 2004 [81]). Of course, we will argue here (§3) that it is a *logical* mistake to try to "*define truth*".

Our epigraph touches both ontic and epistemic issues. It is composed (after a poem in *Náhuatl*, an autochthonous Mexican language) in *Palikur*, a northern Arawak language spoken by less than four thousand people living in the Brazilian state of Amapá and in French Guiana. There is a Palikur-Portuguese dictionary (Green, 2010 [82]) and the language displays *ways of knowing* that differ markedly from modern European ones (Green, 2013 [83]), and in particular Palikur speakers are deeply aware of what we would regard as advanced *topological* concepts from the very grammar of their language (Green and Green, 2023 [84]) [28]. The way we think—our very identity—is inextricable from our *language* (and the *Náhuatl* and the *Palikur* poems both express how horrible its loss would be [29]). What we *know* is inexpressible without language [30]. Benjamin Lee Whorf (1941 [40]) drew attention to the converse of this: " . . . *people act about situations in ways which are like the ways they talk about them*", but this only serves to underline our point. If we cannot say it, we cannot know it: this is true for all aspects of reality.

## 3. The Thinginess of Things

Michael Frayn (2006 [86]) has memorably spoken of the "*thinginess of things*" [31], that is, the sure ontological grasp that reality appears to have on us. Things *are*! This has long been resonant with the poets: for example, Wallace Stevens (1954 [88]) spoke specifically of "*A new knowledge of reality*"; we could also mention William Carlos Williams' famous and very influential line "*No ideas but in things*" (Williams, 1926 [25]), which is still widely discussed (see for example Finch, 2013 [89]). Additionally, Iris Murdoch is quoted as saying, "*I'm glad we live in a thingy world*" (Jordan, 2012 [90]); her novels are shot through with this philosophical attitude. In her first published novel (Murdoch, 1954 [91]) she makes one

of her heroes observe that the "*activity of translating*", central to our thesis here, is "*an act so complex and extraordinary that it was puzzling to see how any human being could perform it*". Why is this? Because every thing is "*astonishing, delightful, complicated and mysterious*" (*ibid*. ch. 4, p. 62).

*Thing* is a very ancient word with a surprisingly wide range of connotation (including parliament), and which is thought to be related to the Indo-European root of the Latin word *tempus*, time. Of course, material things only exist—can only exist—in time: Frank Wilczek (2021 [92]; ch.6, p. 159) points out that this underlies Augustine of Hippo's (c.420 CE, [93]) elegant proof that the Christian doctrine of Creation entailed the creation of *time* along with matter [32]. For, Augustine said, we only know time by the movement of things (he fixed their ontology by calling them "creatures"—that is, things made by the Creator); therefore, if there are no things then neither can there be time:

> procul dubio non est mundus factus in tempore, sed cum tempore . . . nullum autem posset esse praeteritum, quia nulla erat creatura, cuius mutabilibus motibus ageretur
>
> verily the world was made with time, and not in time . . . no time passed before the world, because no creature was made by whose course it might pass. But it was made with time, if motion be time's condition
>
> <div align="right">Augustine, City of God XI:6, c.420 CE</div>

There is also a similar statement in a lengthy and acute discussion in Book XI of the Confessions (Augustine c.400 [94]). Thus, Augustine anticipates by a millennium and a half the conclusion of the *Gravitational Singularity Theorem*: that *time* had a beginning is a necessary consequence of General Relativity (Hawking and Penrose, 1970 [95]).

All physicists operate on the assumption (not usually explicitly acknowledged) that the thinginess of the phænomena they investigate is ontologically secure: that is, the world is real [33]. Philosophically and historically, this ontological security ultimately derives from the assertion of Creation by the monotheist religions [34], even if most physicists today assume it tacitly merely as a pragmatic precondition. Interestingly, Gerry Schroeder (1997 [99]) has shown *both* that the Hebrew Creation story successfully resists scientific criticism, *and* that its interpretation is as subtle and elusive as any poetic text. And Iris Murdoch is not the only philosopher to comment on, as she puts it, "*the infinite elusive character of reality*" (Murdoch 1962 [100]).

It is important to realise that the thinginess of things is *ontologically axiomatic*, as Frayn effectively acknowledges in a long discussion (Frayn 2006 [86]). Our ultimate epistemological reliance on *personal guarantee* is documented by Richard Bauckham (2006 [101]) in the context specifically of historical events: ultimately, we know things only through *eyewitness testimony* [35]:

> The testimony of Holocaust survivors is the modern context in which we most readily recognise that authentic testimony from participants is completely indispensable to acquiring real understanding of historical events, at least events of such exceptionality.
>
> <div align="right">Bauckham, 2006 §18 (p. 499)</div>

We can of course subject *testimony* to the standard critical tests but, more often than not, in the end we have to decide whether or not to trust the witness. In the end, we simply have to *choose* what to believe. Note that "*personal guarantee*" also underlies the peer review system, which cannot operate without good faith. Thus, *testimony* also underlies the epistemology of scientific knowledge.

It seems that all scientists ought to be effectively realists of some sort, whether or not they believe this philosophically (but see the "*Solipsist's Plea*" below, *note*#84). If they did not implicitly believe (a) that the world is *there*, (b) that *laws of nature* exist, and (c) these laws are *discoverable* by us; why would they get up in the morning for another frustrating day in the lab? Surely, they would find something else more lucrative to do? But realists

do not have to be naïve! So the fifth chapter of the dense book by Karen Barad (Barad, 2007 [24]) is titled "*Getting Real: Technoscientific Practices and the Materialisation of Reality*" and has an epigraph by Michel Foucault (renowned as a postmodern structuralist critic even if he himself did not like these labels). Barad's book is an extended, detailed and subtle investigation of "*Reality*" and the "*Ontology of Knowing*" (these terms are taken from the book's chapter headings) in the light of the ontological puzzles forced on us by a deep look at the fundamentals of quantum mechanics. Barad knows not only that the Universe is *there*, but also that our usual naïve ways of thinking about this are false—our grasp of reality is often uncertain and unreliable: the book title ("*Meeting the Universe Halfway*") is a line from Alice Fulton's poem "*Cascade Experiment*" (Fulton, 1989 [102]) which opens: "*Because faith creates its verification* . . . "[36].

Oliver Sacks (1993 [33]) confirms this attitude independently, speaking of Humphrey Davy (the great English chemist) as a poet: "*The poet and the chemist were fellow warriors, analyzers and explorers of a principle of connectedness*" (ibid, p. 23), and he quotes Coleridge (1818 [107]): "*through the meditative observation of a* Davy . . . *we find poetry, as it were, substantiated and realized in nature* . . . *as at once the poetry and the poet*" (emphasis original).

However, reality is elusive. Is knowledge objective? Are the things that science describes and explains really there? Alessandro Fedrizzi and Massimiliano Proietti (Fedrizzi and Proietti 2019 [108]) gloss their paper (Proietti et al., 2019 [109]) as "*Objective Reality Doesn't Exist, Quantum Experiment Shows*". The paper reports an elegant three-photon-pair implementation of a "Wigner's friend experiment" demonstrating a violation of the associated Bell inequality [37]. This means that in this case the results observed are not "objective" (that is, they are not observer-independent). But, as Karen Barad explains in detail, this does not mean that reality itself is illusory, only that knowing it is not necessarily very straightforward:

> Traditional philosophy has accustomed us to regard language as something secondary, and reality as something primary. [Niels] Bohr considered this attitude toward the relation between language and reality inappropriate. When one said to him that it cannot be language that is fundamental, but that it must be reality that, so to speak, lies beneath language, and of which language is a picture, he would reply, "We are suspended in language in such a way that we cannot say what is up and what is down. The word 'reality' is also a word, a word we must learn to use correctly."
>
> Barad 2007 [24], p. 205 (quoting Petersen 1985 [111])

Michael Polanyi in his "*Personal Knowledge*" (1958 [38]) insists that, ultimately, we have only *personal* guarantees of whatever knowledge we think we possess: strictly speaking, *objective knowledge* is an oxymoron [38]:

> . . . the intuition of rationality in nature [must] be acknowledged as a justifiable and indeed essential part of scientific theory. That is why scientific theory . . . [can be] represented as a mere economical description of facts . . . or as a working hypothesis . . . [but these are] interpretations that all deliberately overlook the rational core of science.
>
> . . . great theories are rarely simple in the ordinary sense of the term. Both quantum mechanics and relativity are very difficult to understand; it takes only a few minutes to memorize the facts accounted for by relativity, but years of study may not suffice to master the theory and see these facts in its context.
>
> . . . We understand the meaning of the term 'simple' only by recalling the meaning of the terms 'rational' or 'reasonable' or 'such that we ought to assent to it', which the term 'simple' was supposed to replace. The term 'simplicity' functions then merely as a disguise for another meaning than its own. It is used for smuggling an essential quality into our appreciation of a scientific theory which a mistaken conception of *objectivity* [39] forbids us to acknowledge.

Polanyi, *Personal Knowledge* 1958 [38], §1:4 (emphasis added)

where here by "rational" Polanyi means to imply *our* application of reasoning: it is *people* who do the reasoning! Knowledge is irreducibly personal; the "*rational core of science*" entails reasoning people [40]. So we prefer the Copernican theory over the Ptolomaic one precisely because we think that "*its excellence is, not a matter of personal taste on our part, but an* inherent *quality deserving universal acceptance by rational creatures. We abandon the cruder anthropocentrism of our sense—but only in favour of a more ambitious anthropocentrism of our reason*" (*ibid*, §1:1). Similarly, Polanyi and Prosch (1975 [22], Ch.3) draw attention to "*Noam Chomsky's critique of B. F. Skinner's 'Verbal Behavior' [which] presents many illustrations of such behaviourist paraphrasing. Apparently objective terms . . . are so used that their ambiguity covers the mental terms they are supposed to replace.*"

Jerome Ravetz (1971 [113]) later took up and amplified the social element of Polanyi's characterisation of knowledge as personal in his very influential demonstration that scientific research is a *craft* activity heavily dependent on the tacit knowledge that Polanyi emphasised.

How do we know that nature is rational (and is therefore amenable to scientific description)? We intuit it. Prior to our rationalisations is our belief that rationalisations exist. And in speaking of *rationality* here, Polanyi is also referring to the primacy over common sense scientists commonly give to idealistic thought—we have already mentioned Maxwell's "*aim-oriented empiricism*" (Maxwell, 2020 [63]). Polanyi asks:

> What is the true lesson of the Copernican revolution? Copernicus gave preference to man's delight in abstract theory, at the price of rejecting the evidence of our senses, which present us with the irresistible fact of the sun, the moon, and the stars rising daily in the east to travel across the sky to their setting in the west.
>
> Polanyi, 1958 [38], §1:1

The fact may appear psychologically "*irresistible*"; nevertheless, Polanyi points out that behaving rationally we systematically do resist it. We may "intuit" that the sun goes round the earth; but at a deeper level we intuit that the relation of sun to earth is lawful, and analytically we recognise that the simplest expression of the law has the earth going round the sun. We intuit the existence of the rationality that underpins this lawfulness. It is the business of poets to articulate intuition [41].

Of course, Polanyi is aware of the logical necessity of this attitude to rationality, which becomes clear (as he explains) when Kurt Gödel's *Incompleteness Theorem* (1931 [43]) is understood. Quoting S.C.Kleene's *Introduction to Metamathematics* (1952 [114]), Polanyi says,

> Rules have been stated to formalise the object theory, but now we must understand without rules how those rules work. An intuitive mathematics is necessary even to define the formal mathematics.
>
> Polanyi, 1958 [38], §8:8

This "intuitive mathematics" is called "metamathematics" by everyone—Polanyi, Kleene, Gödel—just as we call the comparable "intuitive physics" by the cognate word "metaphysics". Every narrative has its metanarrative, without which no one can make any sense of it.

Gödel's achievement was to demonstrate by construction that his formula (which we can express in words as "*this sentence is undecidable*") was <u>not</u> meaningless. His demonstration was rather involved, but indicates the processes of mind required to establish this cornerstone of epistemology. We display its flavour with this brief extract from his Introduction (here *R* is an ordering relation for all the definable formulas, and *K* is the set of "Gödel numbers" *q* representing *unprovable* formulas):

> Die Analogie dieses Schlusses mit der Antinomie Richard springt in die Augen; auch mit dem „Lügner" besteht eine nahe Verwandtschaft, denn der unentscheidbare Satz [R(q); q] besagt ja, daß q zu K gehört, [das heißt] nach (1), daß [R(q);

q] nicht beweisbar ist. Wir haben also einen Satz vor uns, der seine eigene Unbeweisbarkeit behauptet.

[13] Man beachte, daß „[R(q); q]" ... bloß eine metamathematische Beschreibung des unentscheidbaren Satzes ist.

The analogy between this result and Richard's antinomy leaps to the eye; there is also a close relationship with the "Liar", since the undecidable proposition [$R(q)$; $q$] states precisely that $q$ belongs to $K$, that is according to Equation (1), [$R(q)$; $q$] is not provable. We are therefore confronted with a proposition that asserts its own unprovability.

(footnote #13:) Note that "[$R(q)$; $q$]" ... is merely a metamathematical description of the undecidable proposition.

Gödel, 1931 [43]

Richard's paradox was stated in 1905, but the Liar Paradox is ascribed to Epidemides of Crete, alluded to by Paul of Tarsus (Titus 1:12, 57 CE [42]), and investigated at length among others by the 14th century John Buridan, who conditioned Galileo's theory of *impetus* (Read, 2002 [115]).

It is well known that Gödel later became fascinated by Anselm's comparable *Ontological Argument* for the existence of God (*Proslogion*, Anselm 1078 CE [116]). Anselm asserted that the idea, "*aliquid quo maius nihil cogitare potest*" ("*that than which no greater can be thought*") was <u>not</u> unthinkable, and therefore God (than which no greater can be thought) must exist in fact. Starting from this premise of "*thinkability*", Anselm actually gave a proof that in its self-referencing form [43] anticipated Gödel's proof by a millennium:

Et certe id quo maius cogitare nequit, non potest esse in solo intellectu. Si enim vel in solo intellectu est, potest cogitare esse et in re, est in solo intellectu: id ipsum quo maius cogitare non potest, est quo maius cogitare potest. Sed certe hoc esse non potest.

And surely *that-than-which-a-greater-cannot-be-thought* cannot exist in the mind alone. For if it exists solely in the mind even, it can be thought to exist in reality also, which is greater. If then *that-than-which-a-greater-cannot-be-thought* exists in the mind alone, this *that-than-which-a-greater-**cannot**-be-thought* is *that-than-which-a-greater-**can**-be-thought*. But this is obviously impossible.

Anselm, 1078 [116], II

The elegance of Anselm's Latin is noticeable. And one can hear an attenuated echo of this ontological argument in Descartes' famous dictum "*cogito ergo sum*" ("*I think therefore I am*" [44]), which George Berkeley (1710 [120]) modified to "*esse est percipi*" ("*to be is to be perceived*") deliberately to contrast the idealism of the scholastic nominalists with the new materialist schools. Anselm goes on to comment on the relation between believing (ontics) and understanding (epistemics) that is central to our present work:

Gratias tibi, bene dominum, gratias tibi, quia quod prius credidi te donante, iam sic intelligo te illuminante, ut si te esse nolim credere, non possim non intelligere.

I give thanks, good Lord, I give thanks to you, since what I believed before through your free gift I now so understand through your illumination, that [even] if I did not want to *believe* that you existed, I should nevertheless be unable not to *understand* it.

Anselm, 1078 [116], IV

This is reminiscent of Augustine's dictum "*nisi credideris non intelligetis*" ("if you do not believe you will not understand": *City of God*, XII:17 [93]; quoting a version of Isaiah 7:9– Anselm himself quotes Augustine explicitly). But Anselm has recognised how the increase in knowledge works—first we *see*, then we *understand*—which is equally true for painters, for poets, and for physicists. First one grasps the *idea*, then one works out the details. Just

because the devil is in the detail does not mean that the initial illumination is dispensable. Just because many ideas turn out to be incoherent does not mean that the fruitful ideas do not originate with illumination. One is reminded of Eric Dodds' comment (1951 [121], in his Preface): "*time and the critics can be trusted to deal with the guesses; the illumination remains*".

We are not here saying that we reliably grasp things by intuition—everyone knows this is not the case! To test the reliability of our ideas we have to do science in the usual way. But where does the idea itself come from? Its origin is the "*illumination*" discussed by Anselm [45]. We *discern* truth: nevertheless, uncertainty cannot be eliminated.

Both Gödel's and Anselm's sentences are self-referencing, and have logical properties entirely due to this recursiveness. Gödel's sentence is proved "*not meaningless*" by construction (and therefore true, by a metamathematical argument), but because of its wider scope Anselm's sentence has resisted such construction [46].

Gödel's proof was a revolution, not only in its overturning of the expectation of the mathematicians that arithmetic could be proved *both* consistent *and* complete [47], but also in its entirely novel style of proof, relying explicitly on a *meta*mathematical argument. It is interesting not only that Anselm anticipated Gödel, but also that he understood the logical status of his argument, which he did not present analytically but poetically (as a prayer). Ultimately, ontic knowledge is, and can only be, intuited. How else can one understand Paul of Tarsus writing in 57 CE (Robinson, 1976 [17]) about God, who:

καλουντος τα μη οντα ως οντα

(Romans 4:17, *Textus Receptus* [67])

calleth thoſe things which be not as though they were

(*transl:* Tyndale, 1526 [69])

In a different context, Thomas Piketty (2019 [34]) gives us a complementary view of the necessarily intuitive nature of the knowledge of thinginess. In a section titled "*On the Complementarity of Natural Language and Mathematical Language*", Piketty says:

> This book will rely primarily on natural language (about which there is nothing particularly natural) . . . There is no substitute for natural language when it comes to expressing social identities or defining political ideologies. . . . Those who believe that we will one day be able to rely on a mathematical formula, algorithm, or econometric model to determine the "socially optimal" level of inequality are destined to be disappointed. . . . I do not contend that "truth" is found only in numbers or certainty only in "facts".
>
> Piketty, 2019 [34], *Introduction*

To be clear: we are distinguishing between the *analytic* language required for scientific work, and the *natural* language we use every day (see *note#7*) together with the *poetic* language needed to express deep meanings; there is no sharp boundary between "poetic" and "natural" language just as there is no sharp boundary between poetry and prose.

Piketty encloses "facts" in quotes since these are always contentious in economics: one person's verity is always another's heresy, and Piketty authoritatively displays the *ideological* nature of such "facts". But it turns out that physics is also ideological in a similar way and for similar reasons [48]. Of course, this is not entirely unexpected: our present essay here could be thought of as merely a footnote to Thomas Kuhn's seminal book of a generation ago (*The Structure of Scientific Revolutions*, 1962 [127]). We proceed to explore this ideology specifically in relation to the development of ideas of entropy since the mid-19th century.

## 4. Entropy and Information

As a specific phænomenological example of the scientific method in action, leading to new knowledge, we will now tell the strange and intricate story of the development of the

idea of "*entropy*", which word is a neologism of Rudolf Clausius in analogy to "*energy*" (an exactly similar Hellenic word). This story concerns the foundations of physics and is still being vigorously developed, which is why we give it so much space here: modern work has underlined its fundamental importance practically, scientifically and philosophically. We hope to give a flavour of this importance to the wider public, despite the very substantial difficulty (both mathematical, and philosophical) of the subject.

We start with considering precisely how the First Law of Thermodynamics ("*energy is conserved*") was originally conceived (Clausius 1854 [35]). It is an effort of imagination for us today to appreciate that at that time the energy-equivalence of *work* and *heat* was not understood, and therefore that Clausius' formulation was a breakthrough. Here we give the long passage explicitly (over 300 words in a complex German) because we want to underline just how complex even "simple" ideas really are. Advances in physics involve revolutions in thought, that is, looking at things *differently*. And the job of poets is precisely to help us to see things differently. Clausius' physics advance was therefore also a break-through in poetics, just as the new appreciation of atomic theory stimulated by the new microscopes in the 17th century was also a breakthrough in poetics as Cassandra Gorman has shown explicitly (Gorman 2021 [128]).

The early work established the idea (§4.1; this story is told in much more detail by Paul Sen, 2021 [45]); then Boltzmann and others developed its implementation in statistical mechanics (§4.2); then we consider information, and the "Shannon entropy" (§4.3); the case of black hole entropy and the Bekenstein–Hawking equation is amazing (§4.4); then we apparently come full circle considering the "geometrical" entropy of Parker and co-workers, who clearly demonstrate a true *isomorphism* between entropy and energy (not merely an "analogy"; §4.5); lastly we consider some implications of this discussion for the very meaning of "causality" (§4.6).

*Causality* is a basic notion in science—we cannot do science at all unless we think that *causes* have *effects*. The trouble is that the closer one looks at things the more entangled everything gets. The modern work on entropy has highlighted this sort of problem since it turns out that *non-local* phaenomena are widespread—these sorts of phaenomena defy cause-and-effect sorts of accounts, being governed instead by the Variational Principles (Least Action, Maximum Entropy etc.). It seems that the very *idea* of "causality" is some sort of approximation that must be tempered by a parallel treatment of the interconnectedness of things.

*4.1. Early Work on the Concept of Entropy*

*Entropy* is a slippery concept. It is the thing you measure (or calculate) if you want to know the effect of the Second Law of Thermodynamics, which says (roughly) that entropy can only increase. It is the Second Law that asserts that you cannot unscramble eggs, or alternatively (there are many ways of saying the same sort of thing) we live first and die afterwards, and not the other way round. These things are obvious, but it is not at all obvious how to introduce these "simple" ideas into physics, and there have been detailed (and frankly arcane) technical arguments for the last couple of centuries on what these things mean. We therefore open with some modern positions to help with orientation.

Edwin Jaynes was responsible for the seminal variational Principle of "*Maximum Entropy*", now widely used across many fields and, in a paper explaining some fundamental aspects of the treatments of (the 19th century giants of physics) Gibbs and Boltzmann, Jaynes says about entropy:

> It is interesting that, although this field [entropy] has long been regarded as one of the most puzzling and controversial parts of physics, the difficulties have not been mathematical. ... It is the enormous *conceptual* difficulty of this field which has retarded progress for so long.

> Jaynes, 1965 [129] (emphasis original)

The Oxford English Dictionary (OED [53]) is very helpful. Rudolf Clausius introduced the term *entropy* in 1865 specifically as a Hellenistic neologism: from ἐν + τροπή (transformation; literally 'turning': all the connotations of *trope* are also present in English). The OED comments:

> Clausius assumed that (German) *Energie* literally meant 'work content' (*Werkinhalt*) and devised the term *Entropie* as a corresponding designation for the 'transformation content' (*Verwandlungsinhalt*) of a system.
>
> Oxford English Dictionary, 3rd Edition (September 2018)

And then, in sense 1a ("*Physics and Chemistry*"), the OED elaborates:

> Entropy was first defined by the German physicist Rudolf Clausius (1822–1888). Scottish physicists Peter Guthrie Tait (1831–1901) and James Clerk Maxwell (1831–79) were the first to interpret entropy as *a measure of the unavailability of energy for work*.

> The modern mathematical definition of entropy, in terms of the possible microstates ... of a thermodynamic system, first appears in the work of Austrian physicist Ludwig Boltzmann (1844-1906), who viewed entropy as *a measure of the disorder of a system*.

> [Sense 3 "*Statistics and Information Theory*")] ... mathematician Claude Shannon (1916–2001) coined the term in the context of information theory (see sense 3b)
>
> Oxford English Dictionary, 3rd Edition (September 2018)

The OED gives a variety of definitions, three related to scientific concepts. (We will show below that these do not exhaust the meanings assigned to the term.) This is not merely a philological variety, but a real scientific discrepancy that has led to much confusion. It is still not entirely clear to everyone that the multiple definitions do actually refer consistently to a coherent idea. But the confusion has certainly resulted in error. Indeed, as Jaynes noted near the end of his life, regarding his variational approach to providing an underlying principle to entropy: " ... *the long confusion about order and disorder (which still clutters up our textbooks) is replaced by a remarkable simplicity and generality*" (Jaynes 1992 [130]).

The very logical status of the Second Law of Thermodynamics has long been debated, as hinted at above. Is it a fundamental Law? Or is it a consequence of the other Laws, which are all time-reversible (except for the CP-violation by K-mesons discovered by Cronin and Fitch: Christenson et al. 1964 [131]) [49]? Either way, consistency is a problem. How can time reversibility be consistent with time irreversibility (see below on the "Arrow of Time", §4.6)? Clausius first clearly stated a version of the Second Law in 1854:

> es kann nie Wärme aus einem kälteren in einen wärmeren Körper übergehen, wenn nicht gleichzeitig eine andere damit zusammenhängende Aenderung eintritt.

> Heat can never pass from a colder to a warmer body without some other change, connected therewith, occurring at the same time.
>
> Clausius, 1854 [35]

In the same 1854 paper, Clausius also recognised (before he had introduced the term) that entropy remains unchanged for reversible cyclic processes ("*umkehrbaren Kreisprocesse*"), calling the identity $\int dQ/T = 0$ the "second law of the mechanical theory of heat" ("*des zweiten Hauptsatzes der mechanischen Wärmetheorie*"). Of course, the "first law" was $Q = U + A \cdot W$, where $Q$ is the total quantity of heat ("*die ganze Wärmemenge*"), $U$ is how much heat is in the system before work is done on it, $W$ is the external work ("*die äußere Arbeit*"), and $A$ is the factor converting work to heat ("*das Wärmeaequivalent für die Einheit der Arbeit*", literally: "the heat equivalent for the unit of work"). It is instructive to see how Clausius reasons here:

Bei dieser Bestimmungsweise kann man den Satz von der Aequivalenz von Wärme und Arbeit, welcher nur einen speciellen Fall der allgemeinen Beziehung zwischen lebendiger Kraft und mechanischer Arbeit bildet, kurz so aussprechen:

*Es läfst sich Arbeit in Wärme und umgekehrt Wärme in Arbeit verwandeln, wobei stets die Gröfse der einen der der anderen proportional ist.*

... Betrachten wir nun die bei einer Zustandsänderung gethane innere und äufsere Arbeit zusammen, so können sich beide, wenn sie von entgegengesetzten Vorzeichen sind, theilweise gegenseitig aufheben, und dem Reste mufs dann die gleichzeitig eintretende Aenderung der Wärmequantität aequivalent seyn. Für die Rechnung aber kommt es auf dasselbe hinaus, wenn man für jede von beiden einzeln eine aequivalente Wärmeänderung annimmt.

Sey daher *Q* die ganze Wärmemenge, welche man einem Körper, während er auf einem bestimmten Wege aus einem Zustande in einen andern übergeht, mittheilen müfs, (wobei eine entzogene Wärmemenge als mitgetheilte negative Wärmemenge gerechnet wird), so zerlegen wir diese in drei Theile, von denen der erste die Vermehrung der wirklich in dem Körper vorhandenen Wärme, der zweite die zu innerer und der dritte die zu äufserer Arbeit verbrauchte Wärme begreift.

Von dem ersten Theile gilt dasselbe, was schon vom zweiten gesagt ist, dafs er von der Art, wie die Ver-änderung stattgefunden hat, unabhängig ist, und wir können daher beide Theile zusammen durch eine Function *U* darstellen, von der wir, auch wenn wir sie sonst noch nicht näher kennen, wenigstens soviel im Voraus wissen, dafs sie durch den Anfangs- und Endzustand des Körpers vollkommen bestimmt ist.

Der dritte Theil dagegen, das Aequivalent der äufseren Arbeit, kann, wie diese selbst, erst dann bestimmt werden, wenn der ganze Weg der Veränderungen gegeben ist. Nennen wir die äufsere Arbeit *W*, und das Wärmeaequivalent für die Einheit der Arbeit *A*, so ist der Werth des dritten Theiles $A \cdot W$, und wir erhalten daher als Ausdruck des ersten Hauptsatzes folgende Gleichung:

(I) $Q = U + A \cdot W$

With this means of determination, one can now concisely express the relation between the equivalence of heat and work (which is only a special case of the general relationship between active power and mechanical work) by the following saying:

*Work can be turned into heat and* vice versa *heat can be turned into work, so that the magnitude of the one is always proportional to the other.*

... Let us now consider, in the event of a change of state, the internal and external work together. These both, taken together, can partially compensate each other if they are of opposite signs. Then the remainder must be equivalent to the change of the quantity of heat that occurs at the same time [i.e., during the change of state event]. For the calculation however, it comes back to the same thing, if one assumes an equivalent change in heat from the two separate entities [i.e., for each of internal work and external work separately, one takes the heat equivalent].

Let *Q* be the entire quantity of heat that must be imparted to a body, while going on a certain path from one state to another (where heat removed is counted as a negative quantity of heat imparted) [this is in the context of the Carnot cycle]. This can be broken into three parts, of which the first is the increase in heat actually present in the body, the second is the heat used for internal work and the third the heat used for external work.

Of the first part one can say the same as has already been said about the second part: that it is independent of the way that the change of state happened. We can

therefore combine both parts together into a function *U*, for which we know in advance (regardless of how little knowledge we otherwise have) that it is completely (sufficiently) defined by the initial and final states of the body.

On the other hand, the third part, i.e., the equivalent of the external work, can only be calculated when the whole path of change is given. We call the external work *W*, and the heat equivalent for the unit of work *A*, so that the value of this third part is the product *A·W*, and we come into view of the resulting first law in the following equation: (I) *Q = U + A·W*

<div align="right">Clausius, 1854 [35] (emphasis original)</div>

It is plain that the equation, *Q = U + A·W*, derives its meaning from the previous discussion, which is in a "*verschachtelt*" (literally "*nested*") German that is both syntactically and semantically complex: it defies a literal translation and it is hard to translate into a comprehensible English. This extract concerns what we now know (and what Clausius himself called) the "First Law" (of thermodynamics). Clausius later in the paper tries to describe the effect of *entropy*, involving the "Second Law" (both his term and ours), without knowing its explicit existence or name (he only coined the term in his 1865 paper [132]), hence the apparent confusion and inarticulacy of this complex text of 1854. We leave the linguistic analysis as an exercise for the interested reader, but we conclude that Clausius is carefully constructing ("*negotiating*", Edwardes, 2019 [26]) meanings for the terms he wishes to manipulate mathematically in just the way that Barfield says is characteristic of poets.

This is a rather clear example of metaphysical priority in a physical argument. We will discuss the logical properties of metanarratives later (§5.2): here we see Clausius using a natural language replete with its natural metaphors and ambiguities, but intending to restrict the unavoidable ambiguity as much as possible. It is only by using natural language that we can say anything at all, but then if we care about the meanings we are constructing we have to also address the formal poetics. Usually this step is tacit, but we are here drawing attention to it.

Physicists tend to think that they can manipulate the behaviour of *phænomena* symbolically (since we all believe that the symbols truly represent reality), but in fact they only symbolically manipulate the *ideas* they have constructed of those behaviours [50]. Whence arise the ideas? And what relation (both ontic and epistemic) has the idea to the phænomenon?

### 4.2. Entropy and Statistical Mechanics

All students of thermodynamics start today with the model of the ideal Carnot cycle, which establishes the ideas of "waste heat" and "maximum thermodynamic efficiency". Clausius depended on the Carnot cycle to model his idea of "entropy" as the accessible useful work available in some quantity of heat—in his time, the steam engine powered the world: is it any wonder that (as we shall see) the ideal gas laws should be the natural exemplar of heat engines [51]?

It is also by considering the ideal gas as a model for heat engines that today's students learn the basics of statistical mechanics, first developed with great brilliance by the mid-nineteenth century giants of physics: Gibbs, Boltzmann and Maxwell. Ludwig Boltzmann is remembered by his eponymous constant *k*, and by the formula engraved on his tombstone (which in this form is due to Max Planck) [52]:

$$S = k \log W \tag{1}$$

It is well known that this "simple" treatment ignores or obscures a number of severe difficulties. The usual definition makes entropy an *extensive* quantity even though it is well known that this is an approximation appropriate only in certain circumstances:

Entropy is just as much, and just as little, extensive in classical statistics as in quantum statistics ... entropy stands strongly contrasted to energy.

Jaynes, 1992 [130]

Strictly speaking, entropy is an *intensive* quantity [53], as Jaynes observes in a penetrating discussion in the same place of the so-called *Gibbs Paradox*:

> [Gibbs] had perceived that, when two systems interact, *only the entropy of the whole is meaningful*. Today we would say that the interaction induces correlations in their states which makes the entropy of the whole less than the sum of entropies of the parts; and it is the entropy of the whole that contains full thermodynamic information. This reminds us of Gibbs' famous remark, made in a supposedly (but perhaps not really) different context: "*The whole is simpler than the sum of its parts*." How could Gibbs have perceived this long before the days of quantum theory?

Jaynes, 1992 [130] (emphases original)

Jaynes earlier had made an astonishing statement of the *subjectivity* of the concept of entropy in his acute comparison of the Gibbs and Boltzmann formulations:

> . . . not only in the well-known statistical sense that it measures the extent of human ignorance as to the microstate [but also] *[e]ven at the purely phenomenological level, entropy is an anthropomorphic concept.* For it is a property, not of the physical system, but of the particular experiments you or I choose to perform on it.

Jaynes, 1965 [129] (emphasis original)

The point here is that the result of the entropy calculation depends on how the Partition Function [54] of the system is specified, that is, which particular measurements are being contemplated. The Partition Function describes how the phase space (which enumerates all of the *microstates*) is specified. Then the observables are specified by the *macroscopic parameters*, which can hopefully be calculated from the thermodynamics. Roger Penrose puts this quite sharply:

> . . . we can . . . appreciate . . . [that] Boltzmann's formula . . . put forward in 1875 . . . represented an enormous advance on what had gone before . . . There are, nevertheless, still certain aspects of vagueness in this definition, associated, primarily, with the notion of what is to be meant by a "macroscopic parameter".

Penrose, 2010 [142], §1.4

Carlo Rovelli made essentially the same point very recently when he argues that "*we are blind to many variables [that are] at the heart of Boltzmann's theory*", adding:

> Thermodynamics . . . is a description of these variables of the system: those through which we assume we are able to interact with the system

Rovelli, 2017 [143] (ch.10, n.4; *emphasis original*)

However, it was Max Planck who in 1900 [144] first recognised "Boltzmann's constant" *per se* (see Equation (1)) as fundamental to entropy in the seminal paper in which he explains black body radiation in terms of quantised resonators; and where he gives the quantisation constant, $h$, in units of action correct to almost 1% [55].

### 4.3. Information

We go into apparently arcane details in this section following Lars Lundheim's useful review (Lundheim 2002 [145]), not only because the details are both surprising and very interesting, but also because it is the assimilation of Claude Shannon's *information entropy* (or the eponymous *Shannon entropy*) that has enabled the proliferation of today's high-speed networks, a technology that would otherwise be inexplicable.

The first transatlantic "telegraph" cable was laid in 1858 but only operated for three weeks. A lasting transatlantic connection was established in 1866. In addition to its technical triumph, this was commercially very valuable (and expensive) technology, and the search for efficiency naturally attracted great scientific attention. The first message was

transmitted (by Morse code, in 1858) at 10 min per word. The second (1866) cable already operated almost two orders of magnitude faster, at eight words per minute; but the transmission speed (that is, the *bandwidth*) was necessarily slow because of frequency dispersion in the cable: this was already understood in principle by William Thompson (later Lord Kelvin) who published his analysis in 1854 and was closely involved with the enterprise.

However, although practical development (telegraphy with time- and frequency-division multiplexing, telephony, radio) was very rapid, little advance was made on what we would now call *informatics* until the 1920s, when it became clear that "*bandwidth limitation sets a* fundamental *limit to the possible information transfer rate of a system*" [145]. And the very idea of *bandwidth* depends on the understanding of electrical '*band pass*' *filters*, which were not patented until 1917.

The additional problem of *signal-to-noise* dominated telecommunications science as soon as more reliable long-distance signalling was allowed by usable amplifiers (i.e., valves, exploiting the vacuum tube technology which had originally been developed for the incandescent light bulb). But in the 1920s there was still no standard scientific understanding of noise: Norbert Wiener's work on stochastic noise (Brownian motion) was published between 1920 and 1924, and Harry Nyquist's mathematical model of thermal noise was only published in 1928. The vacuum tube amplifier had been introduced around 1910, but the high gains obtainable by cascading amplifiers had to wait until the feedback principle was patented in 1928. And then noise became important to control, being a limiting factor to transmission systems: "*by the 1930s 'signal-to-noise ratio' had become a common term among communications engineers*" [145].

It is this century of prior telecommunications history that set the scene for Claude Shannon's breakthrough paper of 1948 in which he re-used the term *entropy* to give a measure of "what rate information is produced" in a communication channel. In this work he showed quantitatively how the maximum bit-rate depended both on the noise in the channel and on its bandwidth, and he also established that completely error-free information exchange was possible, as long as the data rate in the channel was below a certain value (the "channel capacity").

> When one compares the generality and power of explanation of Shannon's [146] paper "*A Mathematical Theory of Communication*" to alternative theories at the time, one can hardly disagree with J.R.Pierce [147] who states that it "came as a bomb".
>
> Lundheim, 2002 [145]

Shannon used the term *entropy* as referring to "quantities of the form $H = -\sum p_i \log p_i$" which "play *a central role in information theory as measures of information, choice and uncertainty*" specifically because it had the same form as that "*defined in certain formulations of statistical mechanics*" (citing Richard C. Tolman's magisterial *Principles of Statistical Mechanics*, 1936 [148]), and it is now known as the "information entropy", or the "Shannon entropy". Shannon used the symbol $H$ to invoke "*the* H *in Boltzmann's famous* H *theorem*" (possibly "$H$" originally denoted the Greek letter *eta*—H,h).

Responding to Shannon, Leon Brillouin considered "information" in 1953 [149] as negative entropy: *negentropy*; and Edwin Jaynes' seminal work of 1957 amplified Shannon's observations on probability distributions saying, "*the development of information theory has been* felt *by many people to be of great significance for statistical mechanics, although the exact way in which it should be applied has remained obscure*"; but then adding:

> In this connection it is essential to note the following. The mere fact that the same mathematical expression $-\sum p_i \log p_i$ occurs both in statistical mechanics and in information theory does not in itself establish any connection between these fields. This can be done only by finding new viewpoints from which thermodynamic entropy and information-theory entropy appear as the same *concept*.
>
> Jaynes, 1957 [150] (emphasis original)

Jaynes went on to establish the congruence of the ideas of thermodynamic and information-theoretic entropies, demonstrating that using a probability distribution that *maximizes the entropy* (subject to certain constraints) justifies making inferences from that distribution. Following Jaynes, the powerful "Maximum Entropy" ("MaxEnt") methods are now very widely used across a large variety of technical disciplines.

Rolf Landauer famously drew specific attention to the entropy cost of computation, originally in 1961 (Landauer 1987 [151]), insisting that *computation is physical*. Although many of the steps in a computation can be carried out reversibly, information *erasure* is necessarily irreversible, and carries an inescapable entropy cost, as was emphasised by Charles Bennett:

> Landauer's principle, while perhaps obvious in retrospect, makes it clear that information processing and acquisition have no intrinsic, irreducible thermodynamic cost whereas the seemingly humble act of information destruction does have a cost, exactly sufficient to save the Second Law from [Maxwell's] Demon.
>
> Bennett, 2003 [152]

Today, as Parker and Jeynes (2019 [28]) have pointed out, citing significant recent work in network theory (Parker and Walker, 2014 [153]): the entropic treatment of information is standard in the analysis of the efficiency of communications networks in the presence of noise; also, applying Landauer's Principle [56] to a computation involves the transfer of information and therefore also results in a rise in entropy (Parker and Walker, 2007 [156]) [57].

### 4.4. The Entropy of Black Holes

The celebrated Bekenstein–Hawking equation for the entropy of black holes, $S_{BH}$, is due to seminal work by Jacob Bekenstein (1973 [158]) showing that $S_{BH}$ is proportional to its surface area (that is, the area $A$ of its event horizon). Stephen Hawking (1976 [159]) gave an argument for the value of the constant of proportionality, giving $S_{BH} = \frac{1}{4}Akc^3/(G\hbar)$, where as usual $k$, $\hbar$, $c$ and $G$ are, respectively, Boltzmann's constant, the reduced Planck constant, the speed of light and the gravitational constant. Again in this formula, as for Planck's treatment of the black body radiation, it is the ratio $h/k$ that is significant: Planck recognised that this was directly fixed by the Wien displacement constant $b$, and that $hc/kb$ was dimensionless (see *note*#55).

Bekenstein explicitly uses the Shannon information entropy in his derivation, specifically in the sense of the "*inaccessibility of information about* [the black hole's] *internal configuration*", thereby also implicitly employing Brillouin's concept of "negentropy":

> [here] we attempt a unification of black-hole physics with thermodynamics. In Sec. II we point out a number of analogies between black-hole physics and thermodynamics, all of which bring out the parallelism between black-hole area and entropy. In Sec. III, after a short review of elements of the theory of information, we discuss some features of black-hole physics from the point of view of information theory. We take the area of a black hole as a measure of its entropy—entropy in the sense of inaccessibility of information about its internal configuration [58].
>
> Bekenstein, 1973 [158]

Stephen Hawking's discovery of his eponymous radiation (Hawking, 1974 [162]) confirmed Bekenstein's 1973 suggestion that black holes have a "temperature"; as indeed does any object having a finite entropy. Hawking demonstrated that the black hole behaves as though its event horizon is a (typically very cold) black body with a temperature inversely proportional to the black hole mass (for the central supermassive black hole of the Milky Way this works out as 15 fK). But at the event horizon of a black hole there is no matter that is not infalling: clearly, the idea of "temperature" is here used in a very different sense from normal temperatures, which always refer to a statistical (macroscopic) property of some sort of particle ensemble. But there is no ensemble at the event horizon!

Parker and Jeynes (2019 [28]) showed how the Bekenstein–Hawking expression for the black hole entropy can be used to determine the virial mass of the (heavily idealised) Milky Way galaxy from the known mass of the supermassive black hole at the galactic centre. The galactic virial mass (which is dominated by the mass of the presumed "dark matter") is the galactic mass that can be inferred by the motion of its stars. Their derivation of the virial mass was a simple application of their recasting of the *maximum entropy* condition into an entropic Lagrangian/Hamiltonian formulation of equilibrium thermodynamics (the so-called *Quantitative Geometrical Thermodynamics*, QGT), in which the *double-helix* and the *double logarithmic spiral* are proved to be holomorphic [59] geometries corresponding to maximum entropy entities.

The double logarithmic spiral is a good zeroth order model for (idealised) spiral galaxies and QGT offers an explanation for the MaxEnt *stability* of a spiral galaxy without needing "dark matter", but of course galaxies are necessarily structures that are far from equilibrium [60], and the calculation of galactic virial mass has a number of as yet unresolved associated problems [61]. However, recently Parker and Jeynes (2021 [140]) have shown in the framework of QGT how the Bekenstein–Hawking expression itself is a consequence of Liouville's Theorem [62], expressed in entropic terms [63].

Black holes are extremely simple objects which are specified by only four parameters: mass, charge, angular momentum and the "Planck length" (Frank Wilczek omits the <u>scale</u> of "*elementary particles*" when he characterises them as those having only mass, charge and spin: Wilczek 2021 [92], ch.3, p. 73). It is because black holes are so simply specified that they are so definitely known to be ontologically simple, indeed, unitary (than which exists nothing simpler): their property of being maximum entropy (MaxEnt) objects is also related to this simplicity. Parker et al. (2022 [161]) have shown, using a QGT formalism, that alpha particles are also unitary (ontologically simple).

However, even though black holes (like alpha particles) are very simple MaxEnt objects, nevertheless (unlike alpha particles), they are not in thermodynamic equilibrium. They necessarily accrete mass. As yet, although it has been extended by Parker and Jeynes (2021 [160]) to idealised spiral galaxies to yield an expression for the *entropy production* (a Noether-conserved quantity), the QGT formalism has not been systematically extended to express the evolution of MaxEnt objects in time. But it is already clear that such an extension would be natural to the formalism since an expression for "entropic force" is available (Parker and Jeynes 2019 [28], Equation (30); see also Keppens 2018 [167], Equation (30)).

*4.5. Geometric Entropy: Holography and Entanglement*

The holographic properties of black holes have long been recognised, together with the non-local consequences. So Raphael Bousso said, in a review originating in developments in quantum gravity:

> The holographic principle . . . implies that the number of fundamental degrees of freedom is related to the area of surfaces in spacetime. Typically, this number is drastically smaller than the field theory estimate. Thus the holographic principle calls into question not only the fundamental status of field theory but the very notion of locality. . . . Quantum gravity has imprinted few traces on physics below the Planck energy. Among them, the information content of spacetime may well be the most profound.
>
> Bousso, 2002 [168]

What is striking about the treatment of Parker and Jeynes (2019 [28]) is the *non-local* properties of the entropy, so that the spiral galaxies have their shape (on this account) as a consequence of the holomorphism of the double logarithmic spiral. They say:

> we have shown that the [double logarithmic spiral] structure of the . . . Milky Way . . . is consistent with a holomorphic representation in geometric algebra. In particular, we have shown that the [calculated] galactic shape, aspect ratio, and

structural stability (which are all highly constrained by the algebra) are consistent with observation; and we have also shown that the total galactic [virial] mass is also consistent with observation. Note that this is a simplified ("zeroth order") analytical approximation to reality: . . . the dynamics driving the galactic evolution [are neglected . . . but] this treatment gives the proper weight to the effect of the [central supermassive] black hole entropy

<div align="right">Parker and Jeynes, 2019 [28]</div>

Thus, the galactic shape is a primary *geometric* property, even if it can also be shown in standard treatments to emerge from the kinematics. Parker and Jeynes (2020 [157]) also prove that the stability of Buckminsterfullerene ($C_{60}$) is a *geometrical entropy* property fundamentally related to its representation as a holomorphic object. They say that the stability of $C_{60}$ is:

[a property] of the thermodynamics of the system: [which is] a significant methodological advance since a detailed treatment of the energetics may be avoidable. . . . The spherical $C_{60}$ fullerene molecule therefore represents a least exertion or Maximum Entropy (most likely) topology . . . For $C_{60}$ the double-spiral trajectories have been proved holomorphic and maximum entropy in an exact Euler-Lagrange analytical treatment (given the approximation to a true spherical geometry).

<div align="right">Parker and Jeynes, 2020 [167]</div>

Parker and Jeynes (2021 [160]) also demonstrate directly that the holographic principle itself is a consequence of the *entropic* Liouville Theorem:

The geometric entropy of both the sphere and the double-helix are clearly holographic in nature, since they are proportional to the surface areas of enclosed volumes. . . .

. . . consideration of the geometric entropy of systems ranging . . . from the molecular . . . through to [cosmic] scales yields a common holographic interpretation . . . The holographic principle itself . . . is a consequence of the holomorphism . . . of the objects considered.

The close relationship between quantum mechanics . . . and statistical mechanics . . . is well known ... However, using geometric entropy and the entropic version of Liouville's Theorem . . . we have shown not only how the entropy of a MaxEnt system is holographic in nature, but also that there exists an associated entropic version of the uncertainty principle, based on the Boltzmann constant as the appropriate entropic counterpart to the Planck constant.

<div align="right">Parker and Jeynes, 2021 [160]</div>

Further work has shown that the holographic principle is also effective at sub-atomic scales: Parker et al. (2022 [161]) express the nuclear sizes of the helium isotopes ($^4$He, $^6$He, $^8$He) and the self-conjugate $A = 4n$ nuclei ($^4$He, $^8$Be, $^{12}$C, $^{16}$O, $^{20}$Ne, $^{24}$Mg, $^{28}$Si, $^{32}$S, $^{36}$Ar, $^{40}$Ca) in terms of a single parameter, the "*holographic wavelength*" associated with the entropic geometry: all of these calculated values being entirely consistent with measurement.

In our present context, the point about holography is precisely that *each part represents the* whole, that is, it carries the implication of non-locality. It is of course well known that "individual" electrons in an atom, or "individual" nucleons in a nucleus are strictly indistinguishable in a proper quantum treatment: this implies that in a holographic system all the "individual entities" are actually somehow mutually entangled [64].

Entanglement at the microscopic scale is currently well understood. But the galactic scale also appears to us to have some properties which seem similar. It is clear that our idealised spiral galaxy, expressed as a (holomorphic) double-logarithmic spiral, is treated by the QGT formalism as an object whose entropy is given holographically, just like the

entropy of its central supermassive black hole. But then, should the galaxy not also be considered as *entangled*, just as are quantum objects like atoms and atomic nuclei? After all, *entanglement* represents another way to speak of *non-local influence*, and what could be more non-local than the symmetry of well-formed spiral galaxies, which are common in the Universe? [65]

*4.6. The Arrow of Time, and Teleology*

Time asymmetry is a problem because all the laws of physics we know are apparently time-symmetrical, apart from the Second Law of Thermodynamics (and the CP properties of the K-meson—see *note*#48). Whence then the Second Law? Is it independent of the other laws? In any case, how can it be consistent with the other laws considering that it is *not* time-symmetrical but almost all the other laws we know of *are*? This is known as the *Loschmidt Paradox* (see for example Lucia 2016 [171]) which has been addressed directly in the formalism of QGT (Velazquez et al. 2022 [72]).

One approach to this adopted recently by widely disparate authors is to deny that the arrow of time is real: that is, time does not have a beginning. Carlo Rovelli (2017 [143]) claims that the reality is that the arrow of time is a matter of perspective ("*Time is Ignorance*"), justifying this by a discussion of Boltzmann's statistical mechanics apparatus (a discussion amplified in detail with considerable subtlety by John Earman, 2006 [172]). Roger Penrose claims to have found a way of extending Time back beyond the Big Bang singularity with his detailed suggestion of *Conformal Cyclic Cosmology* (Penrose 2010 [142]). Ilya Prigogine claims that *Time Precedes Existence* (Prigogine 1996 [173]). All of these eminent scientists recognise that they here venture into "metaphysics" [66], but we dissent from their conclusions essentially on physical grounds.

Robert Bishop discusses the problem of the arrow of time in the nonequilibrium statistical mechanics of Prigogine's "Brussels–Austin Group" (Prigogine 1977 [175]): he considers "*the observed direction of time to be a basic physical phenomenon due to the dynamics of physical systems*" and continues:

> One claimed virtue [of this approach] . . . is the ability . . . to provide time-asymmetry. . . . Why then do we not observe [entropy decreasing]? To answer this question . . . [and by] translating their conception of entropy into information-theoretic language [they] showed that their formulation of the second law requires infinite information for specifying the initial states of a singular distribution evolving in the negative [time] direction, but only finite information for specifying the initial states for evolution in the positive [time] direction.

> This would render the initial conditions for systems to approach equilibrium along the negative *t*-axis physically unrealizable . . . Since singular probability distributions are supposedly operationally unrealizable, they argue it is physically impossible for unstable systems to evolve to equilibrium in the negative [time] direction. Hence, their version of the second law acts as a selection rule for initial states.

> This argument is supposed to show why anti-thermodynamic behavior in the real world is impossible . . . Nevertheless, the argument is problematic. The most fundamental difficulty is that it conflates epistemic concepts (e.g., information, empirical accessibility of states) with ontic concepts (e.g., actual states and behaviors of systems).

> Bishop, 2004 [176]

Here again we see *entropy* (the subject of the Second Law) intricately tied up with *information*, a relation we have already explored above. We also have an explicit statement of how even the best minds can experience "fundamental" epistemological and ontological difficulties in this whole subject.

In this context, we wish to point out the teleology apparently implicit in the *Principle of Least Action*. Photons apparently "decide" which path to take on the basis of this Principle. That is, they can be represented as doing a variational calculation over all possible paths, and choosing the *least action* path. Of course, we know that such anthropomorphising language cannot be used properly of photons, but what precisely is it that constrains them to take the paths they do? They behave as though they had a purpose, and the consequence of the Second Law is that the universe behaves as though its purpose is to maximise entropy. But we exorcised teleology from science when we abandoned Aristotle in the 17th century (and a very good thing too!).

It turns out that there is an entropic counterpart to the *Principle of Least Action*: the *Principle of Least Exertion*. Parker and Jeynes (2020) explain:

> [Parker and Jeynes, 2019] have shown that the *principle of least action* has the entropic analogue of a *principle of least exertion*: where "action" is the path integral of the kinematic Lagrangian, "exertion" is the path integral of the entropic Lagrangian—which still satisfies the various canonical conjugate-pairing relationships. Roughly speaking, in the energy domain where the Hamiltonian represents the total energy of a system (that is, the sum of potential and kinetic terms), the Lagrangian represents an energy balance (the difference of potential and kinetic terms). The entropic Hamiltonian-Lagrangian treatment emerges from a consideration of information as the orthogonal complement to entropy
>
> Parker and Jeynes, 2020 [157]

(although the two Principles are mathematically *isomorphic* and not merely "analogues").

It seems that a proper consideration of *entropy* (implying the arrow of time) is intimately linked up on the one hand with the physical quantity *Exertion* and the variational *Principle of Least Exertion*, and on the other hand with holographic properties of objects which can be at any scale, from sub-atomic to cosmic (entropy being essentially scaleless, as is witnessed by the logarithm in Equation (1)). And these holographic properties are essentially non-local, giving those wedded to mechanical cause-and-effect [67] modes of thought the impression of teleology.

Michael Stöltzner has investigated the teleological aspects of the *Principle of Least Action* (the PLA) [178], showing that the logical empiricists (such as Moritz Schlick, Hans Hahn and Philipp Frank) ignored the PLA on account of these apparently teleological aspects even though Max Planck and David Hilbert emphasised it, and Jennifer Coopersmith has recently underlined its fundamental nature in an elegantly deep and wide-ranging treatment (Coopersmith 2017 [179]). Planck considered "*the PLA as formal embodiment of his convergent realist methodology*", and Hilbert "*took the PLA as the key concept in his axiomatizations of physical theories*"; serving "*one of the main goals of the axiomatic method*", that is, "*deepening the foundations*." Stöltzner points out that for Planck and Hilbert and their schools, the PLA did not have the theological connotations ascribed to it by Maupertuis (for example). He says:

> Both its staunchest advocates and those remaining silent about the PLA shared the conviction that final causation, material or organismic teleology, and analogies with human behavior had to be kept out of physics.
>
> Stöltzner, 2003 [178]

Just so! Aristotelian teleology was simply a baleful error that proved far too influential. We could however note here that Stöltzner cautions: "When it comes to philosophy, the German word Zweckmäßigkeit is notoriously difficult to translate. *Teleology*, *finality*, and *purposiveness* capture only part of it". The question of what precisely is the intended meaning of the words we use obtrudes persistently, even in a technical or scientific context. Stöltzner continues:

> Moreover, none of the protagonists of the debate under investigation considered the PLA as an instance of backward causation. The history of physical teleol-

ogy might alternatively suggest a relationship between the PLA and the problem of determinism. . . . neither PLA-advocates nor logical empiricists contemplated any relation between the PLA and the second law of thermodynamics [except Boltzmann]. Rather, they explicitly restricted the validity of the PLA to reversible phenomena regardless of their views on causality.

<div align="right">Stöltzner, 2003 [178]</div>

It seems to us that we need to revisit this debate since the heroes of physics at the beginning of the 20th century knew nothing of *exertion* and the *Principle of Least Exertion* (**PLE**) that Parker discovered (Parker and Jeynes 2019 [28]), and which is demonstrated both complementary to the PLA and also emerging from the QGT formalism. He has shown that this QGT formalism is general, that is, it is also valid for non-equilibrium (irreversible) systems, like (idealised) spiral galaxies whose entropy production has been derived analytically from QGT (Parker and Jeynes 2021 [160]).

However, the new (QGT) treatment of info-entropy is entirely consistent with standard ideas of causality: its treatment of information presupposes this [68]. We suspect that apparent causality paradoxes observed in the past associated with the PLA should instead be viewed as *entanglement* effects of the non-locality. This may have very wide-ranging ramifications, including putting David Bohm's "pilot wave" proposal (Bohm 1952 [181] [69]) in a new light, as Parker et al. comment:

> It is worth pointing out that Bohm's recognition of a "quantum-mechanical" potential U($x$) exerting a "quantum-mechanical" force "*analogous to, but not identical with*" the conventional strong force on a nucleon ([[Bohm 1952] his Equation (8)), can now be understood to be a prescient anticipation of our *entropic force*, familiar from our previous discussion of galactic geometry ([Parker and Jeynes 2019, their Equation (23)).

<div align="right">Parker, Jeynes and Catford (2022) [161]</div>

Alastair Rae has observed: "If, as a result of the modern work on irreversible processes, we were to be led to a fundamental physics that took as its central theme the idea that time really does flow in one direction, I at least would certainly welcome it" (Rae 1986 [183]). Parker's info-entropy formalism presupposes the arrow of time, since it treats the Second Law of Thermodynamics as axiomatic [70]. And since the fundamental nature of the variational Principles is uncontroversial (and since the info-entropy formalism naturally generates the PLE as the entropic isomorph of the PLA), it seems that Rae's desire is satisfied.

## 5. Knowledge of Meaning

We are arguing in this essay that "knowledge" has to *mean* something, and that this meaning must be *grasped*. Understanding is not a mechanical process: on the contrary, we commonly experience understanding "dawning" on us. Understanding is by illumination. This process cannot be adequately spoken of in purely analytical language.

In this section, we explore these things. We first make some observations about the properties of *Definitions* (§5.1); then some remarks on the properties of *Metanarratives* (§5.2); then we discuss the properties of *rationality* (§5.3); and finally we point out some consequences for poetics, using as an example some canonical poetry (§5.4).

Both knowledge and *meaning* are very ancient words in English [71], with roots in ideas that have always exercised humans as far back in time as we can tell. We have already shown that our modern knowledge of physics is rooted in our humanity: but we wish to underline that *knowledge is personal*, and always has been. We have a tendency to be dazzled by the huge advances in both mathematics and philosophy by the ancient Greeks, but in truth human interest in (and knowledge of) these things long predated the Greeks.

*5.1. The Problem of Definition*

Things are what they are: ultimately, they are ineffable (except to poets [72]): things-in-themselves are hard to speak of, and they *cannot* be defined. We can only define the ideas we have of things, not the things themselves. But to speak coherently about things we *must* define the ideas we have of them. We cannot speak of any thing without having some more or less clear idea of what it is. It should be obvious that although the ontology of the thing (its thinginess) and its epistemology (how we know it) are intrinsically separate ideas, yet in any specific case the two must be inextricably bound together. We cannot know anything about the thinginess of the thing without also knowing how we know. This is true despite the fact that this knowledge is almost invariably implicit (or "tacit").

The problem then is the propensity we have of confusing our idea of the thing with the thing itself. We think that because we have a satisfactory idea of the thing, we know the thing in itself. If I ask, *What is entropy?* [73] you may answer, with early Clausius: *It is a measure of how much work is available in a quantity of heat*; or with later Clausius: *It is a closed line integral of the change in heat of a body at the absolute temperature of the body at the time of the change*; or with Boltzmann (as later interpreted by Planck): $S = k \ln W$; or with Shannon $\{S = k \sum p_i \ln p_i \}$; or with Parker and Jeynes: *the maximum entropy of a holomorphic body is a holographic property of its geometry*.

All of these answers are correct in their own terms, but an observer could be forgiven for thinking that they do not all describe the same thing: the "thermodynamics" used by Parker and co-workers might be almost unrecognisable by Clausius and Boltzmann. Is it the same? Is Parker's "entropy" the same as Clausius' "entropy"? The mathematical apparatus of both have recognisable similarities, but does this establish identity? We have already quoted Edwin Jaynes (1957 [150]) on this: "*The mere fact that the same mathematical expression occurs . . . does not in itself establish any connection*." But Jaynes went on to show that in fact statistical mechanics (Boltzmann's achievement) and information theory (Shannon's achievement) really are both truly thermodynamics. And Parker's entropy is too, since his achievement is firmly built on Jaynes'. This conclusion is clearly a real semantic development in word usage, as well as being a startling development of the mathematical apparatus.

The very word *thing* itself was originally used of immaterial things, as we have seen. In fact, the first group of meanings listed in the Oxford English Dictionary are entirely of immaterial things ("*A meeting, or the matter or business considered by it, and derived senses*"): only the second group of meanings (§§8-17: "*An entity of any kind . . . in the most general sense, in fact or in idea*") concerns material things, and then only in a secondary way. It is only in sense §11 that the word is used to denote explicitly *material objects*. It is clear that in standard English usage a property of a thing is also itself a (different) thing. The curious fact appears to be that *things* are no less thingy for not being concrete. In which case one can hardly be surprised if things turn out to be hard to tie down. Indeed, in 1991 Landauer wrote a popular paper "*Information is Physical*" on the thermodynamics of information erasure (Landauer 1991 [186]) which precisely emphasised the *thinginess* of a quantity that most people assumed was too abstract to be a thing.

Recently, rather similar and highly relevant observations have emerged in a different context. Mari et al. (2013 [187]) have drawn a careful philosophical distinction between *being a quantity*, and *being measurable*. They point out that this distinction is an *ontological* one, and moreover, that "*measurement is primarily an epistemic process*". Underlying this treatment is the recognition that "*knowledge is constructed by humans*": that is, as we have already insisted above, ultimately knowledge is *personal*. This position is explored in more detail by Maul et al. (2016 [188]) who deprecate "*the appearance of rigor and objectivity* [achieved] *by reducing abstract ideas to observables*". Knowledge, being constructed by humans, is necessarily and intrinsically ideological: these authors we cite are metrologists who include a member of the JCGM [74], a committee of the BIPM.

The recognition that *knowledge is personal* is widespread: we have noted the positions of Michael Polanyi and the metrologists above and we will consider *gnosis* in more detail

below, but it is worth underlining the generality of this observation by mentioning the comment of Oliver Sacks (1985 [191]) on a case of clinical agnosia he describes in detail:

> . . . our mental processes, which constitute our being and our life, are not just abstract and mechanical, but personal, as well—and, as such, involve not just classifying and categorising, but continual judging and feeling also.

> Oliver Sacks, *The Man who mistook his Wife for a Hat* (1985)[191]

Sacks considers that *knowledge* is not merely an abstract philosophical idealism, it is actually what allows us to *live*.

It is important to emphasise that the desire for "objective" (that is, *impersonal*) knowledge is a form of chasing the rainbow. When we cast our ideas in mathematical terms *we do* not *remove* ambiguity! The mathematics beloved of physicists is their way of thinking clearly (and unambiguously) about seriously complex ideas. Ideally, mathematical methods do not *introduce* ambiguity, but neither can they *remove* the fundamental ambiguities implicit in the basic ideas. Reality itself is elusive! [75]

### 5.2. Metaphysics and Metanarratives

We have already noted that we will use "*metaphysics*" neither in its standard philosophical usage, nor with its usual (normally pejorative) ordinary meaning, but as a term cognate with "metanarrative", "metadata", "metamathematics" (see *note*#15). That is, "*metaphysics*" will here denote "*the metanarrative of physics*".

In the proof of the Incompleteness Theorem Gödel himself explicitly makes parallel use of two strands of argument, the mathematical and the *meta*mathematical. He says (Gödel 1931 [43]):

> Der im System PM unentscheidbare Satz wurde also durch metamathematische Überlegungen doch entschieden.

> So the proposition which is undecidable in the PM system yet turns out to be decidable by metamathematical considerations

> Gödel, 1931 [43]

where "PM" here refers to Russell and Whitehead's axiomatisation of arithmetic in *Principia Mathematica* (Russell and Whitehead 1910, 2nd ed. 1925 [125]). Note that Whitehead himself said of this work (Whitehead 1929 [14], p. 8), "*even in mathematics the statement of the ultimate logical principles is beset with difficulties as yet insuperable*". Whitehead goes on to comment acerbically that "*peccant premises*" in incorrect philosophical arguments are notoriously hard to locate (*ibid*.).

Aristotle's book τὰ Φυσικά (*The Physics*) has a title perhaps most helpfully translated *Natural Philosophy*. Similarly, the title of his τὰ μετὰ τὰ Φυσικά (*The Metaphysics*) might be *Beyond Nature.* The one deals (largely) with material things, the other mainly with the immaterial. There is a widespread prejudice today that the immaterial has little or no real existence. The way Gödel proved his Theorem, by the formal use of a *meta*mathematical argument, demonstrates that such an assumption is without foundation. It seems that strict materialism is irrational [76]. Be that as it may, it seems logically inescapable that every narrative necessarily has its metanarrative without which it can make no sense. And this is true also in pure physics, as we have seen by showing how the metaphysics is implicit in the history of the idea of *entropy*.

The obvious question then arises, what is the meta-metanarrative of the metanarrative [77]? and the meta-metametanarrative of that (and so on)? We have seen that Clausius uses a 'natural' *language* for what we have called his metaphysics, even if his German is of a rather special sort: this type of 19th century academic German is known as "*verschachtelt*" (that is, "complex" or "nested"). But the metalanguage of this *verschachtelten Deutsches* is still only German: a natural language is its own metalanguage. Anselm already recognised this recursiveness of language, and Gödel only formalised it mathematically.

On metaphysics and poetry, in particular the "metaphysical poets" of 17th century England, we should also mention the intense interest in the *philosophical* idea of the "atom" that turns out to be highly relevant to our story. When the 17th century philosopher-poets used "atom" (of course meaning the ultimate *indivisible* particle)[78] they were invoking wonder at the new marvels that flowed from their enthusistic use of the newly-invented microscope[79]. As Cassandra Gorman says: "*Minute and unfathomable in and of itself, the concept of the atom takes on immense possibilities within poetic spaces that recognised the value of its indivisibility. The enormous potential of the atom—the indivisible at the core of matter; the building block of all things—captured the seventeenth-century poetic imagination and created new opportunities to 'pierce deep' (Bacon) into ontological subjects*" (Gorman 2021 [128], p. 36). This is particularly relevant to us here since the new thermodynamics we have discussed in §4 has underlined the fundamental importance of *unity* (Henry More's neologism, "*monad*", was famously picked up later by Leibniz: "The *monad* ... is nothing but a simple substance ... *simple*, that is to say, without parts"—Leibniz 1714 [163]), and the theological importance of the *atom* as intrinsically *unitary* was well-recognised in the 17th century.

*5.3. Ambiguity and Coherence*

We have seen Clausius carefully constructing ("*negotiating*": Edwardes 2019 [26]) unambiguous meanings for the terms he wishes to manipulate mathematically—using linguistic means. Strictly speaking, this is specifying the physics by means of a *metaphysical* discourse. Philosophers of science have tended to obscure this step as much as they can, but it is ***explicit*** even in the proof of the Incompleteness Theorem, as we have seen. Even to do fundamental mathematics we are forced to recruit the help of metamathematical methods: is it then surprising that at the fundamentals of physics also lurk metaphysical methods?

But rational speech is not limited to analytical speech. Poetic speech derives any power it may have from its *internal* coherence[80]: and coherence is a property of rationality. The epistemology of physics rests on the foundation of socially verified personal testimony, which is a form of poetic speech. Form and the knowledge of form are both prior to all scientific knowledge. Prior to all rationalisation is the knowledge of the possibility of rationalisations. Rationality itself is a poetic, not an analytic property.

Three millennia ago, the Psalmist underlined the rationality of the (necessarily intuited[81]) knowledge of God, *insisting* (in Hebrew) that the epitome of rationality—that is, the regularity of the heavens—was specifically a pointer to the knowledge of God, saying (and we give the ancient Greek translation of the Hebrew)[82]:

οι ουρανοι διηγουνται δοξαν θεου ... ο νομος του κυριου αμωμος επιστρεφων ψυχας ... τα δικαιωματα κυριου ευθεια ευφραινοντα καρδιαν

Ps.18:2,8, 9 (Lxx, c.250 BCE [196])

The heauens declare the glorie of God ... the law of the Lord is perfite [perfect], conuerting the ſoule ... the ſtatutes of the Lord are right, and reioice the heart

Ps.19:1,7,8 (Geneva, 1560 [184])

Some four centuries after the Psalmist, the prophet Jeremiah took up the same idea, insisting that the *rationality* of God was an earnest of the *dependability* of God:

ιδου ημεραι ερχονται φησιν κυριος και διαθησομαι τω οικω ισραηλ και τω οικω ιουδα διαθηκην καινην ... ουτως ειπεν κυριος ο δους τον ηλιον εις φως της ημερας σεληνην και αστερας εις φως της νυκτος ... εαν παυσωνται οι νομοι ουτοι απο προσωπου μου φησιν κυριος και το γενος ισραηλ παυσεται γενεσθαι εθνος κατα προσωπον μου πασας τας ημερας

Jeremiah 38:31,36,37(Lxx, c.3rd century BCE [196])

Beholde, the daies come, ſaith yᵉ Lᴏʀᴅ, that I wil mak a newe couenant with the houſe of Iſraél, and with the houſe of Iudáh ... Thus ſaith the Lᴏʀᴅ, which giueth the ſunne for a light to the day, and the courſes of yᵉ moone and of the ſtarres for

a light to the night ... If thefe ordinances departe out of my fight, faith the Lord, then fhall the feed of Ifraél ceafe from being a nation before me, for euer.

Jeremiah 31:31,35,36 (Geneva, 1560 [184])

Whence the laws of physics on whose nature all physicists depend? Today the tendency would be to say something equivalent to: never mind the ontic antics, shut up and calculate! [83] But it seems that much that we are interested in is non-calculable, that is, it "*inherently is non-algorithmic and, therefore, cannot be surrogated and simulated in a Turing machine*" (Rubin and Crucifix, 2021 [197]).

Until quite recently, the standard answer to the ontological question "*whence natural law?*" would have been *to* point to Jeremiah's καινη διαθηκη (*new testament*; c.f. Luke 22:20) which underlies European civilisation in the last two millennia. And Jeremiah asserts that this "new testament" is a covenant guaranteed by the testimony of the very heavens: "*if ever these laws* (νομοι) *depart from before my face* (προσωπου) . . . "

These ancient poets were *poets*, not scientists: even Jeremiah predated the peak of Hellenic science with *Thales* being his younger contemporary. The later Alexandrian scholars responsible for the Greek text we display could not help interpreting the Hebrew, but even their Greek is a complex text with multiple ambiguities; ambiguity which is clearly intended by the poet, and which is enhanced by the *coherence* of the text.

How then does the little *cohere* with the large? [84] the early *cohere* with the late? [85] We have said that thermodynamics (that is, entropy) is scale-less: and this must apply to both of the conjugate *quantities*, time and energy: [86] energy is clearly the province of the physicists, but we regard time as the province of poets (as well as historians), and we wish to peer back into the mists of time provided we may do so securely (that is, where verifiably reliable records exist).

### 5.4. Meaning in Poetry

| | |
|---|---|
| Lo giorno se n'andava, e l'aere bruno | The day was dying, and the darkening air |
| toglieva li animai che sono in terra | Brought all the working world of living things |
| da le fatiche loro; e io sol uno | To rest. I, only, sweated to prepare |
| | For war, the way ahead, the grind that brings |
| m'apparecchiava a sostener la guerra | The battler to hot tears for each yard gained: |
| si' del cammino e si de la pietate | To bitter tears, and memories more real |
| , che ritrarra la mente che non erra. | Than what was real and which is thus retained |
| | Unblunted, edged with even sharper steel. |
| | My Muse, my schooled and proven gift, help |
| O muse, o alto ingegno, or m'aiutate | me: |
| ; o mente che scrivesti cio ch'io vidi, | It's now or never. Fortify my mind |
| qui si parra la tua nobilitate. | With the vivifying skills of poetry, |
| | For what I saw needs art of a great kind. |
| | I saw great things. Give them nobility. |

Dante Alighieri, *Divina Commedia*, 1320 (*transl.* Clive James, 2013); Canto II

We quote Dante's masterpiece because, at the start of Canto II, the poet is thinking about how to say what he wants, and how hard it is. Also because the form of the work is untranslatable, as is most *poetry* (and the idea of "*translation*" is essential to this thesis). We have chosen Clive James' translation (James 2013 [198]) because he asserts that Dante's *terza rima* simply does not work in English: instead, he uses quatrains, sometimes expanded, as here. And also because Dante deliberately makes use of a variety of poetic means to convey his meaning. James says:

> Dante was one of the most educated men of his time even in the conventional sense, quite apart from the proto-scientific sense in which he was original without parallel. But [Byron and other translators] couldn't, or wouldn't, get down to the level where syllables met each other and generated force. That had to be the aim, impossible as it seemed; to generate the force, both semantic and phonetic: the force of both meaning and sound. Indeed, in the original, some of the meaning

was *in* the sound. Unless the translator did something to duplicate how the poem sounded, he, or she, wouldn't get near what it meant.

James, 2013 [198] (emphasis original)

The comment that James is pointing to the *thinginess* of Dante's epic is irresistible. Both poetry and ordinary language deliberately use multiple layers of meaning to express the *thing in view*. Ambiguity is built-in to poetic expression at a fundamental epistemological level: there is no unambiguous knowledge of a *thing in itself*. Scientists wish to speak unambiguously about the *thing* presently in view. But this is impossible in principle! What to do?

We have considered *the* example of the evolution of the idea of *entropy*, showing that at each stage the *thing in view* is replaced by an *idea* of the thing delineated in a natural language which aligns its salient features (that is, the properties of the thing then considered salient) with mathematical (that is, well-defined) ideas. This is a well-known progression that is usually presented as a version of solipsism [87], but that this cannot be the reality is demonstrated by the (almost) uniform belief of physicists that they are really describing the world as it is [88].

We have shown instead that although physics represents real knowledge about the real world, this knowledge is inaccessible in principle without the use of natural language, with all *the* ambiguity that entails. Ultimately, physics relies ostensibly on this "natural" language, but used (as natural language often is) with poetic overtones: that is, with the deliberate intention of creating new meanings, as we have shown Clausius doing (§4.1).

So Quentin Meillassoux (2006 [201]) asserts that the laws of nature are epistemologically *contingent*, not necessary (thereby escaping a number of otherwise intractable *philosophical* problems, although he sidesteps the problem of ontology altogether). So much is obvious from our previous discussion, but Meillassoux's "demonstration" is entirely in an analytic language which in English sounds deeply obscure. Is this a problem of translation? I think not. In the terms of the present discussion, Meillassoux has failed to acknowledge the necessity of using poetic language to express the ontological verities he sees, and the translator therefore has no warrant to write good English.

Unlike Meillassoux, Karen Barad asserts that the laws of nature are also *ontologically* contingent: "*phenomena are real material beings*" (Barad 2007 [24], p. 361). Like Meillassoux, much of Barad's discussion is analytically intricate, however Barad explicitly anchors the discussion in the poetic, with the book's title being a line quoted from Alice *Fulton's* poem (Fulton 1989 [102]). Barad opens with a discussion of Michael Frayn's play "*Copenhagen*" (Frayn 1989 [202]) specifically to introduce a close discussion of Niels Bohr's "philosophy-physics". The question in all this is, who was right? was it Werner Heisenberg saying that the Uncertainty Principle expressed merely our *epistemological ignorance* or was it Bohr's view that it expressed a fundamental *ontological indeterminacy* of reality? [89] Barad's assessment is clear that Bohr was right.

Meaning is elusive. Even in physics, meaning is elusive. *What is an electron?* The simple answer is easy: *we* do *not know*! [90] Which does not mean that we know nothing at all, nor that what we think we know is illusory, nor that truth itself is a mirage. No. But we cannot do without poetic language to *really* say what we mean.

## 6. Image and Knowledge: Seeing and Knowing

We know what we see, we see what we know [91]. Reality is elusive and its epistemology is not *easy* to tie down. What we see is conditioned by what we know. And of course, what we know is informed by what we see. In any case, what is certain is that our knowledge is not only incomplete but also uncertain: we are fallible!

Therefore, we wish to widen our view into historical considerations. Physics is concerned with the present behaviour of the systems we are interested in, but if we take the *complementarity* of energy and time seriously (which physically speaking we must) then we need also to look into the past (the future being inaccessible to us). In particular, human

knowledge is a property of humans, and humans develop in time and in societies that develop in time. We will now enquire into the most ancient knowledge available to us that we have reliable evidence on.

Paul of Tarsus said in 55 CE [67] [92]: αρτι γινωσκω εκ μερους (*now I know in part*: I Corinthians 13:12), but because we know things only partially does not mean we do not know any thing. Our survey of the idea of *entropy* has highlighted how partial our knowledge remains: even such a basic idea of physics remains controversial. However, even though the way we think of entropy has changed dramatically over the last century and a half, we can still obtain real and useful results. The fundamentals shifting beneath our feet is uncomfortable, but physicists are familiar with this feeling from the quantum revolution a century ago.

The idea of *knowledge* is ancient and has very deep roots for us, both ontic and epistemic. The English word "*knowledge*" is etymologically related to Paul's γινωσκω, as the OED notes (even the Latin "*science*" appears also to be a derivative of the ancient Greek γιγνώσκειν): the word carries the strong connotation of *personal experience* or *first-hand acquaintance*, as is seen in the Gospel ("*I know not a man*": ανδρα ου **γινωσκω**, Luke 1:34) echoing the *koine* Greek rendering of the ancient Hebrew ("*Adam knew Eve his wife*": Αδαμ δε εγνω Ευαν την γυναικα αυτου, Genesis 4:1; Lxx [196]). The first (Hebrew) creation account in its present (unpointed Hebrew) form dates from c.500 BCE, and its original written source is probably 7th century BCE (only a little later than Homer). But these are themselves probably derived from written sources which may have been a millennium older, and which in turn are probably remembering even more ancient oral sources [93].

We have spoken above of the idea of Creation: in the Hebrew Creation account (Genesis 1:26; Hebrew Bible c.500 BCE [195] [94]), the climax is about the creation of mankind (man-and-*woman* together; for convenience, the English indicates the four Hebrew words):

| | |
|---|---|
| Hebrew (unvocalised) | נעשה אדם בצלמנו כדמותנו |
| ποιησωμεν ανθρωπον κατ' εικονα ημετεραν και καθ' ομοιωσιν | Lxx, c.3rd century BCE |
| Let-vs-make man in-our-image, according-to-our-lickenes [*likeness*] | Geneva, 1560 |

It is interesting that the Greek of the second half of the line might be viewed in terms of formal Welsh poetry (Hopwood 2004 [208]) as (imperfect) *cynghanedd groes*: k-k-m-n/k-k-m-n (*kat eikona emeteran* / *kai kath omoiosin*). Note that the Lxx scholars considered that they did not have to repeat ημετεραν ("after *our* likeness": the possessive plural *form* is indicated in the Hebrew suffix) since the *cynghanedd* "sound" of the line allowed the hearer to imply it from the και. The Hebrew is a consonantal text which can be transliterated as: *n'śh 'dm bṣlmnw kdmwtnw*, and vocalised as ^na·'ă·śeh adam bə·ṣal·mê·nū kid·mū·tê·nū. The second half of the line might be viewed as (imperfect) *cynghanedd draws*: m-n / m-(t)-n.

Of course, there is no virtue in pretending that Greek or Hebrew poetry can be forced into the *formal* rules of the Welsh *cynghanedd*: we here only wish to draw attention to the fact that, as for all poetry, the lines are composed with an ear to their *sound*, invoking both the *breath* and the *word* (and also, obviously, the *inspiration*). And the purpose of this iconic poetic composition—which has been heavily influential in European cultural history almost up to the present day—is precisely to address the *ontological* questions: what is man? what is woman? who am I?

We should note that the Hebrew knowledge long predated and may have underpinned the Greek: Eusebius, citing precisely the antiquity of the Hebrew alphabet, insisted that "Moses *taught Plato*" (*Praeparatio evangelica*, c.320 CE [209]; see Barker 2003 [210] ch.11). We have drawn attention to the antiquity of the roots of the ideas we have been exploring: the reason for introducing Hebrew texts is specifically that they are the most ancient remaining in current use.

It is known that the modern Hebrew script [95] was preceded by a more ancient script, "palaeo-Hebrew" (Davies and Aitken 1991 [211]), in which the source documents for the modern text were probably written, and which is witnessed most famously by the "*Lachish*

*Ostraca*", confidently dated c.590 BCE: these are letters in carbon ink on clay "ostraca" that appear to be military communications in the campaign during which the city of Lachish was lost to the Babylonians. This palaeo-Hebrew script was used by Simon ben Kosevah ("Bar Kokhba", leader of the second Jewish rebellion against Rome 132–135 CE, bloodily put down by Hadrian) on the coins he minted for "Free Jerusalem" (Figure 1). But so far as we know, it was never used subsequently.

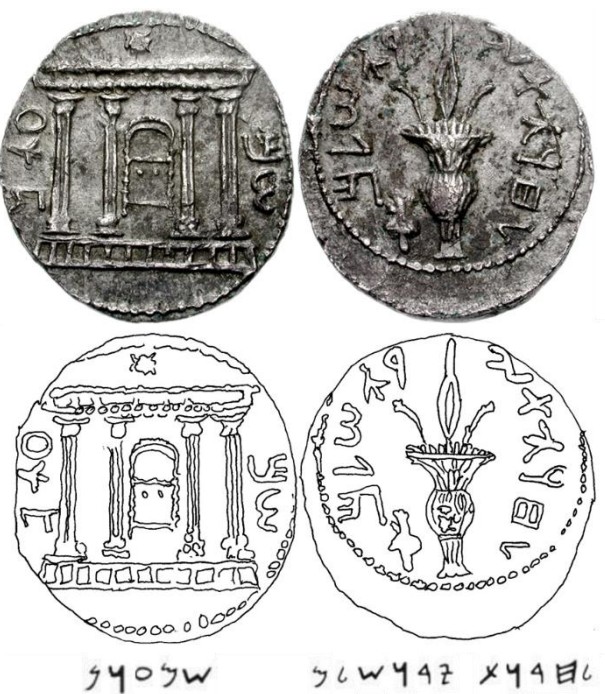

**Figure 1. Bar Kokhba silver Shekel (134/5 CE).** *Obverse:* the Jewish Temple facade with the rising star, surrounded by [שמעון] ("Shimon"). *Reverse:* a lulav and etrog, the text reads: [לחרות ירושלם] ("to the freedom of Jerusalem") (the script is palaeo-Hebrew: see Supplementary Materials for more information). *Image:* Classical Numismatic Group, Inc. http://www.cngcoins.com, CC BY-SA 3.0, https://commons.wikimedia.org/w/index.php?curid=2420146 (accessed on 14 December 2022); *Tracing:* Margaret Barker.

Light may be shed on the canonical text of the modern Hebrew Bible (c.500 BCE [195]) by referring to a gloss in palaeo-Hebrew that has recently been found in the so-called "lead books" (see Figure 2): curious traditional artefacts that have recently come to light from Bedouin communities in Jordan that are "pages" cast in an impure lead with a sophisticated pattern in relief. Many such pages can be found, usually "bound" together in a "book". The presence of a form of palaeo-Hebrew on them indicates that the original design was passed down from the 2nd century CE at the latest, and probably earlier [96]. A characteristic page is shown (the "Menorah" page, Figure 2) not only because it comments directly on our Creation text (Gen.1:26) but also because it comments in a way that clearly indicates the ambiguity and allusiveness characteristic of poetic texts that we have emphasised here. This Menorah Page can be read as a sophisticated and very extensive gloss on various aspects of Temple theology, quite possibly remembering the time *before* Solomon's Temple in Jerusalem was destroyed in 586 BCE by the Babylonians. We give only one very simple example of this gloss: there are many more which we will report elsewhere.

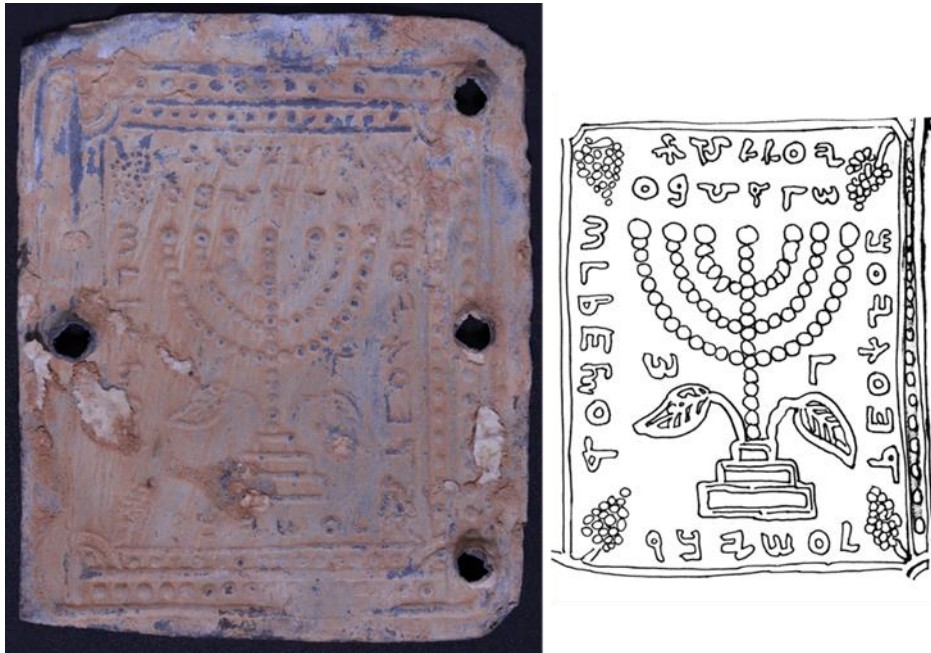

**Figure 2.** *"Menorah" page from a Lead Book. Image (left)* courtesy of Jean-Paul Bragard. *Tracing (right)* by Margaret Barker. See Supplementary Materials for more information.

In the Creation text (Gen.1:26), the Hebrew word for "*after-our-likeness*" is [כדמותנו] (*kiḏ·mū·t̠ê·nū*) from the word [דמות] (*də·mūṯ*) which in this Page (Figure 2) is read from the three letters transliterated as [דמת]. The letter *waw* ([ו]) must be added to the transliteration following the rules known as *matres lectionis*: the original palaeo-Hebrew is a pure *abjad*, but modern Hebrew is slightly impure since some letters are used to indicate vowels. The Menorah Page (Figure 2) uses a modified palaeo-Hebrew, with Hellenised symbols as well as symbols indicating double letters: unfortunately, transliterating into a readable modern Hebrew script is not entirely trivial (see Figure 3 and the Appendix in Supplementary Materials).

| # | Palaeo-Hebrew | | | Modern | Modern+ML | English |
|---|---|---|---|---|---|---|
| 0 | | | | תמד מדת דמת | תמיד מדת דמותו | continually / garments / likeness |
| 1 | | | | ישב צבא בצי | ישב בצבא בוצי | enthroned / army / fine linen |
| 2 | | | | ישע שעי עצי | ישע שעי עצי | he who saves / gazing on / tree |
| 3 | | | | יצץ צצי צצי | יצץ צצי צצי | bloom / flowers of / my flower |
| 4 | | | | יעץ עצי שעי | יעץ עצי שעעי | counsellor / my tree / delight |
| 5 | | | | חשב שבח בצה | חשב שבה בצחיה | returns / praised / radiance |
| 6 | | | | השע שעה עצה | השעע שעה עצה | delight / gazes / on her tree |
| 7 | | | | העץ עצה שעה | העץ עצה שעה | the tree / counsels / he who gazes |
| 8 | | | | העב עבה בעה | העב עבה בעה | cloud / overshadows / enquirer |
| 9 | | | | יעב עבי בעי | אעוב עבי בעי | overshadow / my cloud / seeker |

**Figure 3.** Reading the Menorah Page (see Figure 2 and text).

This word *likeness* ([דמת] *də·mūṯ*) can be found nine times in the Menorah Page, and if each triple and its mirror image is taken in three of its six possible permutations, we obtain the 10-line "poem" shown in Figure 3. The first line of this poem is formed from [דמת] permuted twice, and the following nine lines are the nine mirror images, permuted the same way (see the Appendix in Supplementary Materials for explicit details and discussion of the significance of reflections, especially Supplementary Appendix Figure A4). There are of course other ways to read this mnemonic, but we have chosen the simplest coherent

version available: a more extensive treatment is outside our present scope. It is interesting to note how the formal rules of modern information theory (involving the mathematical combinatorial and permutation operations) underpin profound truths, as perceived by the creators of this ancient artefact.

The "reflection" operation that yields the poem of Figure 3 has a general importance. We have seen how Buckminsterfullerene (Parker and Jeynes 2020 [157]), and DNA and the Milky Way (Parker and Jeynes 2019 [28]) all have Maximum Entropy geometries precisely because these all involve *holomorphic pairs*: the logarithmic double *spiral* for the spiral galaxy, the double helix for DNA, a spherical double spiral for $C_{60}$, and also, presumably, a pair of holomorphically bound "deuterons" for the alpha particle (Parker et al. 2022 [161]). "*Holomorphic*" is used here in its full mathematical meaning but applied to real objects, which can therefore be realistically thought of as *unitary objects*. "Two" have become "One" for all of these, and the two entities that form the holomorphic pair are reflections of each other.

Line #0 is the three letters transliterated [דמת] (*dǝ·mūṯ*, "likeness", Gen.1:26) with its permutations. This word occurs 9 times on the Page: lines ##1-9 are the corresponding mirror images (see Supplementary Appendix Figure A4). Hebrew is read right-to-left (*the English is read left-to-right as usual*). Readable modern Hebrew requires the addition of certain letters: the *matres lectionis*, see the column "Modern+ML". The transliteration (including ML and grammatical prefixes and suffixes) is described in the Appendix in Supplementary Materials. An abbreviated indication of the English translation is also given in Figure 3.

Then the whole *poem* can be tentatively interpreted in English as:

| | | |
|---|---|---|
| 0. | Clothed perpetually in His likeness | תָּמִיד מִדֹּת דְּמוּתו |
| 1. | he is enthroned among the heavenly host in shining linen | יֹשֵׁב בְּצָבָא לָבוּשׁ בֵּצִי |
| 2. | delighting that I know he saves | יוֹשֵׁעַ שָׁעִי עֵצִי |
| 3. | he establishes the flowering of my blooms | יְצֵץ צְצֵי צְצֵי |
| 4. | my Counsellor delights in me | יֵעֵץ עֵצִי שֹׁעֵעִי |
| 5. | He returns in glory to praises | הַשֹּׁב שֶׁבַח בַּצָּחִיחַ |
| 6. | he delights who gazed upon Wisdom's tree | הַשֹּׁעֵעַ שָׁעָה עֵצָה |
| 7. | she blesses him with wisdom | הָעֵץ עָצָה שָׁעָה |
| 8. | She overshadows the seeker | הָעָב עָבָה בֹּעֶה |
| 9. | My cloud will overshadow him who enquires of me | אָעוּב עָבִי בֹּעִי |
| 0. | Clothing him perpetually in the likeness | תָּמִיד מִדֹּת הַדְּמוּת |

where we have also given the Hebrew original in a vocalised and slightly expanded version for the convenience of Hebrew speakers (see the Appendix in Supplementary Materials for an extensive commentary).

This rendering interprets the literal text obtained from a simple 3-letter word (with its permutations and reflections), but indicates something of the mnemonic value of this *Page*. We have attempted to make the variety of allusions of the literal text explicit in the interpretation (see the Appendix in Supplementary Materials for further explanations).

While exploring the meaning of *entropy*, we have seen how we understand things. The ancient artefact shown in Figure 2 and interpreted above shows that such understanding is manifestly a property of our humanity that has been current for at least thirty centuries, where the underlying ideas date from at least a thousand years earlier. We are, and always have been, fundamentally curious about the ontological questions. Physics *uses* analytical language, the language of mathematics, as central to the coherent definition and correct manipulation of complex ideas. But the activity of definition, essential to doing physics and a precursor to explicating meaning, is itself a delicate issue. We do not wish to fall into the error of Aristotle, who thought that his definitions had an ontic reality. But neither do we wish to fall into the opposite error of thinking that because we cannot define any real *thing* our definitions can have no reality at all. Indeed, although we cannot define real things, we can define our *ideas* of them, and we can then test these ideas against reality

to see how far they are true. And insofar as our ideas are proved correct, we can without solipsism claim a (partial) grasp of reality itself, a grasp that is both ontic and epistemic.

But how can we "grasp ideas"? For this basic purpose analytic language cannot help. The strength and purpose of analytical language is to construct logically valid arguments: one can check the *consistency* of one's premises (or axioms), but one has simply to *assert* their truth. *How* do we form ideas that we are willing to assert axiomatically? How do we speak of them, and how do we understand others' ideas? When it comes to *seeing* and *knowing*, and for speaking of the *meaning* of things, we find that poetic language is indispensable.

## 7. The Poetics of Physics

Physics is the most definite and quantitative of all the sciences and, one might superficially think, the least *poetic*. Physics is the description of elemental matter—what could be simpler? what could be less poetical? Yet it turns out that we need poetry (or at least, poetic language) to be able to express our knowledge of what things are in themselves— especially such things to which common sense cannot apply—and it turns out *that* we also need this poetic language to discern how it is that we *know*. For prior to the sophisticated mathematical treatments that pervade physics is the *making of the terms* in which such treatments are done. Clausius said it in 1854: "*Bei dieser Bestimmungsweise . . . bildet . . .* ": we paint our picture (*Bild*) of things from our knowledge, and we try to make this knowledge as sure (*Bestimmung*) and as wise (*weise*) as we can (reinterpreting the meaning of Clausius' text using the range of connotations audible in German and resonating with the Anglo-Saxon roots of much of modern English). Language always has a palette of meanings even if analysts (and physicists) try to eliminate the ambiguity that the poet relishes.

We have described the development of the idea of *entropy* over the last century and a half: entropy is a *notoriously* difficult concept, even though it is fundamental to modern physics. The different uses to which the idea of entropy has been put—the assertion of the impossibility of perpetual motion, the derivation of the ideal gas laws (and the design of steam engines), the design of telecommunications networks, the properties of black holes, the stability of galaxies—these all look so vastly different, and are thought of in such different ways that it is a great leap of the imagination to see any underlying common entity. In this essay, we have tried to describe this imaginative leap.

Of course, we do not assert that physics should be (or indeed can be) done by poets (even if some poets *may* also be physicists, and some physicists poets). All must look to their own business. What we assert is that in the end the *understanding* of physics—indeed, the understanding of *any* thing—depends on inspiration. Knowledge is unitary: physicists and poets know things in similar ways, we are not so different! And physicists along with everyone else depend ultimately on language: what if our very language itself is endangered, as the Palikur poet of our epigraph bewails?

Before it is anything else, language is poetic.

## 8. Conclusions

How do we sum up? We have drawn a distinction between poetic language, and the analytic language *used* for physics. We have shown that although analytical language is designed to be unambiguous, the ambiguity inherent in all language cannot be entirely eliminated but must emerge at the foundations of any scientific argument. This was illustrated by a discussion of the foundations of thermodynamics, and the meaning of the term "entropy" which has had surprising development continuing to the present. This could be seen as an example of the '*integrated history and philosophy of science*' recommended by Bobby Vos (Vos 2021 [212]).

Although *ideas* can be defined, real *things* defy definition: ultimately, our knowledge of them must be intuited, inevitably leaving space for ambiguity and incomplete *understanding*. Mathematics is a calculus of ideas, but even mathematics is not complete: the foundations of mathematics must be established with *meta*mathematical methods!

For many scientific purposes, we can avoid the basic ontological questions, like: "*What is entropy?*" Provided we know how to calculate the quantities of interest, we can be satisfied for practical purposes. But questions of a different nature require a more developed philosophical approach: How secure is the knowledge our scientific advisors claim to have? Why should I trust scientific advice? How do I evaluate conflicting advice given by various technical experts? Such questions have become particularly salient during the recent COVID-19 pandemic.

We have already cited Iris Murdoch's acknowledgement of the ethical importance of our *understanding* of *thinginess*: we here wish to refrain from venturing into ethics despite her insistence on a valid philosophical account of *goodness*[97]:

> The necessity of the good is then an aspect of the kind of necessity involved in any technique for exhibiting fact
>
> Murdoch, On 'God' and 'Good', 1966 [78]

where of course, in speaking of "*exhibiting fact*" she has our activity of science in mind. She says, "*we are moral agents before we are scientists, and the place of science in human life must be discussed in* **words**" (*ibid.*, emphasis original)[98], and she adds, sarcastically:

> Let us consider the case of conduct. What of [Jesus'] command, 'Be ye therefore perfect' [Matt.5:48]? Would it not be more sensible to say, 'Be ye therefore slightly improved'?
>
> Murdoch, On 'God' and 'Good', 1966 [78]

What scientists writing a paper would be happy with a text they knew was barely adequate? All scientists strive for perfection even as we know it is unattainable, strictly speaking. We take Murdoch's position on "*the infinitely elusive nature of reality*" (Murdoch 1962 [100]), and note that she links the *good* firmly to reality: "*Good is indefinable . . . because of the infinite difficulty of the task of apprehending a magnetic but inexhaustible reality*" (*ibid.*)[99]. And on "science", we note that, in the context of the description of the unfortunate man in 1975 who remembered *nothing* after 1945, Oliver Sacks comments that "*Empirical science . . . takes no account of the soul, no account of what constitutes and determines personal being*" (Sacks 1984 [214]). The knowledge of reality is irreducibly *personal*: and in the end cannot adequately be described by a limited ("empirical") view, whatever immediate utility such a view might have. Michael Polanyi has highlighted how essential morality is to the defence of the "free society" in general, and to the effective pursuit of science in particular, speaking of "*what we as a people are passionately committed* to" (Polanyi and Prosch, 1975 [22], Ch.13). A Reviewer has called this present essay "passionate", and so it is; nevertheless, we wish here to limit our attention (as dispassionately as possible) to the philosophical issues, since these are widely overlooked—or misinterpreted.

For any answers to the ethical, social and political questions hinted at above all depend on our basic understanding of what *knowledge* itself is. We have here sought to show how, at the very foundations of physics, we rely on a *poetic* handling of language even to define the terms we use to articulate the ideas we need to understand the world. Just as the foundations of mathematics are established with metamathematical methods, so the foundations of physics must be established with metaphysical methods.

And natural language is its own metalanguage: this is why, ultimately, we must rely on poetics. The *meanings* of things are always intuited: Leon Wieseltier has said, "*The knowledge of a thing is more decisive than the sight of it*" (Wieseltier 1999 [215], p. 22): note that we insist on the distinction between *information*, which is physical, and *knowledge*, which is mental. Formally, physics can treat only information: to treat knowledge we properly have to use metaphysics—or, ultimately, poetics. Wieseltier also said, "*The place of science in life cannot be scientifically established*" (*ibid.* p. 256), and McLeish (2019 [7], p. 28) quotes Shelley: "*Poetry is . . . at once the centre and circumference of knowledge; it is that which comprehends all science*" (Shelley 1821 [216]).

We have explored the articulation of some basic ontological ideas with the help of a tangible artefact (the "Menorah Page", Figure 2) whose design appears to be very ancient, remembering times long prior to the Hellenic schools of philosophy. We have reconstructed from this artefact a mnemonic that makes an exceptionally sophisticated poetic gloss on the idea of [דמת] (*də·mūt̲: likeness*) which is a word in the Hebrew language with a similar set of connotations to Plato's "*forms*": that is, it is specifically concerned with the *knowledge* of things, underlining that the search by humans for knowledge and the desire of humans to grasp reality long predated the invention of what we now think of as "scientific" language.

How do we *know* things? Rationality itself is as fundamental to poetic language as it is to analytic language: the appreciation of poetry depends critically on the recognition of poetic *form*. We have interpreted this artefact (the "Menorah Page") as a *mnemonic* of ideas looking back at least three millennia, probably much more: it focusses our attention on the *logical continuity between different sorts of knowledge*. We conclude that whether the knowledge is of concrete or numinous things, the *rationality* used to handle and articulate it is common.

Before anything else, language is poetic. The foundations of physics cannot in principle be established analytically: they must be constructed metaphysically, using the poetic properties of language.

## 9. Epilogue

God is One

Man is two: woman is too

Love is three: binding mankind whole

Right is four-square: breath of man and breath of woman

Mercy is prime five: both two summed with three and one summed with four

Mankind's number is six: love's mutuality

God's number is prime seven: the resting of creation

The first perfect cube conquers death: man and woman joining in life

<div align="right">© 2020 C.Jeynes</div>

**Supplementary Materials:** The following supporting information can be downloaded at: https://www.mdpi.com/article/10.3390/philosophies8010003/s1, See Supplementary Materials for the audio file of the poet speaking the poem in *Palikur* (and for its *Portuguese* translation), also for the *Náhuatl* original of *Cuando Muere Una Lengua* (and its translation into *English* from the *Spanish*). Supplementary Materials also include the important ***Appendix*** with supporting documents.

**Author Contributions:** Conceptualization, C.J., M.C.P., M.B.; methodology, C.J.; writing—original draft preparation, C.J.; writing—review and editing, C.J., M.C.P., M.B. All authors have read and agreed to the published version of the manuscript.

**Funding:** This research received no external funding.

**Acknowledgments:** We thank Nuno Pessoa Barradas and Les Wheeldon for their translations of the long quote from Clausius (1854), and NPB for detailed comments. CJ is grateful to Michael Mantler for drawing his attention to Korpel & de Moor's book (Korpel & de Moor, 2014 [206]) at the European X-ray Spectrometry conference (Bologna, June 2014), and to Catia Costa and Sulam Peña Mena de Deaville for discussing respectively the Portuguese and Spanish versions of Ku *wown biyuke* and *Ihcuac thalhtolli ye miqui* (see Supplementary Materials).

CJ appreciated comments by Adrian Tuck, and is also grateful to Vesna Goldsworthy for drawing his attention to Wallace Stevens's poem (email May 2021), and to Stephen Lake for drawing his attention to E.R.Dodds' book (with other comments, email September 2021), and to Rachel Holland for pointing him to Karen Barad's important monograph. CJ is also grateful to Julia Jordan for helpful discussions and for alerting him to Iris Murdoch's precedence in the idea of "thinginess" (even if

the neologism itself is due to Michael Frayn), and for introducing him to Anthony Ossa-Richardson's important monograph on Ambiguity [73].

We thank Aldiere Orlando, Silvia Calusi, and [ק׳רם זייצ] (Ram Zajicek) for speaking respectively the Palikur, Italian, and Hebrew texts in the Audio files of the Supplementary Materials. We are very grateful to Diana Green for obtaining (and translating) Orlando's poem and for persuading him to speak it for us. We are also grateful especially to [יוסף גופר] (Yosef Gofer) for extended discussions which dramatically improved the text, and for persuading [ק׳רם זייצ] to speak the Hebrew poem for us.

CJ is grateful to Martin Edwardes, Peter Kevern, Magnus Course, and R.Maxwell King for detailed comments on the text; all of which have substantially improved it. CJ is also grateful to extended discussions with Joan Vaccaro and Jerome Ravetz, and also Cassandra Gorman to whom he was introduced by Kevin Killeen.

We are appreciative of the helpful and perceptive comments of anonymous referees acting for the journals *Studies in the History & Philosophy of Science*, *Synthese*, and *Philosophies*. These have stimulated a much more complete view in this version.

**Conflicts of Interest:** The authors declare no conflict of interest.

## Notes

[1] This strongly suggests Basarab Nicolescu's idea of "Transdisciplinarity" [4] which has many merits (including both the very clear assertion that "*knowledge is open*" and also the recognition that disciplinary boundaries are essentially arbitrary, as we say here) but from our point of view it does not treat the "subject-object" distinction satisfactorily. Nicolescu correctly attacks the idea that it is possible to completely separate "subject" from "object", but still thinks that the "*three axioms of the methodology of transdisciplinarity*" may establish his (transdisciplinary) discipline. Clearly, his is a valuable contribution, but here we are saying something different: that *meaning* is necessarily *poetic*.

[2] A.N.Whitehead is famous for saying, "*the safest general characterisation of the European philosophical tradition is that it consists of a series of* footnotes *to Plato*" [14]. Everyone (including Whitehead himself) who comments on this aphorism hedges it about with qualifications, but nevertheless the Socratic analytical style is thought to be properly "philosophical" where the poetical (or "mystical" or "religious") styles of other ancient peoples is not. We think this is misleading, and will adduce evidence for this long predating the Hellenic period.

[3] So Odin plucked out an eye to gain knowledge, and when Jesus said, "*seeing they see not*" (Matthew 13:13 *passim*; 1st century CE) he was quoting the pre-Hellenic seer (Isaiah 6:9f; 7th century BCE) who wanted the people to "*understand with their hearts*". This saying of Jesus is one of the few attested in all four Gospels, underlining its central importance in European culture of the last two millennia: this attitude to *seeing* also underlies (almost invariably tacitly) the European philosophical tradition.

[4] At one point in a detailed monograph on unconscious and/or unexamined bias, Jessica Nordell says, "*It may be that I did not even see [the operation of* bias*]*" [15]. We see only what we believe, to an extent that is surprising (and often shocking). In another place she comments: "*We have to develop the ability to truly recognize what we see*" (*ibid*, p. 164, citing Kahneman and Klein 2009) [16].

As a STEM Ambassador I regularly play the "*how many eyes have you got*" game with 9-year-olds. To do science you have to *see* with your *mind's eye* (so you must have at least *three* eyes). St.Paul, writing to the Ephesians in 58 CE (using the conveniently definite dating of Robinson, 1976 [17]) wishes that their "*eyes of the understanding be enlightened*": πεφωτισμένους τοὺς ὀφθαλμοὺς τῆς καρδίας ὑμῶν (*pephōtismenous tous ophthalmous tēs kardias hymōn*; literally, "being-enlightened the eyes of-the heart of-you": Eph.1:18). Note that the old authors considered that we *feel* with our *heads* but *think* with our *hearts*; they would consider the modern head-heart dichotomy deranged. Classically, the *mind* is located in the *heart*, which is why we speak of "*the heart of the matter*".

"*Seeing*" is a central idea in this essay—John Berger's contribution, "*Ways of Seeing*", [18] is now iconic, but is equally applicable even in physics. Bohm and Hiley (1993 [19], §14.1, p. 323) say "*Our theories are not primarily forms of knowledge about the world but rather, they are forms of insight that arise in our attempts to obtain a perception of a deeper nature of reality as a whole . . . new concepts . . . serve as ways of looking at reality*". They go on to speak of "*unbroken wholeness*" (*ibid*. §15.2, p. 352). Michael Berry points out to me (private communication 5 December 2022) that Bohm's formalism was anticipated in detail by Erwin Madeling [20], but Bohm apparently was unaware of this. Berry himself [21] (§4) has noticed "a germ" of the same idea in the treatment of caustics in optics going back to Newton himself.

[5] A Reviewer has suggested that since Jorge Wagensberg [23] and others "*affirm the metaphorical foundation of sciences*" one can hardly treat the "*unity of truth*" as a given. This is an error. It is true that science (along with all knowledge, as we here insist) does have a metaphorical *foundation*, but here *metaphor* does not denote a mere "figure of speech": on the contrary, *metaphor* is fundamental to all language, as we will show in detail (this indeed is a commonplace of the poets). And if truth is not unitary then we may *feel* many things, but we can *know* nothing at all. Note that saying "*truth is one*" is <u>not</u> the same as saying "*truth is simple*". Reality is elusive.

6     The ontology of "*physical* law" is well-trodden ground (see for example Bird [29]) and out of our present scope.

7     Barfield says of this dictum: "*The* **use** *of them [the terms] is left to the Logician, who, in his endeavour to keep them steady and thus fit them to his laws, is continually seeking to* **reduce** *their meaning. I say seeking to do so, because logic is essentially a compromise. He could only evolve a language, whose propositions would* **really** *obey the laws of thought by eliminating meaning altogether. But he compromises before this zero-point is reached*" (*ibid.*, emphases original). This is entirely consistent with our view, *mutatis mutandis*.

      The idea that the "*poet's vocation . . . [is as] a* **maker** *of words*" (emphasis original) is now a cliché—this quote is taken randomly (from Oliver O'Donovan's 2021 essay [30] on Psalm 119).

      In a subtle work now largely overlooked, Barfield (1957 [31]) also investigates the "modern" attitude to *hypotheses* in a treatment close to ours here in certain important respects. He says that the post-Copernican attitude is that "*if a hypothesis saves all the appearances it is identical with truth*" (*ibid.* Ch.VII) where of course his debt to Pierre Duhem (1908 [32]) is clear and explicit in his treatment. Much has happened since 1957 (including the establishment of the reality of both black holes and the cosmic microwave background), and this description of *hypotheses* (which characterisation does not adequately represent the breadth of Barfield's argument) is clearly a version of naïve realism, which is no longer tenable (if it ever was). In our terms he claims that the modern (post-Copernican) view confuses the formalism with the reality (for further on "reality" see below, *note*#31).

      Moreover, his gloss on the Hebrew Scriptures, not only regarding the (temporal) priority of Hebrew thought over Greek, but also underlining both the *otherness* of the Hebrew mode of thought and the metaleptic nature of the Christian (Greek) Scriptures' dependence on the Hebrew Scriptures, is entirely consistent with our view (see §6 below).

8     We will quote Thomas Piketty (2019, [34]) saying "*there is nothing 'natural' about language*" (§3): by "natural language" we mean the language we use every day without any special technical vocabulary. This is the same language used for novels and the same language used for poetry, although in both cases the authors or poets may use a register of language that can hardly be called "everyday". We always compose (or write) for an audience: this will determine our style. But composition for a general audience cannot use any special language. We will call this, for want of a better term, "natural" language.

      Note that this is *not* a "definition", and is not intended to be one. We are not attempting to construct any sort of "analytical metaphysics" (on "metaphysics" see *note*#15 below). Formal definitions use analytic (not poetic) language, but we here merely explain informally which meaning we intend.

9     Martin Edwardes treats the idea of "negotiating meaning" at length, but for convenience we summarise it here with an example. If, driving down a single track road, I meet another car, who has priority? We both have right of way, but neither of us has priority, and we have to "negotiate" who backs up to the nearest passing place. Without agreement, we cannot move. If I want to say something straightforward so that you understand me, we already know how to do that. But if I want to say something new, you may not at first know what I mean: then we have to together "negotiate the meaning" until you are satisfied.

      This idea of "negotiating" meaning is far-reaching. Sidney Dekker [36] (p. 16), in a deep and influential discussion of systematic ways to avoid disasters in various industrial processes, points out that "*Systems are not basically safe. People create safety while negotiating multiple system goals*". He continues: "*Human errors do not come unexpectedly. They are the other side of human expertise—the human ability to conduct these negotiations while faced with ambiguous evidence and uncertain outcomes.*

10     The "*scientific method*" itself has been discussed in detail recently by Jorge Wagensberg [23], who proposes a well-defined ("unique") programme which is supposed to distinguish what is properly "scientific". We cannot digress to adequately critique this paper here, but we point out that he relies on concepts such as "objectivity" (and he calls Popper's "falsifiability" idea "*sublime*"). Whatever the merits of these, our discussion includes the ontology of "objectivity" where his does not.

11     Kahneman and Klein in their classic paper investigate this *intuition* in detail: "*expert intuition is sometimes remarkably accurate and sometimes off the mark*" [16], of course they are considering real technical ("expert") knowledge. They explicitly point out that the cues experts use to make their judgments "*involve tacit knowledge [which is] difficult for the expert to articulate*". On tacit knowledge of course see Polanyi 1958 [38] (and our discussion below §3). On the proper use of (and limits to) *expertise* see also Collins and Evans (2007) [39].

12     This is evidenced by all the physics papers cited in this work, including Gödel, 1931 [43]. Interested readers could also look at the historical papers collected in translation by Stephen Hawking [44], in particular the seminal work of George Boole, Richard Dedekind, Kurt Gödel himself, and Alan Turing (Hawking's other selections are mostly extracts). The introductory sections of these major papers have no equations: but they are not "wordy"—diffuse and vague—far from it! They are crucial to understanding the mathematical arguments that follow.

13     Interestingly, the dancer Emily Coates [46] has recently written about "*The Poetics of Physics in Dance*". She points out that a simple "translation" of physical ideas into dance terms cannot succeed either in effectively communicating the scientific ideas or of creating effective art: she says, "*science is not art . . . as historians of science have been noting for some time, the practices of science and art have more in common than many have assumed, and using the framework of translation doesn't allow for their overlap.*" Instead, she speaks of "intensification" and (particularly relevant to our purposes here) "*an intensification of our encounter with time . . . dance has the power to perceive and move with its subjects to such a degree as to reveal scientific* **ontologies** *that would otherwise be inaccessible*" (my emphasis). She speaks of "*ideas in motion: scientific laws made hyper-perceptible*". She cites Steve Paxton [47], in whose "*alluring poetics, physics is social, and sociality is expressed through physics*". We are engaged in something rather different here, but in her conclusion Coates underlines our guiding principle in this essay, that physics is *irreducibly personal*: she says, "*When scientific knowledge assumes* bodily *form . . . the unavoidable human source of science becomes poignantly clear*" (emphasis original).

14  see Part II—"*The Tacit Component*"—of his *Personal Knowledge* [38]. See also Mary Jo Nye's recent assessment of Polanyi [48].

15  We are not here concerned with how much animals know: so far as we are aware, science is human. Here we are interested only in how *we* know things, not in how other creatures may possibly know things.

16  "*centerless*": perhaps recalling the strange case of Phineas Gage and his loss of responsibility following his extraordinary injury as described famously by Hanna and Antonio Damasio [50,51]. Their account is expanded (and corrected) by Macmillan (2000) [52].

17  We use "*metaphysics*" as a term cognate to (*e.g.*,) "*metanarrative*", "*metamathematics*", "*metadata*": that is, a "natural" language (see *note*#7) in which one may speak of the underlying features of the language or activity in question (the art of narrative, mathematics, the structure of the dataset). Note that Graeber and Wengrow [11] (p. 31) refer to "*a standard historical metanarrative*" in their important work. So here we use "*metaphysics*" to mean "*the metanarrative of physics*". See below §5.2.

  We do *not* intend the conventional (often pejorative) range of meanings normally associated with "*metaphysics*". In particular we will in this essay directly address (as "ontology" or "epistemology") what is usually spoken of as *metaphysics*, the first meaning of which is given by the Oxford English Dictionary: "*The branch of philosophy that deals with the first principles of things or reality . . . theoretical philosophy as the ultimate science of being and knowing*" (OED 3rd Edition 2001) [53].

  As Rasmus Jaksland points out, "*there is no univocal definition of metaphysics*" [54], although he proceeds on the basis that metaphysics ought to be "naturalised" with a subject matter essentially indistinguishable from physics proper, except that the proper physical treatment has not yet become clear. We think that this approach is overcomplicated as well as apparently begging the basic ontological and epistemological questions: even the scaled back proposal of "*Moderately naturalistic metaphysics*" (Morganti & Tahko [55]) acknowledges that "*metaphysics*" (in the OED meaning) may explore "*a basic possibility space in such a way that the grounds for the interpretation of scientific theories are laid*" in just the way we will describe here. We wish to speak of *metaphysics* simply as the ("natural") language used to set up the model of the physical system (with *entropy* as an example: §4.1).

  Therefore, we think that James Ladyman's suggestion that "*The first task of the metaphysician, like the scientist, is to construct a hypothesis that accounts for the phenomena in question*" [56] is beside the point. It is the job of the scientists to account for the phænomena, but what terms are available to them? New terms for the new science must be constructed ("*negotiated*", [26]) out of existing language in a way comparable to how Clausius did it for "entropy" in 1854 [35] (see §4.1). In any case, as Karen Barad says: "*the primary ontological unit is the phenomenon*" [24](p. 333). We will discuss this further in §5.1 below.

  In the context of a discussion of the unity of truth, Michael Polanyi criticises an eminent professor who, attempting to escape positivism, creates "*a labyrinth of subterfuges in order to say ("scientifically") that [certain people] regard the seeking of the truth to be their right and duty*", calling such pseudo-scientific circumlocutions a "perversion", and saying that the positivists were "*repudiating metaphysics along with dogma*" [22] (Ch2). Here Polanyi uses "metaphysics" in our sense.

18  Charlotte Higgins (2022)[59] quotes Daryna Gladun (a Ukrainian poet based in Bucha before the Russian invasion on 24 February 2022): "*[my work] sits at the boundary between literature and journalism. It is poetry in uniform. I set aside metaphors to speak about the war in clear words*." Higgins comments on this: "*It interests me that she uses metaphor to describe this process at the same time as apparently eschewing it: in the end metaphor will out. If war involves a fracturing of language, it is poetry that will eventually creep in to fill the gaps*". Metaphor will out! We cannot say what is close to our hearts without "metaphor". Metaphor is ***not*** merely a "figure of speech", it is itself the very lifeblood of speech.

19  The writer to the Hebrews (Robinson, 1976 [17]) says, Ἔστιν δὲ πίστις ἐλπιζομένων ὑπόστασις (Heb.11:1; "*faith is the substance of things hoped for*"): ὑπόστασις (*hypostasis*) is a deep idea explained in theological (Trinitarian) terms by Lienhard, 2002 [60]: the point here is that the idea of *substance* is intrinsically metaphorical, just as Iris Murdoch says.

20  Our thesis is very restrained: it uses only the English, Palikur, German, Italian, Latin, Greek and Hebrew languages (although also alluding to Portuguese, Náhuatl, Spanish, Gaelic, French and Welsh). The comparative linguistics, which are pregnant with ontological implications (Perunović 2021 [61]), are here only hinted at. These implications exist in an attenuated way even for "simple" data handling (Forkel 2018 [62]).

21  "Standard empiricism", believed to be "objective", has its usual meaning (Maxwell 2020 [63]). Nicholas Maxwell shows how the usual treatment smuggles our own *ideas* in in the same way that Polanyi (1958 [38]) shows that knowledge is personal (the unavoidable subjectivity is smuggled in by how we speak of "simplicity"—see §3). That is, our understanding is *idealistic*: it depends on our own personal (or "subjective") ideas.

  Note that *measurement* itself is now acknowledged by the metrologists to have an irreducibly subjective element, see in particular §5.1 and *note*#73. Note also that the subjectivity of our knowledge (that is, its necessarily *personal* nature) does not undermine it in any way, since it is socially validated by the peer review system (Hicks 2016 [64]).

22  One sort of "naïve realist" might be what Timothy Lyons (2006 [65]) calls the "*deployment realist*" who says essentially that scientific theories are true because they work. Lyons explains in detail the multiple problems of this view, also confronting the literature at length. Michael Berry (2020 [66]) wittily contrasts the "idealist" and the "realist" with his aphorism "*True but not real*" (applied to scientific explanations) contrasting with the opposite idea ("*real but not true*") applicable to our (real enough, if possibly misleading) perceptions of the phaenomena. The example is the table—we push the explanation down to the quantum level—all true enough, but where has the table gone?

23  We give dates for the *New Testament* texts (conveniently but anachronistically reproduced in a miniscule Greek script with word spacing) as authoritatively discussed by John Robinson (1976 [17]).

24      Tyndale's translations were printed in a blackletter font, but for convenience we transcribe here in a Latin font. German texts were usually printed in a blackletter font (known in German as "*Fraktur*") until it was discontinued in 1941. But the German texts of Gödel and Clausius (§§3,4.1) were printed in a Latin font.

25      *Ambiguity* has been explored in detail in an important monograph (Anthony Ossa-Richardson, 2019 [73]), showing that although our modern appreciation of poetic ambiguity can be traced back to William Empson's seminal work (Empson 1930 [74]), it has a long prehistory in Europe reaching back to Augustine and the Christian understanding of the Hebrew Scriptures. Ossa-Richardson underlines our point here when he quotes Eugen Bleuler (1914 [75]) approvingly as "*seeing in ambivalence . . . 'one of the most important mainsprings of poetry'* [*Die Ambivalenz is eine der wichtigsten Triebfedern der Dichtung*]" (*op.cit.* p. 373). Ossa-Richardson (*ibid.*) comments that Robert Graves would later agree with this judgment.

26      "*Wovon man nicht sprechen kann, darüber muss man schweigen.*" (Wittgenstein 1921 [76], §7)

27      "*Man kann* für *eine große Klasse von Fällen der Benützung des Wortes »Bedeutung«—wenn auch nicht für alle Fälle seiner Benützung— dieses Wort so erklären: Die Bedeutung eines Wortes ist sein Gebrauch in der Sprache*" (Wittgenstein 1953 [77], §43; emphasis original). This is translated as "*For a large class of cases—though not for all—in which we employ the word "meaning" it can be defined thus: the meaning of a word is its use in the language.*" Wittgenstein died in 1951: the *Philosophical Investigations* was published posthumously.

28       . . . although Wittgenstein is usually spoken of either as a positivist (early) or a pragmatist (late) of the sort that Iris Murdoch takes aim at. So he says, "*Wie haben wir denn die Bedeutung dieses Wortes (»gut« zum Beispiel) gelernt? An was für Beispielen; in welchen Sprachspielen?*" ("*How then did we learn the meaning of this word ("good", for example)? What examples are there; which language games are being played?*"; Wittgenstein 1953 [77], §77) but Murdoch specifically attacks this point of view in her essay "*On 'God' and 'Good'* " (Murdoch 1966 [78]).

29      So when one young Palikur man saw a large city for the very first time (from an airplane) he shouted, "*Babamnamnapadmin!*", which means, literally, "*repeated protrusions, many, tall, widespread, vast beyond comprehension*". It would take an English speaker a paragraph to describe a city to someone who had never seen one before, but a Palikur speaker can pack it all into one adjective. This entire word was coined on the spot and likely had never been spoken before, yet any Palikur speaker would immediately understand it, and they would have a good idea of what a city looked like from the air. Impressive! (Private communication, email from Diana Green 9 Septembr 2022.)

30      Course and MacMillan (2021 [85]) have similarly observed, speaking of the threatened loss of the Gaelic language: "*The culture, the people,* everything *would disappear forever*".

31      This is strictly correct if we include the *language* of music or art (McLeish 2019 [7]), and if we note that Polanyi's "tacit knowledge" is predicated on students picking up this tacit knowledge in the lab by doing things in collaboration with (and *talking* with.) supervisors and colleagues (Polanyi 1958 [38]).

32      "*Thingification*" is an interesting word whose first usage the Oxford English Dictionary (OED) attests in 1935; the OED lists *thinghood* as used by A.N.Whitehead, but sadly does not list the (better) synonym *thinginess* (philosophers might speak instead of "*reification*", a Latinist neologism of the mid-19th century). The OED also attests all of *thingly* (adjective), *thingy* (both as a noun and an adjective), *thingness*, and *thingliness* (respectively 1450, 1787, 1891, 1840, 1662). Bruno Latour (2004 [81]) also spoke of "*thinginess*". We are not much concerned here with Bill Brown's "*Thing Theory*" (Brown, 2004 [87]), which is interested more in *our relation with things* than the things-in-themselves.

33      Of course, Augustine was thinking about "massive" things, not mass-less ones like the photon. It is an elementary result of special relativity that the closer one's speed is to the speed of light the slower "time" passes: thus for photons "time" stops altogether. This very simple and straightforward result is entirely counter-intuitive which remains very shocking however familiar it is: see further on the individuality of things in general (and electrons in particular) at *note*#89 below.

34      "*Real*" is another nice word whose extensive cluster of connotations includes *royalty*—well known today by the many followers of premier-ranked *Real Madrid*: monotheists would say that the idea of *reality* derives from God the 'King of kings' (that is, God the Creator). See also the discussion of "Wigner's friend" scenarios in *note*#37.

35      This point of view was famously proposed by Herbert Butterfield in his *The Origins of Modern Science* (1957 [96]), which remains important as is discussed (for example) by Regis Cabral (1996 [97]) and Cunningham and Williams (1993 [98]). Butterfield's style of argument may appear dated, but it seems to remain uncontentious that his substantive points remain valid.

36      Charlotte Higgins (2022 [59]) reports the Ukrainian writer Oleksandr Mykhed as saying: "*It is true that when Russia's full-scale invasion started, on 24th February, literature was the last thing on people's minds—you could not protect your family from a rifle with a poem.*" Higgins continues: "*But as the conflict has continued, the power of writing to record, to testify and to witness has seemed more and more important . . . diaries . . . able to gather experience and emotion in its rawest form.*" Higgins reports the Ukrainian scholar Oksana Maksymchuk speaking of poetry's ability to "*crystallise a particular moment in time, or an emotion that is fleeting.*" If poetry flourishes even in war, does that not underline our desire—and need—to *know* things truly? And true knowledge is essentially poetic.

37      It is today a commonplace that *we see only what we believe* (for an experimental example of this see Zhaoping 2007 [103]). Of course this is an exaggeration, but it is always an effort to allow for our biasses (and we never completely succeed). So in the line after "*meeting the universe halfway*" Alice Fulton [102] says (as it were in explanation), "*nothing will unfold for us unless we move towards what looks to us like nothing.*" We creep towards knowledge in ignorance. But however fallible and incomplete, we may

still know truly, we may still touch reality. Fulton concludes: "*let my glance be passional towards the universe and you.*" Note that for the poets knowledge is *always* personal.

On *bias*, Caroline Criado-Perez (2019 [104]) draws attention to what surely is an unforgivable bias expressed in the systematic disregarding of differences between men and women in medical research. Richard Douglas Fields (2014 [105]) opines that "*Testing Males and Females in Every Medical Experiment Is a Bad Idea*", and Sarah Richardson (and others, 2015 [106]) opine more subtly that focussing "*on preclinical sex differences will not address women's and men's health disparities*" which may be true but the main issue that Criado-Perez is drawing attention to is the enormous and systematic *data-gap* regarding sex-based differences in medical outcomes.

38　Such "Wigner's friend" scenarios mean that observer-independent "facts" cannot be determined in these cases, but this does *not* mean that these experimenters are thereby not realists. Just because an idea of "objective reality" can be shown to be illusory in at least some cases does *not* mean that the reality itself is illusory. Instead we should conclude that our ideas of it may be mistaken. As Karen Barad points out, such experiments are seriously difficult and to successfully carry them out one must be pretty sure that Nature really is there and will play ball. Barad says, quoting from Greenstein and Zajonc (1997 [110]): "*it is not trivial to detect the extant quantum behaviour in quantum eraser experiments. The experimenters must be clever enough to design an experiment that can detect the entanglement*" (Barad 2007 [24] p. 348). But asserting (in our terms) that the *thinginess* of the phænomena being investigated is ontologically secure is *not* asserting even that we can know it! So Barad asserts, citing Bohr himself, that "things" cannot be taken as "*ontologically basic entities*" (*ibid*. p. 138). The thinginess of things is ontologically secure precisely because it is *phænomena* that are "*ontologically primitive*" (*ibid*. p. 333).

39　"Subject" and "object" are primarily terms of grammar. In the phrase "*objective knowledge*" the adjective "objective" is being used as a (grammatical) metaphor which we here regard as empty: with Polanyi and Barad, we regard the philosophical conception of "objectivity" as being essentially mistaken. To minimise confusion we will therefore avoid the use of the terms "subjective" and "objective".

40　"Objectivity" is a strangely slippery idea, for all that it is supposed to straighforwardly underlie the "scientific method" (and see note#9). Caroline Criado Perez (2019 [104]) gives an interesting example when she criticises the Bank of England's (allegedly) "objective selection criteria" for choosing "key figures from the past" to grace British banknotes.

Michael Polanyi says further: "*Thus the ideal of pure objectivity in knowing and in science has been shown to be a myth*" (Polanyi and Prosch, 1975 [22], Ch.3). He insists: "*the processes of knowing (and so also of science) in no way resemble an impersonal achievement of detached objectivity . . . Scientific inquiry is accordingly a dynamic act of the imagination and is rooted in commitments and beliefs about the nature of things. It is a fiduciary act . . . Its method is not that of* **detachment** *but involvement.*"

41　Jessica Nordell insists that this applies to maths too: "*Math after all, is personal, emotional*" (Nordell 2021 [15], p. 245; Nordell quotes Erica Klarreich, 2017 [112], saying "*Anybody who does mathematics knows this*")

42　Poets that is, and anyone using poetic language such as the novelist: "*We must be ruled by the situation itself and this is unutterably particular. Indeed this is something to which we can never get close enough, however hard we may try to as it were crawl under the net.*" (Murdoch,1954 [91]; ch.6, p. 90). This is "Annandine" (Hugo's alter ego) speaking. The novelist (who identifies with Hugo's point of view) makes her point by indirection: going "*under the net*".

43　Using the conveniently definite dating of Robinson (1976 [17]).

44　This recursive form of logic remains of interest today (Leonard, 2021 [117])

45　First published (in French) in the *Discourse on Method* (Descartes, 1637 [118]): "*Je pense, donc je suis*". The Latin version (*Principles of Philosophy*, Descartes 1644 [119]).was dedicated to Elizabeth of Bohemia.

46　And we may grasp this illumination by the "*leap of faith*" (properly, the "*leap by faith*" over Lessing's "ditch") proposed by Søren Kierkegaard explicitly in his "*Unscientific Postscript*" (Kierkegaard 1846 [122]) and implicitly in his discussion of Abraham's faith in "*Fear* and *Trembling*" (Kierkegaard 1843 [123]). It has been heavily debated ever since; Kierkegaard was very influential in 20th century philosophy. Earlier we called the same thing an "*intuitive leap*".

47　To be precise, where Gödel proves a tautology (all theorems are formal manipulations in logic) Anselm asserts something about the real world. I think nobody believes that the reality of any thing can be established merely by "*the analysis of concepts*" (Ward 2008 [12], ch.6), however, as Keith Ward [124] points out, the Ontological Argument "*shows that God is either necessary or impossible (that is, either cannot fail to exist, or that the concept is incoherent). But we cannot, simply by thinking, establish which.*" (*ibid*.) Here we are interested in the analysis and epistemology of concepts in general (including scientific concepts), but not in the theological or ethical discussion of them. Such discussion is of course important, but outside our present scope.

48　"consistent" means that no false proposition can be proved in the system; "complete" means that there exists a proof in the system for every true proposition. Gödel cites the 1925 edition of Russell and Whitehead's monumental work *Principia Mathematica*, and also David Hilbert's work in arithmetic (Russell and Whitehead 1910 [125], Hilbert and Ackermann 1928 [126]): note that "arithmetic" is emphatically *not* "child's play"!

49　where by "ideological" we intend to mean merely "of ideas". Of course there is a difference between the ideology of economics (which is is highly political), and the ideology of physics (which is mostly philosophical). We are *not* here alluding to the politics of science, which is not our present focus important though power relationships are in the development of science.

[50] The following summary is given by Velazquez, Parker and Jeynes (Velazquez et al., 2022 [72]), although it is still not entirely clear how to interpret this thermodynamically:

"Before 1956, physicists believed that the so-called parity P, charge C and time T symmetries were unbreakable in quantum field theories. However, different experiments concerning weak forces confirmed along the years the violation of P-symmetry (Lee and Yang *Phys.Rev.*, 1956; Wu et al. *Phys.Rev.*, 1957), the violation of C symmetry (Ioffe et al. *JEPT* 1957, Lee et al. *Phys.Rev.*, 1957), as well as the violation of the combined CP-symmetry (Christenson et al. *Phys.Rev.Letts.*, 1964, Alavi-Harati et al. *Phys. Rev. Letts.*, 1999, Fanti et al. *Phys.Letts.B*, 1999).

"Nowadays, the standard model of particle physics only admits the combined CPT-symmetry (Schwinger *Phys.Rev.*, 1951), in which the effect of time reversal is equivalent to renaming particles as antiparticles (and vice versa: see Lüders *Dan.Med.Fys.Med.*, 1954). By itself, the existing violations of the CP-symmetry requires the violation of T-symmetry to complete the CPT-symmetry. The direct observation of a time-reversal violation (but maintenance of the CPT-symmetry) was confirmed by the experiment conducted by the "BaBar" collaboration in 2012 (Lees et al. *Phys.Rev.Letts.*, 2012). This experiment took advantage of entangled B0 and $\overline{B}$0 mesons in the U(4s) resonance produced in positron-electron collisions with the goal of determining the proper time difference between the decay of the two B states. These were indeed directly observed, showing that the laws of microphysics are not identical whether time runs forwards or backwards, even though CPT-symmetry is maintained.

"In conjunction with the detection of both the Higgs boson (Chatrchyan et al. *Phys. Letts. B,* 2012, Aad et al. *Phys.Letts.B,* 2012) and gravitational waves (Abbott et al. *Phys. Rev. Letts.,* 2016), the direct observation of a time-reversal violation was a crucial breakthrough of the past decade which showed that the laws of microphysics may also describe irreversible processes without invoking a loss of information. For the first time, irreversibility appeared as a fundamental property of the physical realm and not as an emergent feature. The subtle CP violations (and corresponding T violation) of particle physics experiments can be expected to have a significant impact at the macroscopic level, the evidence for which seems to be everywhere. For example, according to Sakharov (*Sov. Phys. Uspekhi,* 1991) the CP-violation could explain the imbalance of matter and anti-matter abundance in the present-day Universe, and the previous possibility could be connected (via CPT-symmetry) with the irreversible accelerating expansion of the Universe (in accordance with the T violation)."

[51] So Roger Penrose speaks specifically of "*the principles that actually underlie the behaviour of our physical world*" (2004 [133], Preface). Clearly, if pressed everyone will say that the reality is one thing and our ideas of it are something else, but in actual practice we do not tend to distinguish the two things very sharply. This becomes very obvious when string theorists (and others) speak of "before the Big Bang" as a sensible idea (which it cannot be since time itself started then), or of "inflation" or "dark matter" (or the "multiverse") as realities. So here Penrose has apparently made a reality of the *principles*, where strictly speaking our formalisms cannot ever completely capture reality. And Penrose's book is a brilliant and seminal *tour de force*!

Additionally, Carlo Rovelli believes of "*the world*" that it "*does not exist in space and does not develop in time*" (Rovelli 2014 [134], ch.13). Instead he speaks, rapturously and persuasively (like a "poet"), of a "*world made up solely of interacting quantum fields the swarming of which generates—through a dense network of reciprocal interactions—space, time, particles, waves and light*" (*ibid.*). He does not explicitly distinguish the reality itself from our ideas of it, even if he does emphasise that our ideas may be mistaken. Ordinary readers (also including most scientists) are not in a position to judge the strengths and weaknesses of Rovelli's position, and therefore how Rovelli himself may be mistaken (and what shape a more correct view might take).

Then Frank Wilczek (2021 [92]) says in his *Afterword*: "*I have been at pains to be clear that science teaches us what is . . .* " This really sounds like an ontological statement. I am sure that Wilczek himself distinguishes the reality from our formalisms, but it is too easy for the ordinary reader to slide from a strict view into a comfortable (and mistaken) approximation.

As examples of (apparent) confusion beween symbols and reality, I have mentioned three books by by renowned physicists (two of them Nobel prizewinners: Wilczek 2004 [135], Penrose 2020 [136]), all with "*Reality*" in their titles. Physicists really do believe that they touch reality: we are here enquiring into what this means. It seems that a correct view is not very simple.

[52] Actually, Sadi Carnot's seminal treatment in 1824 relied on the false idea of *caloric*: it was Clausius who found the correct interpretation we still use (see Paul Sen, 2021 [45]). It was also Clausius who recognised that the *change* in the internal energy $U$ of the system is path-independent and therefore that $U$ is what we would now call a "function of state" (for a close discussion of this see Jennifer Coopersmith, 2015 [137]).

[53] The symbol "$S$" (denoting *entropy*) originated with Clausius, possibly indicating the integral ("*Summe*": $\int dQ/T$) he introduced to define the "second law" (Paul Sen, 2021 [45] p. 68, mentions "*a charming unsubstantiated story that he did so in honour of Sadi Carnot*"). In modern terms $S$ has the unit Joules per Kelvin (energy/temperature), where the idea of "*absolute temperature*" was clear to Clausius who already knew in 1854 that 0 °C = 273 K (accepting the 1848 value of William Thomson—later Lord Kelvin). "$W$" (supposedly from "*Wahrscheinlichkeit*", probability) denotes the number of different states the system can have. From this formula Boltzmann could derive the ideal gas law in what is now a textbook treatment.

[54] This statement appears to be controversial since conventional (simple) physics treatments hold that entropy is *extensive* (meaning, roughly, "double the size, double the entropy"; for example, *mass* is extensive, *colour* is intensive). However, it is very clear that this is an error. Entropy may indeed be extensive but only for certain sorts of systems. In principle it is *intensive* in general, as explained in detail by Walter Grandy (2008 [138], ch.5: "*The Presumed Extensivity of Entropy*"), who points out (as we also do) not only the anthropomorphic ("subjective") nature of the Partition Function, but also the necessity of considering the indistin-

guishability of *states* (not *particles*). This latter is also asserted by Karen Barad who insists (from an entirely different point of view) that "*the primary ontological unit is the phenomenon*" (Barad 2007 [24], p. 333).

Grandy (*ibid*.) also points out, citing Wolfgang Pauli's famous Lectures at ETH Zurich (conducted in 1952 [139]), that Boltzmann's equation for entropy (see Equation (1)) is determined up to $f(N)$, an arbitrary *function* (not merely a constant) of the number of states $N$ in the state space: "$S = k \{\ln W + f(N)\}$". Note that the *entropic* Liouville Theorem still applies (Parker and Jeynes 2021 [140]), meaning that $N$ does not necessarily have to be defined in statistical mechanics terms: it may instead be defined in terms of "degrees of freedom" which may also apply to small systems. Benoit Mandelbrot (1989 [141]) has elegantly explained some basic thermodynamic ideas which have bedevilled the discussion of small systems.

Grandy says, "*The effect . . . is to convince us that a proper definition of entropy should at bottom be theoretical . . . It is not an empirical question but a conceptual one*" (*ibid*. p. 68). There are of course many incorrect ways to think, but there is no mechanical method of finding a correct way.

55    The Partition Function of a system is an abstract representation of how the system shares out the total energy between its constituent parts.

56    His argument involves a calculation of the *entropy* of the system of resonators, and therefore also involves Boltzmann's relation (which he gives correctly, that is, up to a constant factor and using the natural logarithm: "$S = k \ln W + constant$"). He obtains the values of both eponymous constants (the "Planck" and the "Boltzmann" constants, $h$ and $k$, which he calls "universal" or "natural" constants) by considering the accurate measurements of what we now call *Wien's displacement constant*, $b = 2.94$ mm.K, that had then recently appeared: he obtains $k = 1.346.10^{-23}$ J/K and $h = 6.55.10^{-34}$ J.sec. Today's accepted values are, $b = 2.90$ mm.K; $k = 1.381.10^{-23}$ J/K; $h = 6.63.10^{-34}$ J.sec. Note that Planck recognised that $\hbar c/k$ has the same unit as $b$ (where $\hbar \equiv h/2p$ is Dirac's "*reduced*" Planck constant and $c$ is the speed of light), and in fact $\hbar c/k \approx b$.

57    It is germane to mention that there remains an ongoing technical debate surrounding the meaning of Landauer's Principle, which John Norton alleges to be "unproved", and indeed based on "circular reasoning" (Norton 2013 [154]). The issues are intricate, but involve debate over the applicability to fluctuation theory and to systems not representable by statistical ensembles (such as black hole event horizons). The latest summary of this debate (Ladyman and Robertson, 2014 [155]) criticises Norton's results but also allows that at least some of his analysis appears to be valid. However, everyone believes that Maxwell's Demon fails, and the the Second Law remains valid (whether or not it is fundamental).

58    Parker and Jeynes (2019 [28]) also show that information and entropy should be considered (*contra* Brillouin) not as opposites, but as complementary (that is, orthogonal in complex Minkowski 4-space). And indeed, Parker and Jeynes (2020 [157]) use the Shannon information entropy explicitly to discuss the stability of fullerenes: "*for example, for $C_{60}$ . . . we can calculate an entropy . . . using the Shannon fragmentation metric*".

59    But note that treating the black hole in QGT as *unitary* (than which exists nothing simpler; Parker and Jeynes 2021 [160])—that is, ontologically primary—implies that NO "*internal configuration*" exists at that scale of description. That the alpha particle is also *unitary* in QGT (Parker et al. 2022 [161]) is not a contradiction (considering that the "constituents" of the alpha are 2 protons and 2 neutrons) since to give a QGT account of those constituents would require a change of scale. Of course, the black hole already has the scale of the Planck length (than which exists nothing smaller).

60    Here, the ideas of "*holomorphic*", "*maximum entropy*", and "*unitary*" are closely related: *holomorphic* (literally "the shape of wholeness") has its full mathematical meaning even though it is applied to physical entities; *unitary* (than which exists nothing simpler) on the other hand is related to philosophical ideas of monadology (Leibniz 1714 [163]; see also Thomas 2015 [164]), not to (for example) the mathematical idea of an Hermitian adjoint.

61    Parker and Jeynes (2019 [28]) have shown that the double-logarithmic spiral (DLS) is an eigenfunction of the *entropic* Hamiltonian, and that spiral galaxies are well-modelled (at zeroth order) by the DLS. Moreover, Parker and Jeynes (2021 [160]) have used the QGT formalism to show (i) that this (idealised) DLS system conforms to the *entropic* Euler-Lagrange equations; (ii) that its *entropy production* is therefore Noether-conserved; and (iii) that its entropy production is necessarily non-zero, having two components: one component is small and comparable to the Hawking radiation of the supermassive black hole at the galactic centre, and the other component is many orders of magnitude larger (and presumably comparable to the entropy created by the accretion behaviour of the black hole).

62    These include problems in the definition of the (presumed) holographic "surface" of the galaxy (see Parker and Jeynes 2019 [28]). That is, how *big* it is supposed to be. The astrophysicists treat this problem heuristically, by calculating how much "dark matter" is required as a function of distance from the galactic centre. However, it should be noted that the *stability* of spiral galaxies (which are ubiquitous in the Universe) is an immediate consequence of QGT. This geometrical stability is not easy to account for in standard treatments since the distances involved ($10^5$ light years for the Milky Way) preclude normal feedback mechanisms. But QGT is defined in *hyperbolic* space, and non-locality is built into it—see §4.5.

63    Joseph Liouville (Liouville 1838 [165]) obtained a result whose importance was recognised only subsequently: in 1844 Augustin-Louis Cauchy proved the related result for complex analysis that every bounded entire function must be constant, which follows from the fact (important in QGT) that holomorphic functions are analytic. But it was J.W.Gibbs who was the first to recognise the importance of this equation as the "fundamental equation" of statistical mechanics (Gibbs 1884 [166])

64     Arno Keppens (Keppens 2018 [167]) also independently derives the Bekenstein–Hawking expression by considering the consequences of the underpinning of Raphael Bousso's "holographic principle" (Bousso 2002 [168]) by the quantisation of space-time. Keppens proposes "*viewing the conjectured pre-geometric atoms of quantum space as the ontological micro-constituents of our emergent reality*", although QGT represents ontology as (if anything) scale-less—that is, unitary entities (which either have no "micro-constituents" or do not depend on them ontologically) may be found at any scale including the alpha particle (see Parker et al., 2022 [161]) and black holes (see Parker and Jeynes 2021 [160]). Moreover, Barad (2007 [24]) denies even the existence of "*ontological micro-constituents*". This emphasis on ontology is relevant to Keppens' work since he wishes to aim "*getting as close as possible to the very nature of reality*".

65     It is worth pointing out that Karen Barad approaches this from a fundamental point of view, by denying that "*objects have inherent properties*" (Barad 2007 [24], p. 293), explaining that this is a consequence of the BKS Theorem (Bell 1966 [169], Kochen and Specker 1967 [170]). That is, simple "*individual entities*" do not really exist as such, even if it is often very convenient to pretend they do. The way the local and non-local interact is yet to be described in detail, although Velazquez (2022 [71]) and Velazquez et al. (2022 [72]) coherently treat reversibility and irreversibility together in a novel way: they rely on a complexification of the analytical approach ("analytical continuation"). This is not to deny our own personal individualities (heaven forbid!), but to point out with the poet that "*no man is an island*" and that we are nothing if we are isolated, notwithstanding our individuality. We are locally ourselves even if we are also (non-locally) entangled with everyone else. We make our own (local) decisions, even if those decisions are strongly constrained by our (non-local) circumstances.

66     Note that QGT is defined in *hyperbolic* space: this is a necessary consequence of the logarithmic dependence expressed in Equation (1). It is hyperbolic space that generates the scale-independence of QGT expected from a thermodynamics which must be true at all scales if it is fundamental to physics.

67     Of course, none of these eminent scientists could have taken the recent developments in thermodynamics into account. But however eminent they are, no one really believes them when they claim that the arrow of time is unreal. So Iris Murdoch, speaking of "*simple and obvious facts*" (and referring to the prominent philosophers John M.E. McTaggart and George E. Moore), observes that "*McTaggart says that time is unreal, Moore replies that he's just had his breakfast*" (Murdoch 1962 [100]). Emily Thomas (2015 [164]) points out that the little-known British idealist Hilda Oakeley (1867–1950) had previously also criticised McTaggart (and by implication, Leibniz and other idealist monadologists). Matyáš Moravec (2022 [174]) traces the influence of Henri Bergson (1859–1941) on Oakeley and others, and the reception of Bergson's ideas after Bertrand Russell's ferocious attack on them in 1912. Russell's views on time have been very influential in the physics community to this day but may not survive contact with the new thermodynamics.

68     "Cause-and-effect" is a central feature of science: experiments are specifically *designed* to show that certain causes have certain effects. Nevertheless, in principle "cause-and-effect" is necessarily an approximation since everything is entangled with everything else (how do we determine the boundary between the "local" and the "non-local"?). As Graeber and Wengrow (2021 [11], p. 21) say, "One *must* **simplify** *the world in order to discover something new about it*" (my emphasis). This, surprisingly, is directly addressed by Sidney Dekker (2006 [36], p. 73) when, speaking of the attribution of "human error" in investigations of disasters, he asks "*What is the cause of the mishap?*", saying "*Cause is something we construct*". He goes on to insist (p. 78) "*there are no such things as . . . causes . . . In fact, there is no . . . origin [cause] anywhere (other than the Big Bang)*" precisely because everything is entangled with everything else—especially when things go wrong! Dekker quotes James Reason's (1990 [177]) analysis of the infamous Bhopal accident in 1984 as a prime example of "latent errors" buried in the system. What "caused" this disaster? It makes little sense (and is less use) to say that Union Carbide's management system "caused" it. Certain events triggered it, but viewed correctly these should be seen as "effects" themselves since they were built into the system already.

69     We should point out that QGT has surprising and surprisingly far-reaching consequences, such as the demonstration that DNA must be right-handed because of the Second Law (Parker and Walker 2010 [180]), a result later formally proved in a general treatment (Parker and Jeynes 2019 [28]).(see Appendix in Supplementary Materials).

70     Bohm's proposal is considered by Rovelli to violate his Hypothesis 2 (completion) of his *Relational Quantum Mechanics* (Rovelli 1996 [182]). However, neither Bohm nor Rovelli take account of Parker's *Principle of Least Exertion* (Parker and Jeynes, 2019 [28]) in any way, even though both of them give significant weight to the (physical) quantity Information in their different treatments. But Parker has shown that a physical system cannot be treated completely unless its info-entropy is also considered.

71     Velazquez et al. (2022 [72]) also treat the 2nd Law as axiomatic in a fundamental approach to reversible/irreversible processes (explicitly considering the Loschmidt Paradox).

72     Although in Biblical Hebrew there is no equivalent impersonal word for "*meaning*": [בִּינָה] (◂î·nāh) is invariably used specifically of someone's *understanding*; it is used only once in the Hebrew Bible as syntactically impersonal at Daniel 8:15 (translated "meaning" by Geneva 1560 [184]), but even in this late text it refers semantically to Daniel's *understanding* of the vision.

73     Charlotte Higgins (2022 [59]) comments: "*Ukrainian writers often tell me that now is not the time for literary experimentation, but rather for directness* and *documentary. What is clear though, is that they are experimenting almost despite themselves, as they edge towards the impossible task of expressing the inexpressible.*" True documentary involves speaking meaningfully of what may very well be ineffable when the circumstances are extreme. But it is when the circumstances are extreme that we most seek meaningful expression! Poetry (or poetic speech) is existentially fundamental.

74    Ariel Caticha (2021 [185]) takes an entirely different line here, commenting that the "*search for the meaning of entropy has proved so elusive*" precisely because we do not need to "*know what 'entropy' means—we only need to know how to use it*". In fact he says that "*entropy needs no interpretation*" (italics original). Additionally, Caticha treats "*information*" not as something strictly physical, as we do here, but effectively in anthropological terms, that is: as "*defined in terms of its effects on the beliefs of rational agents*" (underlining the complexity of the philosophical position), although it is clear that Caticha's purpose is to recast "information" as a model-making category. He considers *information* mathematically as a synthesis of Bayesian and Maximum Entropy methods of correct inference in the context of changing "information" (interpreted as constraints on the model).

75    JCGM: Joint Committee for Guides in Metrology; BIPM: Bureau International des Poids et Mesures (International Bureau of Weights and Measures). The JCGM is responsible for the GUM (the *Guide to the Expression of Uncertainty in Measurement*: JCGM 2008 [189]) and the VIM (the *Vocabulaire international de métrologie*: JCGM 2012 [190]). The essentially human nature of *measurement* is underlined by the insistence in the latest revisions of the GUM and the VIM that the measurement is not aimed at determining the "*true value*" (whose very existence is philosophically debatable) but instead seeks to specify a "*process of experimentally obtaining one or more quantity values that can reasonably be attributed to a quantity*" (JCGM 2012). This makes it very plain that, as Karen Barad says, it is the *phænomenon* (not the presumed entity) that is ontologically primitive (Barad 2007 [24], p. 333)

76    There is a very large literature on "realism" which we can only touch on here. Barad (2007 [24]) is as good an entry as any into the philosophical literature since the book opens with a detailed account and discussion of Bohr's physics-philosophy. But "reality" is regularly presented as a preoccupation of leading scientists: "*Process and Reality*" (Whitehead 1929 [14]), "*The Road to Reality*" (Penrose 2004 [133]), "*Reality is not what it seems*" (Rovelli 2014 [134]), "*Fundamentals: Ten Keys to Reality*" (Wilczek 2021 [92]). Although all these authors are realists, none of them are "naïve realists": as Rovelli observes, reality is not what it seems—it is elusive!

77    It might be thought that an argument from mathematics cannot be used for or against any doctrine of materialism. But Gödel proved what Whitehead suspected, that not even maths is a strictly *necessary* thing (since it is Gödel-incomplete). Nevertheless, maths is indispensible to our knowledge of physics (and much else besides). The strict materialist would claim knowledge is ultimately thought, and thought must ultimately be physical (neurons firing and suchlike) since nature is all there is. But this position looks irrational, given that our knowledge is necessarily Gödel-incomplete. Materialists should claim instead that "*thought may (or may not) ultimately be physical*" (and they *believe* that it is physical). However, we suspect not only that mind is immaterial, but also that such a position accords best with the evidence.

78    This usage of "*meta*" and "*metameta*" has been analysed logically in some detail for this context: as Masahiko Sato says, it "*plays a fundamental role when we define formal systems such as logical and computational calculi*" (Sato 2003 [192]).

79    "*Indivisible*" is a neologism of Henry More (More 1647 [193], see Gorman 2021 [128]), as is also "*indiscerpible*" (that is, physically indivisible, from the Latin *discerpere*—to tear asunder; now obsolete). We can now recognise the alpha particle as *indiscerpible* (since it is naturally very stable indeed) although it is not strictly *indivisible*, being composed of two protons and two neutrons. But the Quantitative Geometrical Thermodynamics (QGT) discussed at §4.4 above treats it as a *unitary* entity—in other words, an *atom* in 17th century terms!

80    It is hard to overestimate the importance of the advent of the microscope to natural philosophy. Suddenly things became much more complex and beautiful than anyone had imagined. This is excellently reviewed and described by Kevin Killeen (Killeen 2017 [194]), who thus provides a corrective to our ideas of the emergence of "scientific modernity" which (as he puts it) "*is often still viewed as a sad but necessary putting aside of the poetic*".

81    The Ontological Argument (Anselm 1078 [116]) has puzzled philosophers for a millennium, and was written as a *prayer*, which is also a form of poetical speech. The poet recognises that not everything can be explained, and the philosopher should recognise that this ineffability is a property of *logic*. Logical arguments are necessarily incomplete, but the *coherence* of all effective poetry is a property of *rationality*. The philosophers should take more account of the poetical properties of the Ontological Argument.

82    E.R.Dodds' influential idea of "*irrational intuition*" in his famous book of a generation ago (Dodds 1951 [121], p. 217 *passim*) is in our terms quite mistaken since it is *rationality* itself that guides, structures and mediates intuition (see for example Polanyi 1958 [38]). Intuition may or may not be well-directed, but calling it either *rational* or *irrational* is a category error.

83    We give the ancient (*koine*) Greek text rendering the ancient Hebrew (Hebrew Bible c.500 BCE [195]; see §6) because the Hebrew original (with its vocalisation remembered separately) was not a *text* as we now understand it: the *Masoretic text* ("MT": essentially the "pointed" Hebrew text of Samuel ben Jacob) was only completed in modern times (1010 CE: the "*Leningrad Codex*"). The unpointed (original) Hebrew text is an *abjad*, that is, a very highly compressed mnemonic (consonantal) text: the vowels are preserved separately by the oral tradition (known as the *Masorah*). The MT encodes this vocalisation into the text itself by a sophisticated "pointing" system.

84    The Greek translation of the Hebrew (the *Septuagint*, "Lxx" [196]) was started in the 3rd century BCE by the Jewish scholars in Alexandria at the request of the Pharaoh (Ptolemy II, 285-247 BCE), and probably essentially finished by the 2nd century BCE. Note that the chapter numbering varies between the MT and Lxx for *Jeremiah* and *Psalms*. The Lxx is itself a canonical text for Christians since the (Greek) *New Testament* quotes it *verbatim* in many places.

     We use the English of the *Geneva* translation (1560 [184]) since this was widely reprinted and used in England up to and beyond the Restoration of Charles II (1660), remaining very influential through the substantial dependence on it of the King

James Version (1611, originally printed in a blackletter font) which became the dominant text in English until the mid-20th century. Verse numbers were first introduced by the Geneva scholars. The 1560 Geneva Bible was printed in a clear Latin (*not* blackletter) font. The *New Testament* of both *Geneva* and KJV are heavily dependent on Tyndale's 1526 [69] (blackletter) *New Testament*.

85　Bohm and Hiley (1993 [19], §14.1,p. 320) discuss this in some detail: "*Heisenberg . . . said very explicitly that the essential truth was in the mathematics. This view has become the common one among most modern theoretical physicists who now regard the equations as providing their most immediate contact with nature . . . without an equation there is really nothing to talk about. On the other hand, in the past we began talking about our concept of physical reality and used the equations to talk about them.*" They go on to speak of a "*new creative approach*" which may start either from (informal) physical ideas or from formal mathematical ones.

86　Observation shows that Parker and Jeynes (2019 [28]) correctly calculate an energy of $1.2 \times 10^{-15}$ J required to transform a DNA molecule to a different form: they also reasonably calculate an energy-equivalent galactic mass of the Milky Way as $2.3 \times 10^{59}$ J, an energy 74 orders of magnitude larger. Parker et al. (2022 [161]) also correctly calculate nuclear sizes: the atomic nucleus is some nine orders of magnitude smaller than the DNA molecule treated previously. But exactly the same (thermodynamic) methods are in use, as expected since the Laws of Thermodynamics are expected to apply at all scales.

87　The scalelessness in time has not yet been explicitly demonstrated in quantitative examples by QGT methods: these very new methods have not yet been developed to treat dynamical systems. However, Parker et al. (2022 [161]) point out that radioactive decay (the beta-decay process, which is logarithmic in time) is apparently correctly accounted for in their heuristic treatment of $^6$He decaying to $^6$Li.

88　"*Conjugate quantities*" are those related by the "Uncertainty Relations" (position and momentum, time and energy *etc.*). These relations are discussed in considerable depth by Karen Barad, both of the position/momentum ($x$ and $p$) and the time/energy ($t$ and $E$) conjugate pairs are given explicitly in terms of the Planck constant $h$: $DxDp \geq h/4p$; $DtDE \geq h/4p$ (Barad 2007 [24], p. 300: as Equation (2) and an inline equation). Note that Barad draws a sharp philosophical distinction (reinterpreting the D operator) between Heisenberg's "*Uncertainty Relations*" (which are epistemological) and Bohr's "*Indeterminacy Relations*" (which are ontological).

89　If we only know our *ideas* of the thing (rather than the thing-in-itself) why should we believe that the thing-in-itself has any reality independent of us? This gives rise to what is known as (some version of) "external world skepticism". Grace Helton discussed this in detail, commenting: "*if solipsism is in fact refutable, it is at least not obvious what form its refutation might take . . . My aim is . . . showing that it is not a straightforward matter to see how these difficulties might be overcome, with the result that solipsism cannot be dismissed out of hand*" (Helton 2021 [199]).

90　Physicists who claim to be solipsists do exist. Professor Joan Vaccaro's position is subtle, but she asserts (in part): "*"I" do not exist objectively ... This is the fundamental absurdity of life.*" She goes on to "recite" the "**Solipsist's Plea**": *Oh Universe!/I assume that you exist./Let me feel your far flung matter, / so that your illusion may persist* (Vaccaro 2022 [200]). But she was not willing to debate this with me (private communication, 9 March 2022).

91　Barad criticises Frayn, but does not note that Frayn's device of a *triple* rendering of the Heisenberg-Bohr meeting is a nod to Kierkegaard's similar (*quadruple*) device in his terrifying account of Abraham and Isaac in *Fear and Trembling* (Kierkegaard 1843 [123]) that was so influential in the 20th century.

　　Kierkegaard was translated into English rather late (1930s and later). But he was translated into German repeatedly from the later 19th century. Christoph Schrempf had translated Kierkegaard's complete works into German by 1922, but these were renowned as being rather free. Parts of *Fear and Trembling* and other works were published by Hermann Ulrich in 1925 in a much closer German rendering (Ulrich 1925 [203]).

92　The reason that we cannot say what "an electron" *is* is that all electrons are entangled with each other (being indistinguishable) so that "an electron" does not exist at all as an individual entity. Karen Barad insists on this, observing (in the context of a discussion of the implications of Bell's Theorem) "*that it is no longer possible to embrace the metaphysics of individualism*" (Barad 2007 [24], p. 292).

　　Surprisingly, this was underlined by the poet some three millennia ago, who asserted to God *silence is praise* "[דֻמִיָּה תְהִלָּה] *ḏu·mî·yāh ṯə·hil·lāhʰ* (Psalm 65:2, using the Masoretic Text 1010CE, and following the English translation of Alter 2007 [204]) in the context of God's power both over his Creation (he "*stills the roar of the sea* ") and also over "*the tumult of nations*" (*ibid*. v.8) but starting with the poet's own shortcomings ("*crimes*") on which God is silent (he "*covers*" them: [תְּכַפְּרֵם] *ṯə·ḵap·pə·rêm, ibid*.v.4). Here the personal and indeterminate nature of the ontology of knowledge is explicit, and the poem is undoubtedly early: David was c.1000 BCE and he or one of his musicians (also called "*seers*") may well be the poet. Note that this is the indeterminacy (or *silence*) of the eloquent poet, not of the baffled philosopher (Wittgenstein 1921 [76], §7).

　　This is not an isolated instance. Psalm 19 juxtaposes the wordless praise of the Creator by the cosmos ("*day after day they pour forth speech . . . they use no words . . . yet their line goes out into all the earth*", v.3) with the delight of the poet in God's law ("*the law of the Lord is perfect, refreshing the soul* ", v.7). The physicist knows the same delight. There are also other examples. The via *negativa* has ancient roots.

　　It is not only the ancient poets who notice the apparent entanglement of everything. Jessica Nordell, in the context of a discussion of police bias, also points out that people's very health itself may be measurably compromised by an inability to

recognise our entanglement with each other as social beings: "*Bias requires a firm division between the self and the other. But if the distinction beween the concept of 'I' and the concept of 'you' is not so clear, bias loses its meaning.*" (Nordell 2021 [15], p. 128).

93     It is only rarely that we manage to see things we do not recognise. Michael Berry lauds Newton, saying that he "*was a very careful observer, and paid particular attention to phenomena that resisted his attempts to explain them*" (Berry 2002 [21]).

94     For convenience we follow the dating of *New Testament* texts by J.A.T.Robinson (1976 [17]).

95     The Creation narrative in Genesis in the form we have it almost certainly dates from around the Temple reforms of Josiah in the 7th century BCE: see II Kings 23 and Margaret Barker's gloss on this (Barker 1987 [205], p. 142). The modern Hebrew script we use today was introduced in the early 5th century BCE by Ezra the Scribe. It has been widely thought that this Creation narrative represented a theological innovation at that time, since comparable tropes had not been found in the surrounding cultures. But this position is certainly mistaken, since a reanalysis (Korpel and de Moor 2014 [206], see Jeynes 2014 [207]) of two tablets from the Ugarit tell at Ras Shamra confidently dated late 13th century BCE show that the Ugaritic creation story has many remarkably close similarities with the Hebrew one. Therefore, the traditional ascription of the Hebrew story to Moses (perhaps 16[th] century BCE or earlier) cannot be rejected out of hand.

96     For convenience, and to be definite, we accept the traditional account of the genesis of the Hebrew text of the Hebrew Scriptures that we have in our hands as being the work of Ezra the Scribe working in Babylon around 500 BCE, although nothing much rests on this dating in this present work. Ezra introduced (or restored) the "Assyrian script" that is used to this day in Israel: previously the common (Canaanite) script was completely different. All this is discussed in detail in the Supplementary Materials Appendix part.

97     To repeat: "Modern" Hebrew here means the unpointed script that Ezra the Scribe introduced a mere two and a half millennia ago. This script is still today in everyday use in Israel. If modern Israelis wish to use an "archaic" script, they use the Masoretic pointing system introduced by Samuel ben Jacob (and the Masoretic scholars) only a thousand years ago.

98     The alternative possibility that such artefacts were created for the antiquities market is considered in the Supplementary Material and ruled out on the grounds that the palaeo-Hebrew is meaningful.

99     It is necessary to limit the scope of this work to exclude questions of ethics, but these persistently obtrude whenever poetics and our basic approach to knowledge are in view. Where is *Justice*? Jon Cruddas (2021 [213], p. 20) analyses the political consequences of "*epochal technological change*" in a deep treatment that insists on the necessity of an "*ethical orientation*" capable of addressing the central political questions of "*justice and human labour*" (p. 26f). The revolutionary advances in thermodynamics we have described in §4 are indeed expected to result in the "epochal technological change" Cruddas speaks of: and therefore our exclusion of ethical questions is in principle unrealistic. Nevertheless, we leave such questions for further work.

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
