# Peer review of "The Poetics of Physics"

_philosophies, doi:10.3390/philosophies8010003_

Round 1

Reviewer 1 Report

This was a well argued and passionate paper making the case for the poetics of physics in a compelling and convincing manner.  The author uses a range of appropriate references and is clearly well-versed (no pun intended) in both the philosophy of physics and the science of poetry.  

I have two suggestions the author could take on board.  The first relates to references. I think that there is a case to be made for including more poetry scholarship, specifically WB Yeats's essay The Symbolism of Poetry (1900).  I would be inclined to include this after the section on Murdoch and metaphor.

The second point is stylistic and a matter of taste - so can be judged accordingly.  I would reconsider the heavy use of exclamation marks throughout the text.  I am a firm believer of 'less is more' when it comes to these.

Author Response

Reviewer 1:

This was a well argued and passionate paper making the case for the poetics of physics in a compelling and convincing manner.  The author uses a range of appropriate references and is clearly well-versed (no pun intended) in both the philosophy of physics and the science of poetry.  

I have two suggestions the author could take on board.  The first relates to references. I think that there is a case to be made for including more poetry scholarship, specifically WB Yeats's essay The Symbolism of Poetry (1900).  I would be inclined to include this after the section on Murdoch and metaphor.

We thank the Reviewer for these encouraging comments.

Yeat’s essay is indeed relevant. His line about “making and unmaking mankind”, which he repeats (in II), is especially germane to our argument since the point of physics is to change the world specifically in response to how we ourselves perceive it.  It was very helpful of this Reviewer to draw our attention to this influential essay, which as suggested we will in turn draw attention to.   

The second point is stylistic and a matter of taste - so can be judged accordingly.  I would reconsider the heavy use of exclamation marks throughout the text.  I am a firm believer of 'less is more' when it comes to these.

Actually, we believe the same! Passion should be communicated by the substance of what is said,  not by the punctuation!  It is sobering to consider that even including spaces between words is a relatively modern innovation (on the timescale reviewed in our paper), so that punctuation should really be treated as a luxury (that is, sparingly). 

However, there are 42 exclamation marks in the MS (with 24 in the main text), almost invariably associated with a rhetorical flourish (such as, “heaven forbid!”). This paper is a rather peculiar work which is explicitly more “passionate” than is usual (as acknowledged by the Reviewer): indeed its purpose is to display the passion that must underpin knowledge, and the desire for knowledge. This is precisely the reason for the poetic epigraph and the poetic epilogue bookending the work. In this case we have permitted ourselves more punctuation than usual on the grounds that our rhetorical flourishes have a (poetic) purpose that is justified, and we wish to underline their validity. Even so, we have reduced the number of these marks by a third (to 28).

Reviewer 2 Report

In the paper, the author makes a very strong argument: “The knowledge of both the poet and the scientist is, ultimately, the same sort of thing (even if they use vastly different methods)” (p. 4). We would need to rediscover the poetics of physics to be able to express our knowledge of what things are in themselves and to discern how it is that we know (p. 46). The author's undertaking is very ambitious and full of intriguing insights, such as the analysis of different uses of the concept of entropy, an analysis aimed at showing the connection between poetic language and scientific language. It seems to me that the underlying intent of the paper is to show how knowledge is always a personal activity, connected to our ability to see, to focus on the object of our attention. Although analytic language is designed to be unambiguous, there is an ambiguity inherent in all language that cannot be eliminated, but on the contrary it must find a place in any scientific argumentation. The attempt to explore the idea of knowledge itself thus leads the author to overcome "the usual Hellenocentric accounts that lead us to believe that philosophy began with Socrates" (p. 3).

I list some suggestions for improving the paper:

1.      I think it is necessary to streamline the text in some places. Summarize or shorten for example part 4 "Entropy and Information".

2.      Better clarify, as early as the introduction, the different passages of the argument. The path taken by the author is not always clear.

3.      I would also suggest citing M. Polanyi and H. Prosch's "Meaning," in which the relationship between science, art and myth is explored.

Author Response

In the paper, the author makes a very strong argument: “The knowledge of both the poet and the scientist is, ultimately, the same sort of thing (even if they use vastly different methods)” (p.4). We would need to rediscover the poetics of physics to be able to express our knowledge of what things are in themselves and to discern how it is that we know (p.46). The author's undertaking is very ambitious and full of intriguing insights, such as the analysis of different uses of the concept of entropy, an analysis aimed at showing the connection between poetic language and scientific language. It seems to me that the underlying intent of the paper is to show how knowledge is always a personal activity, connected to our ability to see, to focus on the object of our attention. Although analytic language is designed to be unambiguous, there is an ambiguity inherent in all language that cannot be eliminated, but on the contrary it must find a place in any scientific argumentation. The attempt to explore the idea of knowledge itself thus leads the author to overcome "the usual Hellenocentric accounts that lead us to believe that philosophy began with Socrates" (p.3).

This is a nice summary, and we thank the Reviewer for reading our work so carefully.

I list some suggestions for improving the paper:

I think it is necessary to streamline the text in some places. Summarize or shorten for example part 4 "Entropy and Information".

This work has already been very heavily edited over a period of several years. We agree that section §4 seems rather long: however, the reason for this is that actually the ideas basic to thermodynamics are acknowledged as conceptually very difficult. Moreover, we have ourselves been involved in what we regard as deeply important (perhaps even seminal) work to develop these ideas in a completely unexpected direction that we think will become very important. And the devil is in the detail! Therefore we regard the current balance of the work as unavoidable. There is copious detail included,  but all of it is necessary for those who wish to really follow the argument (which is emphatically not a conventional one, in scientific terms). We know that only some readers (perhaps rather few initially) will fall into this category, but we suspect that there will be more of them as time passes and the effects of the new thermodynamics become stronger. But we will look again for redundancies in the text.

Better clarify, as early as the introduction, the different passages of the argument. The path taken by the author is not always clear.

This is a helpful suggestion,  thank you.

I would also suggest citing M. Polanyi and H. Prosch's "Meaning," in which the relationship between science, art and myth is explored.

I look forward to reading this book, which I have on order. Hopefully we can add comments in proof.

Reviewer 3 Report

The author of the article devotes his research to demonstrate that the epistemology and ontology of physics are based on poetic language. To do this, the author of the article refers to modern achievements in the field of thermodynamics. This intention is positive from the standpoint of philosophical reflection since the scientific metaphor is part of the modern scientific language. The presence of such a voluminous epigraph in a scientific article looks somewhat ambiguous. Such epigraphs are good in a lecture or monograph. Also puzzled by the epilogue on pages 48-49. It has nothing to do with text. After reading the article, I had the feeling that this text is the result of a very large and high-quality work. The author has studied science and is familiar with the texts of scientists. At the same time, I will express some notes that the author must take into account to finalize the text. First, in this test the material of two articles. No less, if not more. Moreover, blocks of text are qualitatively different. Therefore, the author should dwell on Maxwell (namely, he very clearly states that the scientific language is metaphorical, that is, the terms carry several meanings; science must recognize that the unambiguity of expression is an ideal). An appeal to the texts of A. Einstein can be very clearly presented. The author should refer to his work. Then go to section 7. In the discussion of information, entropy is a text with a different style and direction. It is knocked out of the sound of the poetic language, to which the author aspires and the style of the text corresponds to these sections. Secondly, the author should be more correct in his statement about the beginning of the establishment of the foundations of physics since the time of Galileo Galilei. This point is stated already in the Abstract. We must not forget that already Aristotelian physics was based on philosophical foundations. If the author insists on his position, then the author should prescribe "what physics?". Thirdly, the author must pay attention to the presentation of the text. It is very disadvantageous to use such large subscript blocks. What is considered important should be present in the body text. The author writes a scientific article, so explanations, definitions, and references can have their rightful place in the body of the text, and not be put out of brackets. In general, the text is interesting from a philosophical point of view but it requires serious revision, which will allow the author to achieve clarity and clarity in the justification of the claimed issue.

Author Response

The author of the article devotes his research to demonstrate that the epistemology and ontology of physics are based on poetic language. To do this, the author of the article refers to modern achievements in the field of thermodynamics. This intention is positive from the standpoint of philosophical reflection since the scientific metaphor is part of the modern scientific language. The presence of such a voluminous epigraph in a scientific article looks somewhat ambiguous. Such epigraphs are good in a lecture or monograph. Also puzzled by the epilogue on pages 48-49. It has nothing to do with text. After reading the article, I had the feeling that this text is the result of a very large and high-quality work. The author has studied science and is familiar with the texts of scientists. At the same time, I will express some notes that the author must take into account to finalize the text. First, in this test the material of two articles. No less, if not more. Moreover, blocks of text are qualitatively different. Therefore, the author should dwell on Maxwell (namely, he very clearly states that the scientific language is metaphorical, that is, the terms carry several meanings; science must recognize that the unambiguity of expression is an ideal). An appeal to the texts of A. Einstein can be very clearly presented. The author should refer to his work. Then go to section 7. In the discussion of information, entropy is a text with a different style and direction. It is knocked out of the sound of the poetic language, to which the author aspires and the style of the text corresponds to these sections. Secondly, the author should be more correct in his statement about the beginning of the establishment of the foundations of physics since the time of Galileo Galilei. This point is stated already in the Abstract. We must not forget that already Aristotelian physics was based on philosophical foundations. If the author insists on his position, then the author should prescribe "what physics?". Thirdly, the author must pay attention to the presentation of the text. It is very disadvantageous to use such large subscript blocks. What is considered important should be present in the body text. The author writes a scientific article, so explanations, definitions, and references can have their rightful place in the body of the text, and not be put out of brackets. In general, the text is interesting from a philosophical point of view but it requires serious revision, which will allow the author to achieve clarity and clarity in the justification of the claimed issue.

We thank the Reviewer for a careful reading and these detailed comments which we respond to in turn.

The first thing to say (contra the Reviewer’s view) is that this is not a “scientific article” It is a meta‑scientific article (see our explicit discussion of “metaphysics” at footnote #14). Our whole purpose is to give a new critique of what “doing science” is about, and clearly this critique cannot logically be couched purely in scientific terms since one must resolve the recursion somehow. We therefore systematically explain in philosophical terms how to do this (which indeed we proceed to do).

The second thing to affirm is that we make no mistake in placing the most visible foundations of what we now call “physics” in the time of Galileo. We do not do “Aristotelian physics” today! We had not thought it necessary to labour this point, but the contest in the 17th (and even the 18th century in England) of the modern view with the Aristotelian orthodoxy (which was underpinned by five centuries of Thomist theology) is now a matter of legend (and also very substantial scholarship). Of course, Galileo himself had indispensible scholastic (and Muslim) antecedents, so that the popular account of the “rise of modern science” is largely mythical, but we don’t want to go into all this here.

The third thing it is necessary to say is that we do not wish to split up this treatment into two articles. We note that the Aims of Philosophies include:

“… philosophical reflection concerning themes at the intersection of philosophical, scientific, technological, and cultural studies [seeking] much broader perspectives that integrate the diverse intellectual tools developed in a large variety of scientific disciplines and philosophical systems. The ultimate goal of the journal is to reach for a synthesis of knowledge using the collective wisdom of diverse methodologies.

      Thus, in addition to works devoted to the more traditional themes … we invite articles or essays searching for new methods of inquiry. The journal is also for those who want to cross the borders between different scientific and philosophical disciplines or between different cultural paradigms of intellectual inquiry in the search for better understandings of traditional subjects of study; the journal is suitable for a wide range of philosophical and scientific questions and problems.

… We aim to encourage scientists and philosophers to publish their … philosophical reflections in as much detail as necessary to achieve high quality, self-contained works. There is no restriction on paper length.”

We appreciate that the work is very wide-ranging over fields never (to our knowledge) treated together: this does make it somewhat forbidding (and in fact it is very dense: it should ideally be expanded to monograph length for accessibility). We apologise to the Reviewers for thus making their work very hard, but what we do not want to do is to split the work up – it is precisely the connections we wish to highlight. 

The Reviewer starts by claiming that we wish to: “demonstrate that the epistemology and ontology of physics are based on poetic language”. This is not quite correct.  Actually, we address this directly in §5.1 where we say, “It should be obvious that although the ontology of the thing (its thinginess) and its epistemology (how we know it) are intrinsically separate ideas, yet in any specific case the two must be inextricably bound together. We cannot know anything about the thinginess of the thing without also knowing how we know.” Our purpose is actually to demonstrate that we cannot know any thing (including physical things) without poetic language. In the course of our demonstration we affirm that Karen Barad is correct in principle to speak of “onto-epistemology” – that is to make explicit the necessary interconnectedness of the two ideas of ontology and epistemology.

The Reviewer also claims that “the scientific metaphor is part of the modern scientific language”, later making clear that “metaphor” indicates an irreducible ambiguity (“the terms carry several meanings”). So far so good. Nevertheless, qualifying “metaphor” and “language” (as “scientific”) could be seriously misleading. We use language to speak of whatever things happen to interest us: these could be scientific things too. And of course a scientific discourse has a certain accepted form (which manifestly changes over time!). The language is (as always) adjusted to the audience: one does not speak in the same sort of language to a 9‑year-old as to a teenager or a professor of physics. But language considered semantically is never anything which should be considered “scientific” language. We do address these sorts of issues directly in the paper: even if our approach may appear oblique our insistence on the semantic commonality between languages (including English, German, Palikur, mediaeval Italian, Latin, koine Greek and Hebrew) is obvious. Actually, it is entertaining to comment that Galileo’s book “The Assayer” (1623) was specifically written in a conversational (and very amusing) Italian to make his seminal philosophical and scientific points to a very wide audience. How custom changes with time!

The Reviewer considers our Epigraph “voluminous” and is puzzled by our Epilogue, considering it unrelated to the main text. In some circumstances it is good to abstract resonant lines from poems: we cite both Rachel Carson’s “Silent Spring” (taken from Keats) and Karen Barad’s “Meeting the Universe Halfway” (taken from Alice Fulton). But generally poems should be treated as atoms, that is, indivisible (and Cassandra Gorman points out that the interest in “atoms” in 17th century English poetry – stimulated by the concurrent scientific advances – starts from precisely this point). Therefore we make no apology for reproducing Aldiere Orlando’s poem in full.

On the poem by Jeynes (the Epilogue), the Reviewer is considering an anonymised MS and therefore may not realise that Jeynes is actually the lead author of the article. We regard it as legitimate for authors to make their own view plain, where this is appropriate. Besides, the poem is very clearly related to the paper. It starts with the line “God is one”: all scientists must believe that “truth is one” (stated explicitly at footnote #4) and whether or not you believe in God you can treat the two statements as equivalent (which in fact is what Einstein did). Moreover, the Epilogue is a meditation on the rationality of reality (expressed in numerical terms) which is the same sort of thing (and expressed in the same sort of terms) as the heavily allusive palaeo-Hebrew poem interpreted into English in the main text. But this is hardly the place to engage in literary criticism.

The Reviewer suggests that we cite Einstein. We are quite reluctant to do this since Einstein’s views are very well-known and any brief citation is likely to be clichéd. But the Reviewer has also noticed some marked differences in literary style between the sections. It is not clear to us that this is (or even should be considered) a bad thing.

On the question of footnotes, it can be remarked that modern books (monographs) usually have very extensive endnotes. I recall that Michael Frayn, in his “The Human Touch” (2006), included in one of his footnotes a metrical (rhyming) translation of three stanzas of Pushkin’s classical poem Eugene Onegin (including the Russian original in his footnote). I checked with Russian friends that the translation was a good one. The point is that we make a very wide-ranging argument – but we insist that it is an argument, not merely an unsubstantiated polemic. This means that we need the scholarly apparatus to back it up. Which it is only reasonable to include as footnotes – the text is dense enough already: to try to include this essential material would make it unreadably unwieldy.

Round 2

Reviewer 1 Report

I'm happy with the changes.  I very much enjoyed reading this article.

Reviewer 2 Report

The changes that have been made to the text have improved the work. The general structure of the text is clearer. I think the article can be accepted in present form.